



# Design, characterization, and first field deployment of a novel aircraft-based aerosol mass spectrometer combining the laser ablation and flash vaporization techniques

Andreas Hünig[1,2], Oliver Appel[1,2], Antonis Dragoneas[1,2], Sergej Molleker[1,2], Hans-Christian Clemen[2], Frank Helleis[2], Thomas Klimach[2], Franziska Köllner[1], Thomas Böttger[2], Frank Drewnick[2], Johannes Schneider[2], and Stephan Borrmann[1,2]

[1]Institute for Atmospheric Physics, Johannes Gutenberg University, Mainz, Germany
[2]Max Planck Institute for Chemistry, Mainz, Germany

*Correspondence to*: Stephan Borrmann (stephan.borrmann@mpic.de)

**Abstract.** In this paper, we present the design, development, and characteristics of the novel aerosol mass spectrometer ERICA (ERC Instrument for Chemical composition of Aerosols) and selected results from the first aircraft-borne field deployment. The instrument combines two well-established methods of real-time in-situ measurements of fine particle chemical composition. The first method is the single particle laser ablation technique (here with a frequency-quadrupled Nd:YAG laser at λ=266 nm). The other method is a combination of flash vaporization and electron impact ionization (like the Aerodyne aerosol mass spectrometer). The aerosol sample can be analyzed with both methods, each using time-of-flight mass spectrometry. By means of the laser ablation, single particles are qualitatively analyzed (including the refractory components) while the flash vaporization and electron impact ionization technique provides quantitative information on the non-refractory components (i.e., particulate sulfate, nitrate, ammonia, organics, and chloride) of small particle ensembles. These techniques are implemented in two consecutive instrument stages within a common sample inlet and a common vacuum chamber. At its front end, the sample air containing the aerosol particles is continuously injected via an aerodynamic lens (ADL). All particles which are not ablated by the Nd:YAG laser in the first instrument stage continue their flight until they reach the second instrument stage and impact on the vaporizer surface (operated at 600 °C). The ERICA is capable of detecting single particles with vacuum aerodynamic diameters ($d_{va}$) between ~180 nm and 3170 nm ($d_{50}$ cut-off). The chemical characterization of single particles is achieved by recording cations and anions with a bipolar time-of-flight mass spectrometer (B-ToF-MS). For the measurement of non-refractory components, the particle size range extends from approximately 120 nm to 3.5 µm ($d_{50}$ cut-off; $d_{va}$), and the cations are detected with a C-ToF-MS (compact time-of-flight mass spectrometer). The compact dimensions of the instrument are such that the ERICA can be deployed on aircraft, ground stations, or mobile laboratories. During its first deployments the instrument operated fully automated during 11 research flights on the Russian high-altitude research aircraft M-55 *Geophysica* from ground pressure and temperature up to 20 km altitude at 55 hPa and ambient temperatures as low as -86 °C.

## 1    Introduction

Beyond the experimental determination of physical aerosol properties, detailed measurements of the chemical composition of aerosol particles are essential for studies in the context of urban pollution, health effects, cloud formation, radiative transfer in the atmosphere, and climate change (See for example Fuzzi et al. (2015)). The chemical composition can provide information on the aerosol source −natural or anthropogenic−, and on the state of chemical and physical processing of the particles while aging during transport (IPCC, 2013; Seinfeld and Pandis, 2016).

Besides offline methods, which involve particle collection on suitable substrates by impactors or filter samplers followed by subsequent laboratory analyses (Elmes and Gasparon, 2017), in situ, real-time measurements adopting aerosol particle mass spectrometry have become a widespread established tool. For the implementation of aerosol mass spectrometry two





complementary measurement techniques are commonly used. One uses a pulsed laser to vaporize and ionize individual submicron to micrometer sized particles. The resulting ions are injected into a time-of-flight mass spectrometer (Suess and Prather, 1999). In terms of the deliverables, with this method single particle mass spectra of refractory and non-refractory components like of soot, salt, mineral dust, and meteoric dust particles, as well as metal-containing particles can be detected.

The other method is based on thermal vaporization and electron impact ionization (Davis, 1973), to quantitatively measure non-refractory species (sulfate, nitrate, ammonium, chloride, and organic compounds) in ensembles of particles. While the latter method provides quantitative mass concentrations of non-refractory components, the mass spectrometer signals of the previous method can only be used for the identification of the ions itself and not for determination of absolute mass concentrations. Within certain limitations this may become possible, if the data of other instruments are included in the analysis

(e.g., in Froyd et al. (2019)). Details on the methodologies, limitations, and considerations of the inherent experimental errors of these measuring techniques can be found in Kulkarni et al. (2011) and the references therein. Compact and mobile online instruments based on these methods have been deployed on research aircraft to measure particle chemical composition at high temporal and spatial resolution. The PALMS (Particle Analysis by Laser Mass Spectrometry; Murphy et al. (1998)) operated at altitudes of up to 20 km. Other aircraft-based, online single-particle laser ablation aerosol mass spectrometers, which are

operated at lower altitudes, are for example the A-ATOFMS (Aircraft Aerosol Time-Of-Flight Mass Spectrometer; Pratt et al. (2009)), the ALABAMA (Aircraft-based Laser ABlation Aerosol MAss spectrometer; Brands et al. (2011) and Clemen et al. (2020)), and the miniSPLAT (miniaturized version; Single Particle Laser Ablation Time-of-flight mass spectrometer; Zelenyuk et al. (2015). The thermal vaporization and electron impact ionization technique were deployed on research aircraft using a C-ToF-MS (Compact Time-of-Flight Mass Spectrometer) beside others by Bahreini et al. (2009), Morgan et al. (2010), Schmale

et al. (2010), Brito et al. (2018), Schulz et al. (2018), and Haslett et al. (2019), while a mAMS (mini Aerosol Mass Spectrometer) was used for example by Vu et al. (2016) and Goetz et al. (2018). An HR-ToF-MS (High-Resolution Time-of-Flight Mass Spectrometer) was adopted, for example, by Dunlea et al. (2007), Willis et al. (2016), and Singh et al. (2019). However, as these references show, for aircraft-borne measurements of aerosol chemical composition usually only one of the two mass spectrometry methods is implemented on a single aircraft mostly as consequence of limitations in weight and space.

Although several aerosol instruments can be operated simultaneously at one location during ground-based measurements or in a laboratory environment, e.g., Möhler et al. (2008), Dall'Osto et al. (2012), and Roth et al. (2016), up to now rarely two different aerosol mass spectrometers were available on the same aircraft, e.g., Murphy et al. (2006a), Hodzic et al. (2020), Schneider et al. (2019), Guo et al. (2021), and Köllner et al. (2021). Since the two techniques deliver complementary information on the aerosol composition and also cover slightly different size ranges, a single instrument implementing both

methodologies in one apparatus has obvious advantages, provided that it is sufficiently small and light. Also, since the repetition rate of high-power UV ablation lasers limits the number of particle detections per second, the addition of a thermal vaporization and electron impact ionization unit largely enhances the data yield for the particle analysis. Furthermore, the opportunities for measurements at high altitudes are rare, such that an aerosol instrument which provides a high information output is advantageous.

Subject of this paper is the ERICA (i.e., ERC Instrument for Chemical composition of Aerosols), which has been developed in our laboratories at the Johannes Gutenberg-University and the Max Planck Institute for Chemistry in Mainz. It is a hybrid instrument implementing both of the aforementioned particle vaporization and ionization methods in one single fully automated apparatus. The adopted techniques for automatizing the operation are detailed in the companion paper by Dragoneas

et al. (2021). The ERICA was deployed for the first time during the aircraft field campaigns of the StratoClim project (Stratospheric and upper tropospheric processes for better Climate predictions; Brunamonti et al. (2018), Bucci et al. (2020), and http://www.stratoclim.org, last access 30.08.2021) in August and September 2016 at the Kalamata International Airport (KLX; 37.07°N, 22.03°E, Kalamata, Greece) and during July and August 2017 at the Tribhuvan International Airport (KTM;



27.70°N, 85.36°E, Kathmandu, Nepal). Although the instrument was initially designed for implementation on the Russian high altitude research aircraft M-55 *Geophysica* (Borrmann et al., 1995; Stefanutti et al., 1999) and operation in the low particle number density environment of the upper troposphere and lower stratosphere (up to 20 km altitude), the ERICA can be integrated in suitable racks to be implemented into other research aircraft such as NASA's DC-8 (Schneider et al., 2021).

Furthermore, the ERICA can be used for a variety of ground-based stationary or mobile applications. In this manuscript we show the design of the ERICA, results from laboratory characterization measurements, as well as results selected for a proof-of-concept demonstration from the field campaign in Kathmandu, Nepal. The instrumental design and characterization is presented here in some detail (in particular in the supplement) in order to support potential design efforts of other groups, and to provide benchmark tests and values.

## 10  2   Instrument description

### 2.1   General principle and design of the ERICA

The principal configuration of the ERICA with its inlet system, the laser ablation section (denominated as ERICA-LAMS), and the thermal vaporization section (ERICA-AMS) is shown in Fig. 1. During aircraft operation the sample air flow is provided by a constant pressure inlet (Molleker et al., 2020) serving as a critical orifice at the instrument's front end. The

particles are focused in the aerodynamic lens (ADL) into a narrow beam and accelerated into the vacuum chamber, where they first reach the optical particle detection units (PDU1 and PDU2 in Fig. 1) of the ERICA-LAMS. Here, optical particle detection and sizing are realized via a particle flight time measurement by means of light scattering. For this purpose, two parallel continuous wave laser beams are directed onto the particle beam. The light scattered from the passing individual particles is focused by ellipsoidal mirrors onto photomultiplier tubes (PMTs). The time elapsing between the two light scattering signals

is used to derive its vacuum aerodynamic diameter $d_{va}$ (Hinds (1999), Jimenez et al. (2003a), Jimenez et al. (2003b), and DeCarlo et al. (2004)) by involving a calibration (see Sect. 3.2) and to determine the point in time the particle reaches the ablation spot of the ERICA-LAMS. If well positioned and timed, the particle gets vaporized and ionized by a triggered 266-nm UV pulse from a frequency-quadrupled Nd:YAG laser. The resulting cations and anions are accelerated into a bipolar time-of-flight mass spectrometer (B-ToF-MS) and detected by micro-channel plates (MCPs). A large fraction of the particles is not

ablated by laser pulses, either because the laser pulses miss the particles, or because the particles are too small for the optical detection. However, even most particles amenable for laser ablation, which pass through the ablation region, remain undestroyed, because the laser is firing at a limited maximum repetition rate of 8 pulses per second. These un-ablated particles pass through the B-ToF-MS region of the ERICA-LAMS and enter the continuously operating ERICA-AMS. There, in analogy to the Aerodyne AMS (aerosol mass spectrometer) principle, flash vaporization is followed by electron impact ionization. A

filament provides the electrons (70 eV) for ionization of the vapor molecules emanating from the vaporizer. The resulting cations are injected into the C-ToF-MS and eventually detected by its MCPs. The detectable particle size range ($d_{va}$) of the ERICA-LAMS is between ~180 nm and 3170 nm (see Sect. 3.3.3). However, the signal-to-noise ratio of optical particle detection is sufficient for particle time-of-flight calibration between 80 nm and 5 µm (see Sect. 3.2). The detectable particle size range of the ERICA-AMS is assumed to be the same as published by Xu et al. (2017) for the deployed lens type.: ~120 nm

to 3.5 µm. The design details of the ERICA-AMS are very similar to the Aerodyne AMS and are well-described in the literature (e.g., Jayne et al. (2000), Jimenez et al. (2003c), Drewnick et al. (2005), and Canagaratna et al. (2007). A fundamental difference to the commercial Aerodyne AMS is the use of a simple shutter mechanic instead of a chopper to block the particle beam for the reference background measurement.

Since the two instrument components share a single vacuum system, weight is saved due to common components like pumps,

power supply units, and vacuum chamber. Furthermore, the mechanical components of ERICA are designed to operate under the demanding conditions like heat and vibrations aboard an aircraft. The final design of the compact instrument was

implemented into an aircraft rack (Dragoneas et al., 2021) of 60 cm x 74 cm x 140 cm (height x width x length) with a total weight of 200 kg. Such a compact and light-weight design is essential for aircraft implementation, especially aboard a high-altitude aircraft. To visualize the orientation of the major components, a three-dimensional drawing of the instrument body is provided in Sect. S1.1 in the supplement as well as a photograph of the instrument mounted in the M-55 *Geophysica*-rack for

the StratoClim campaign.

### 2.2    Aerosol particle inlet and vacuum system

A continuous flow of sampled air containing particles enters the instrument via a critical orifice at the sample inlet (see Fig. 1). For ambient, ground-based measurements at ambient ground pressure, a pinhole diameter of 100 µm maintains a

volumetric flow rate ($\Phi_{ERICA}$) of 1.48 cm³ s⁻¹. However, in order to achieve a constant pressure in the ADL ($p_{ADL}$ = 4.5 hPa), the mass flow rate needs to be kept constant during flight operations with largely varying ambient pressures (for the M-55 *Geophysica* ranging from ground pressure to 50 hPa). If $p_{ADL}$ is not maintained constant, the transmission of the particles through the inlet into the vacuum system becomes altitude dependent (Zhang et al., 2002). For this purpose, a newly developed, automatically-controlled compressible rubber O-ring setup is deployed (Molleker et al., 2020). As ADL we integrated the

intermediate pressure lens IPL-013 (Peck et al., 2016; Xu et al., 2017) to focus the particles into a beam with sufficiently small divergence, i.e., less than the diameter of the vaporizer element at a distance of 55 cm downstream of the exit of the ADL. The lens itself contains six apertures (excluding the first critical orifice) with decreasing diameters (from 5.0 mm down to 2.9 mm) and the exiting particles are accelerated to velocities of up to 200 m s⁻¹. The inner end of the ADL tube protrudes from a holder plate through a radially sealed feed-through and is attached to a ball joint inside the first pumping stage of the vacuum chamber.

Four fine threaded screws, two of them with scale, enable the operator to tilt the lens precisely in two dimensions in order to adjust the particle flight direction so that it gets aligned with the vaporizer. By means of this design, the particle beam remained stable during flights even in the presence of vibrations caused by turbulence in the convective anvil outflows of tropical cumulonimbus at 12 to 18 km altitude.

The vacuum chamber was purchased from Aeromegt GmbH (Germany) and is a modified design of the LAAPTOF (Laser

Ablation Aerosol Particle Time-Of-Fight mass spectrometry; Gemayel et al. (2016)). During mobile operation on aircraft, two diaphragm pumps (model MD 1 VARIO SP, Vacuubrand GmbH + Co KG, Germany; pumping rate of 5·10² cm³ s⁻¹) yield 3 mbar for the backing pressure of the four-stage turbo pump. As in the Aeromegt LAAPTOF, the four-stage turbomolecular pump (see Fig. 1; SplitFlow 270, Pfeiffer Vacuum GmbH, Germany) is utilized for pumping the entire single particle mass spectrometer (ERICA-LAMS part). Its first pumping stage (PS1) operates at a rate of 3.0·10⁴ cm³ s⁻¹. The second pumping

stage (PS2; see Fig. 1) reduces the pressure of the chamber, containing PDU1, down to a pressure of 3·10⁻⁴ mbar (pumping rate of 1.55·10⁵ cm³ s⁻¹). A pinhole of 1.8 mm opening diameter placed perpendicular to the particle beam separates PS2 from the third pumping stage (PS3). For the particle detection unit PDU2, PS3 provides a vacuum pressure of 8·10⁻⁷ mbar with a pumping rate of 1.55·10⁵ cm³ s⁻¹. The fourth pumping stage (PS4) is attached to the chamber of the B-ToF-MS, which is maintained at a pressure of 4·10⁻⁷ mbar (pumping rate of 2.0·10⁵ cm³ s⁻¹). The particle detection unit PDU2 and the mass

spectrometer chamber are connected through a centered 4 mm-aperture.

The shutter unit (SU) separates the ERICA-LAMS mass spectrometer chamber from the ERICA-AMS ionizer vacuum chamber (see Fig. 1). The latter is separated from the SU by an orifice of 7 mm in diameter. The turbomolecular pump TMP5 (see Fig. 1; model HiPace® 80, Pfeiffer Vacuum GmbH, Germany; pumping rate of 6.7·10⁴ cm³ s⁻¹) is attached to the ionizer chamber keeping it at a pressure of 1·10⁻⁷ mbar. The turbomolecular pump TMP6 (model HiPace® 30, Pfeiffer Vacuum GmbH,

Germany) provides a pumping rate of 2.2·10⁴ cm³ s⁻¹ in the C-ToF-MS such that here the operational pressure is 2·10⁻⁷ mbar. Both HiPace® pumps, TMP5 and TMP6, are backed by the third pumping stage (PS3) of the SplitFlow pump.



### 2.3 ERICA-LAMS: Optical particle detection and sizing by light scattering

The setup of the optical single particle detection module for ERICA-LAMS consists of the two particle detection units PDU1 and PDU2 (see Fig. 1), based on the design of the ALABAMA (Brands et al., 2011; Clemen et al., 2020). Each of these particle detection units (PDU1 and PDU2) contains a continuous wave laser (LD1 and LD2), an ellipsoidal reflector, and a PMT (PMT1

and PMT2). By that, each particle passing the both laser beams causes two light scattering signals. The distance from the exit of the ADL to the focal point of the first ellipsoidal reflector (i.e., the first particle detection point) is 58.8 mm, the distance between the first and second detection point is 66.5 mm. A scheme of the geometry with dimensions of the ERICA is provided in Sect. S1.2 in the supplement. The laser sources are 150 mW UV-laser diodes operating at a wavelength of 405 nm (model SF-AW210 distributed by InsaneWare Deluxe, Germany) mounted in a heat sink.

The continuous wave laser light is focused by a plano-convex lens with a focal length of 4.02 mm to a $1/e^2$-radius $w_0$ of 30 µm (see Sect. 3.1). To reduce optical disturbances like diffraction fringes, the laser beam passes through a baffle of four apertures before the beam enters the detection region. Finally, approximately 40 mW of light illuminate the particle detection region. Each PDU is individually mounted on a disjoined micro XY translation stage (1 µm precision, model MKT 30-D10-EP by OWIS GmbH, Germany) and thus, they can be tilted in two dimensions for adjusting the laser foci onto the particle

beam.

In order to focus the light scattered by the individual particles to a detector, ellipsoidal reflectors (model E50NV-01 AF coated, Opti-forms, Inc., Temecula, CA, USA) were used. A detailed description of the ellipsoidal reflector setup can be found in Sect. S1.3 in the supplement.

A plano-convex lens collimates the scattered light towards the sensitive area of the PMT (model H10721-210, Hamamatsu

Photonics K.K., Japan). This design collects approximately 75 % of the total scattered light, not considering the losses at the pinholes. The acquired PMT signals are processed by an in-house built electronic board, hereafter referred to as trigger card (TC) following the design from the ALABAMA (Brands et al., 2011; Clemen et al., 2020).

### 2.4 ERICA-LAMS: Single particle laser ablation

The ablation laser is triggered by the TC that counts the particle flight time between the two PMTs, computes the precise time of the particle arrival at the "ablation spot" by multiplying the particle flight time between PDU1 and PDU2 by a factor, considering the geometry of the instrument (see Sect. S1.2 in the supplement). The triggering of the ablation laser considers the time span of 145 µs between triggering the laser flash lamps and the Q-switch. The precise values for this timing are set experimentally. Also, this unit triggers the high-voltage switches for the ion extraction.

As a consequence of the ablation laser pulse, the material of an aerosol particle is vaporized and ionized in a single step by a multi-photon process (Suess and Prather, 1999). For the ablation, a frequency-quadrupled Nd:YAG laser (model Ultra 50, Quantel, France) generates 6-ns-long pulses with 266 nm wavelength and typical values of around 4 mJ for the pulse energy. The simultaneously emitted additional light from the laser at wavelengths of 1064 nm and 532 nm is not filtered by a wavelength separator inside the laser head in order to minimize the number of optical elements in the light path before the

ablation spot.

As shown in Fig. 2, the emitted laser beam is oriented orthogonally to the particle flight axis and focused onto the particle beam by a plano-convex lens (anti-reflection coated model L-11612, Laseroptik GmbH, Germany). From the laser head, the beam is directed towards the mass spectrometer chamber by the dichroitic mirror DM1 (see Fig. 2; model G340722000, Qioptiq Photonics GmbH & Co. KG, Germany). This mirror also separates the UV light from the light at the other wavelengths (1064

nm and 532 nm) by reflecting > 99.5 % of its intensity. Only 12.6 % of the intensity of light at other wavelengths are reflected towards the ablation spot. The laser beam, now mostly consisting of UV light, enters and exits the vacuum chamber through uncoated and 3° tilted quartz glass windows in order to reduce back-reflections towards the laser head. The exiting beam is



directed by a second dichroitic mirror DM2 through an attenuating UV-absorbing glass filter (model UG11, Qioptiq Photonics GmbH & Co. KG, Germany) to an optical energy meter (EnergyMax™-USB, model J-25MB-LE, Coherent, Inc., USA) by which the energy of each pulse can be measured such that the laser pulse energy is detected and stored. The focal length of the lens ($f$ = 76 mm) is such that a high UV light intensity is centered at the "ablation spot" within the ionization region (see Fig.

1). This spot is located at the center between the extraction plates (EP) of the B-ToF-MS (from Tofwerk AG, Switzerland). For adjusting the beam waist of the UV laser to the ablation spot, the dichroitic mirror DM1 is mounted on a holder, which allows tilting the mirror with two degrees of freedom. The minimum beam at the ablation spot, which can be obtained with this setup, has a $\frac{1}{e^2}$-diameter $w_{0,dia}$ of 250 µm (see Sect. 3.1). For this fine adjustment, the focusing lens can also be moved in the direction towards the vacuum chamber. By means of this setup, the diameter of the laser beam at the location of the particle

beam can be enlarged from the minimum of 250 µm up to approximately 740 µm so that the energy density at the ablation spot can be reduced in a controlled way (Brands et al., 2011). After each pulse the laser has to idle for at least 120 ms in order to keep the output energy constant; this fact limits the repetition rate for ERICA-LAMS to 8 s⁻¹ (instead of the nominal 10 s⁻¹ according to the manufacturer's specification). This maximum repetition rate imposes a limit to the number of particles analyzed per time unit, which affects the spatial resolution for measurements from a fast flying aircraft.

For the analysis of the single particles, the generated ions are accelerated into the B-ToF-MS using an electric extraction field in the ablation region. The acceleration field between the EP is turned on only for the short time interval of 2 µs which is long enough for sufficient ion extraction. For this purpose, fast solid-state high-voltage transistor switches (model HTS 61-03-C, Behlke Power Electronics GmbH, Germany) are triggered by the TC and switch within 18 ns about 1.2 µs before the Q-switch actually fires the laser. During the time when no particles are detected by PDU1 and PDU2 or the ablation laser is in its idle

time, the EPs are connected to ground. Upon connection to ground, the electric field decays with an RC constant of approximately 10 ms. The HV switch was implemented, since the electric extraction fields cause charged aerosol particles to deviate from their straight flight direction (e.g., Chen et al. (2020) and Clemen et al. (2020)) and as a result, they might not hit the vaporizer in the ERICA-AMS part. In order to also reduce particle deflection caused by an electric field forming outside the ion optics, in addition the particle flight path through the ERICA-LAMS part is shielded by grounded plates. Inside the

time-of-flight mass spectrometers, reflectrons serve to enlarge the ion flight path (see Fig. 1) and to increase the mass resolution $R_{MS}$ to up to 700 (see Sect. 3.5.2).

The generated ion signal is picked up by MCPs (model MCP 40/12/10/8 D 46:1, Photonis USA Inc., Sturbridge, MA, USA), amplified, and collected by a digital oscilloscope (model Picoscope 6404C, Pico Technology, UK). The oscilloscope features four channels with 8-bit vertical resolution and a maximum sampling performance of 5 gigasamples per second (GS s⁻¹). The

time resolution is set to 1.6 ns per sample. The two MCP detector outputs for the anions and cations are conditioned and sampled concurrently by two separate channels with different input voltage ranges, an approach for extending the dynamic range of the A-to-D conversion. A GUI was developed for the control of the oscilloscope and the fast export of raw data to binary files. These files are converted to a format that is compatible to the in-house developed evaluation software CRISP (Concise Retrieval of Information from Single Particles) by Klimach (2012) for a-posteriori analysis. In each file the bipolar

mass spectrum, the time of ablation (time stamp), and the particle flight time ("upcounts") between PDU1 and PDU2 is stored.

**2.5   ERICA-AMS: Aerosol mass spectrometry by flash vaporization and electron impact ionization**

During the idle time of the Nd:YAG laser particles remain unablated, even if they are successfully detected by the units PDU1 and PDU2. This actually is by far the largest fraction of the sampled particles emerging from the ADL. If, for example, the

ambient number density of particles with diameters above the detection limit is 100 cm⁻³$_{Std}$, then, at most only 5.4 % (8 shots per second and sampling volumetric flow rate of 1.48 cm³ s⁻¹) of the detectable particles are hit by the laser. Second, particles for which the calculation of the trigger failed continue their travel towards the ERICA-AMS vaporizer. Third, particles that





primarily consist of materials that are transparent at a UV wavelength of 266 nm, such as pure sulfuric acid, are hard to ablate (Murphy, 2007). We selected a UV laser with 266 nm wavelength due to smaller dimensions and the fact, that chemical substances show less fragmentation compared to ablation with shorter wavelengths (Thomson et al., 1997). In general, however, it is also possible to implement excimer lasers operating at shorter wavelength to ablate pure sulfuric acid droplets.

Also, pure sulfuric acid is detected by the ERICA-AMS.

All those aerosol particles which are not ablated in ERICA-LAMS continue their flight towards the ERICA-AMS instrument part, where non-refractory components are flash-vaporized by a tungsten vaporizer (with a surface diameter of 3.8 mm) operating at a temperature of approximately 600 °C. The vapor molecules and fragments become ionized by electrons, with an impact energy of 70 eV, continuously emitted by a filament (emission current of 1.6 mA). This vaporization and ion

generation unit was manufactured by Aerodyne (Aerodyne Research Inc., Billerica, MA, USA). The generated cations are then extracted through an electrostatic lens stack into the C-ToF-MS. At its entrance section, perpendicular to their flight path into the mass spectrometer (see "extractor" and "grid" in Fig. 1) the ions are periodically extracted with a frequency of 50 kHz. This extraction defines the starting time and point for the time-of-flight mass spectrometric ion analysis (Drewnick et al., 2005; Canagaratna et al., 2007). After passing through the C-ToF-MS, the ions reach the MCP (model MCP 40/12/10/8 D 46:1,

Photonis USA Inc., Sturbridge, MA, USA) and generate a signal, which is amplified and finally collected by the data acquisition card (DAQ card; model ADQ1600 USB3, Teledyne Signal Processing Devices Sweden AB, Sweden). The DAQ card serves for both, the generation of periodic trigger pulses for ion extraction, and the acquisition of ion-generated signals from the MCPs. This device samples at 1.6 GS s$^{-1}$ with a high vertical resolution of 14 bits. Multiple consecutive spectra are processed at hardware level over a time period of user-selectable length (typically 400 ms) and finally streamed via a USB 3.0

connection as one averaged raw spectrum to the main control computer.

For quantitative aerosol composition measurements, the background signal, which originates from air molecules and residual vapor molecules inside the chamber, has to be considered and is subtracted from the aerosol sampling signal. For this purpose, in the commercial Aerodyne AMS (Canagaratna et al., 2007) the particle beam is periodically blocked by a chopper inside the low vacuum stage. By means of the chopper it is also possible to distinguish between different vacuum aerodynamic particle

sizes, as the particle flight time duration between passing the (open) chopper and arriving at the vaporizer is size dependent. However, this flight time duration -and the corresponding flight distance between chopper and vaporizer- need to be long enough to achieve such size-resolved sampling. For ERICA-AMS the distance from the shutter to the vaporizer is very short. This would not be the case if we had placed a chopper directly behind the ball joint of the ADL. However, by periodically blocking the particle beam with a chopper at this position, the detection frequency of ERICA-LAMS would have been reduced

accordingly. Thus, we decided to use a simple shutter device instead of the chopper. It consists of a C-shaped profile made of metal and is mounted on the shaft of a high-vacuum magnetically-coupled feed-through (Pfeiffer Vacuum GmbH, Germany). The shaft periodically rotates the C-profile by 90° into and back out of the particle beam axis. In this way, the particle stream to the vaporizer is blocked and permitted, respectively, for adjustable time periods.

Based on experience from flight operation and laboratory experiments, one measurement cycle has a length of 10 seconds

consisting of 25 measured averaged raw spectra. 12 spectra are recorded with shutter position open (4.8 s) and 11 with shutter position closed (4.4 s) for background measurement. Two spectra are recorded during the switching of the shutter with an unclear position and are thus discarded for data evaluation. These open-closed cycles can be set in the acquisition software (TofDAQRec by Tofwerk AG, Switzerland). The collected data are evaluated by the software tool "Tofware" from Tofwerk AG (Fröhlich et al., 2013; Stark et al., 2015; Timonen et al., 2016).



## 2.6 Influence of the ERICA-LAMS on the ERICA-AMS

The assembly of the two instrument parts, i.e., the ERICA-LAMS and the ERICA-AMS, in a serial configuration might lead to interactions. On the one hand, it can safely be assumed that the ERICA-LAMS is largely unaffected by the ERICA-AMS presence and operation. On the other hand, particles which are ablated or distracted in the ERICA-LAMS are excluded from

the total mass measured by the ERICA-AMS.

The first loss mechanism for particles to be analyzed by the ERICA-AMS is the ablation of the particles in the ERICA-LAMS. The impact of this instrument-induced loss depends to the number concentration of particles within the sampled aerosols and cannot be compensated. Two examples illustrate this for different conditions:

i. In pristine conditions like the summertime Arctic boundary layer, particle number concentrations rarely exceed 5 cm$^{-3}$

(Köllner et al., 2017) in the size range relevant for our instrument (see Sect. 3.3.3). For the typical sampling volumetric flow rate ($\Phi_{ERICA}$) of 1.48 cm³ s$^{-1}$, around 7 particles per second would be detected at maximum by the ERICA-LAMS. Even with the ablation laser being restricted to a maximum of 8 shots per second, theoretically this can result in an 100 % loss for the ERICA-AMS, since all particles can potentially get ablated and ionized with an assumed ablation efficiency $AE$ (definition see Sect. 3.4) of 100 %. This is a conservative estimation since some of the detected mass would not have been measured by

the ERICA-AMS due to the particle composition of refractory material. Also, small particles ($d_{va} < 100$ nm, see Sect. 3.3.3) cannot be detected by the detection units and will not lead to any losses at the ERICA-AMS. Furthermore, in practice, the $AE$ is particle size-dependent and, for all particle sizes, lower than unity. Thus, the parameter $AE$ is not applicable to estimate the losses of the non-ablated particles. The value of the $AE$ might not be lower than unity because of the failure of the laser pulse hitting the aimed particle, but because of the ionization efficiency within the ablation process. Thus, at such low ambient

particle concentrations, the quantitative results of the ERICA-AMS measurements must be viewed critically, and possibly measurement strategies like including periods of short inactivity for the ERICA-LAMS can be adopted. Further studies and additional instrumentation (size distributions) need to be considered to quantify the ERICA-AMS results at low particle concentrations.

ii. Usually during our field deployment, around 100 particles s$^{-1}$ being detected by the PDUs during ambient aerosol

measurements in the planetary boundary layer. Considering $\Phi_{ERICA}$, 8 laser shots per second, and an overestimated maximum $AE$ of 100 %, about 5.4 % of the particles are ablated and thus will not reach the vaporizer. For the same reasons as those discussed above, this is a conservative estimate and the actual losses cannot be determined. However, the losses can be neglected considering the commonly assumed uncertainty of 30 % in AMS instruments.

Another loss mechanism is the deflection of charged particles caused by the temporarily applied electrical field between the

high-voltage extraction plates of the ERICA-LAMS. This will lead to losses which are impossible to be compensated for because typically the charge distribution of ambient aerosol particles is not known. Therefore, measures have been taken in order to minimize these losses as much as possible. As described in Sect. 2.4, the high-voltage (HV) for ion extraction is only applied shortly before a particle is ablated. The deflection caused by the electric field is dependent on the particle size and charge; the resulting losses consequently depend on the dimensions and shape of the vaporizer, meaning that not all deflected

charged particles get lost. The HV-switch unit was specially designed to keep the deflection losses to a minimum. The HV is applied for 10 ms per shot, resulting in a duty cycle of 8 %, assuming the laser is shooting 8 times per second. A dedicated measurement of ambient air in Mainz, Germany, with the HV and ablation laser applied shows that both loss mechanisms together induce less than 5 % reduction of the particle mass compared to a reference measurement without HV and ablation laser, which agrees with the estimation above.





## 3 Instrument characterization

### 3.1 Detection and ablation laser beam waists

For characterization of the laser beams of the PDUs and the ablation laser outside the vacuum chamber, a razor blade was moved stepwise perpendicularly into the respective laser beam (with steps of 0.01 mm). The remaining energy was measured

using a bolometer (model High sensitivity thermal sensor 3A, Ophir Optronics Solutions Ltd.) in case of the diode lasers, and by an energy meter (model EnergyMax™-USB, J-25MB-LE, Coherent, Inc., USA) for the pulsed UV ablation laser. The results of the measurements are provided in Sect. S2 in the supplement.

To measure the beam waist radius $w_0$ of the detection laser in two dimensions (x and y), the razor blade was positioned directly at the focal point. Curve-fits of the Gaussian error function (Eq (1)) were applied to all data sets, with $P_0$ for the power offset

of the fitted curve, $P_{max}$ the maximum power, $pos_0$ the central position of the Gaussian distribution, $pos$ the horizontal position of the blade ( i.e., the independent variable), and $w_0$ the beam $1/e^2$-radius of the Gaussian intensity profile (Araújo et al., 2009).

$$P(pos) = P_0 + \frac{P_{max}}{2} \cdot \left(1 - \mathrm{erf}\left(\frac{\sqrt{2}\,(pos - pos_0)}{w_0}\right)\right) \qquad (1)$$

It was found that the laser spot has an oval cross-sectional shape with the dimensions of $w_0 = (30.3 \pm 1.2)$ µm and

$w_0 = (20.0 \pm 0.9)$ µm (measurement in x- and y-direction, respectively). Thus, the $1/e^2$-diameter ($w_{0,dia} = 2w_0$) can be determined for the x-direction as $w_{0,dia} = (60.6 \pm 2.4)$ µm and for the y-direction as $w_{0,dia} = (40.0 \pm 1.8)$ µm. The irradiance can be estimated as $2.1 \cdot 10^3$ W cm$^{-2}$. Since the detection units are identical in construction, this measurement represents both detection units.

The procedure of the characterization of the ablation laser beam is similar to the one adopted for the detection lasers. Here,

however, a cross-sectional scan is performed at eight different positions along the laser beam's optical axis. To evaluate the whole beam waist, the $\frac{1}{e^2}$−radii $w$ were plotted versus the position of the razor blade from the lens $z_{pos}$. To determine the focal length $z_0$, the Rayleigh range $z_R$, and the beam waist radius $w_0$ at the axial position $z_{pos}$, the curve-fit of the Gaussian near field equation (Eq. (2); Siegman (1986)) was applied:

$$w(z_{pos}) = w_0 \cdot \sqrt{1 + \left(\frac{z_{pos} - z_0}{z_R}\right)^2} \qquad (2)$$

From exposures on photosensitive paper, the laser beam profile appeared radially symmetrical, and this measurement was done only in one orientation. The curve-fitting results in a Rayleigh range $z_R$ of 7.5 mm, focal length $z_0$ of 76.4 mm, and a beam waist radius $w_0$ of 125 µm. Thus, the beam waist diameter $w_{0,dia}$ is approximately 250 µm, resulting in an irradiance of $1.36 \cdot 10^9$ W cm$^{-2}$. The ablation laser beam waist radius and energy density are sufficient for the ablation of submicron particles and the measured values are comparable to those of other single particle mass spectrometers, like ALABAMA (Köllner, 2019)

and the A-ATOFMS (Su et al., 2004).

### 3.2 Vacuum aerodynamic diameters derived from particle flight times

For the particle sizing, using particle flight times, a calibration measurement using NIST-certified size standard PSL (polystyrene latex) particles was conducted. In addition, laboratory-generated monodisperse ammonium nitrate (AN) particles,

size-selected by a differential mobility analyzer (DMA), were measured. Details on the experimental setup are provided in Sect. S3 in the supplement. AN is not only the standard reference substance for the AMS calibration (Jayne et al., 2000; Canagaratna et al., 2007), but also one of the key components (Höpfner et al., 2019) during the StratoClim aircraft deployments of ERICA in the Asian Tropopause Aerosol Layer (ATAL; e.g., Vernier et al. (2011)).





The particle time-of-flight is dependent on the aerodynamic diameter in the free molecular regime, the so called "vacuum aerodynamic diameter" $d_{va}$ (definition see Sect. S3 in the supplement; DeCarlo et al. (2004)). Unless otherwise specified, $d_{va}$ is used for particle sizes within this publication. To determine the particle flight time, the time between the light scattering signals at PDU1 and PDU2 is measured by the TC in units of clock cycle counts (denoted by the variable "upcounts", $upc$),

where one cycle equals 40 ns. For the calibration measurement with PSL particles, 15 different PSL size standards in the range from 80 nm to 5145 nm were used (see Sect. S3 in the supplement). Considering $upc$ and the clock cycle time of the trigger card, the particle time-of-flight $t_{ptof}$ can be determined for each particle size. For the evaluation of the calibration measurement, $d_{va}$ is plotted versus $t_{ptof}$ (Fig. 3a). To determine a calibration curve, various functions are described in the literature (e.g., Allan et al. (2003), Wang and McMurry (2006), and Klimach (2012)). For our instrument, a polynomial fit of

second order, as described by Brands et al. (2011), was found to be the most suitable. The deviation of the NIST particle size standard from the calibration curve $DVI_{rel}$, i.e., the accuracy, is shown in Fig. 3b. $DVI_{rel}$ was calculated according to Eq. (3), where $d_{va,fit}$ is the $d_{va}$ value on the calibration curve and $d_{va,particle}$ is the $d_{va}$ value of the particle measurement for the same $t_{ptof}$ value.

$$DVI_{rel} = \frac{d_{va,fit}(t_{ptof}) - d_{va,particle}(t_{ptof})}{d_{va,particle}(t_{ptof})} \tag{3}$$

For PSL particles, the deviation from the calibration curve is lower than 5 % except for the deviating measurements with 158 nm and 421 nm particles. To compare the PSL calibration curve with measurements of AN particles, the described procedure determining flight times of PSL particles by histograms was also applied to AN particles in the size range of 138 nm to 814 nm (red markers in Fig. 3, see Table S3 in the supplement). Apparently, the PSL particle time-of-flight calibration can be applied to AN particles (Fig. 3a). The relative deviation from the PSL calibration curve $DVI_{rel}$ (Fig. 3b) was calculated

according to Eq. (3) and is less than 10 % for AN particles with sizes between 213 nm and 548 nm. Although the particle time-of-flight calibration was conducted with PSL particles, the calibration is also valid, over the total $d_{va}$ size range, for pure AN particles, since the deviation of AN particles is in the same range as the deviation of PSL particles.

### 3.3 Characterization of the particle beam and the particle detection

#### 3.3.1 Methodology to determine the optical particle detection efficiency and the particle mass detection efficiency

Knowing the particle beam properties at the PDUs, the ablation laser area, and the vaporizer is essential for interpreting and evaluating measured data. For proper detection of the sampled particles, a sufficient overlap of the particle beam with the laser beams and the vaporizer is required. The optical particle detection efficiency of the PDUs was determined by comparison of count rates of the individual detection units (PDU1 and PDU2) with those of either a condensation particle counter (CPC) or

an optical particle counter (OPC) as reference device (see Sect. S3 in the supplement). In this way, the particle numbers or, indirectly, the mass concentrations measured by the ERICA can be associated with the number concentration of the sample air flow. The measured PSL particle sizes and the respective measurement setups are shown in Sect. S3 in the supplement.

To determine the size- and ADL position dependent optical detection efficiency $DE_{PDU}$ at the detection units with PSL particles (see Table S4 in the supplement), the ADL was tilted in steps and $DE_{PDU}$ was measured at different ADL positions

$x_{pos}$, while the position of the detection laser was kept constant. Hereafter, this procedure is referred to as "ADL position scan". $DE_{PDU}$ was determined for each lens position $x_{pos}$ according to Eq. (4).

$$DE_{PDU}(x_{pos}) = \frac{\overline{cts}_{Det}(x_{pos})}{\bar{c}_{ref} \cdot \Phi_{ERICA}} \tag{4}$$

Here, $\overline{cts}_{Det}$ is the averaged value of the number of particles per second counted by each PDU over 30 seconds, $\Phi_{ERICA}$ is the volume flow into the ERICA and $\bar{c}_{ref}$ is the value of the number of particles per volume unit averaged over 30 seconds at the





reference device. Fig. 4 shows a typical result of an ADL position scan for PSL particles, here with particles of a size of 834 nm, at PDU1 and PDU2. The curve fit to the ADL position scan can be described as a convolution integral of a rectangular top-hat function of the effective detection laser width $2r_{eff,L}$, since the scattered light is only detected above a certain intensity threshold, and a 2-D Gaussian distribution function representing the particle beam cross section. The effective laser beam

radius $r_{eff,L}$ is the laser beam radius wherein a particle is registered. For more details on this method see Molleker et al. (2020). The convolution is described by Eq. (5) according to Molleker et al. (2020):

$$DE_{PSL}\left(x_{pos}\right) = \frac{1}{2} \cdot \left(erf\left(\frac{x_{pos}+r_{eff,L}-x_0}{\sqrt{2}\sigma}\right) - erf\left(\frac{x_{pos}-r_{eff,L}-x_0}{\sqrt{2}\sigma}\right)\right) \cdot A_{scan} \qquad (5)$$

The variable $\sigma$ is a measure for the particle beam width, i.e., the particle beam radius, and $x_0$ corresponds to the value of $x_{pos}$ at the peak value. This $x_0$ value is also called the modal value of the ADL position scan. The parameter $A_{scan}$ is a scaling

parameter of the peak value of the ADL position scan and accounts for losses e.g., ADL transmission efficiency values smaller unity. Equation (5) is used as curve-fit function for determining the values of the parameters $r_{eff,L}$, $x_0$, $\sigma$, and $A_{scan}$. A plateau such as the one shown in Fig. 4a indicates a narrow particle beam with respect to the effective laser width for the respective measurement.

For the measurements of particles with sizes from 218 nm to 834 nm, it was assumed that the particle losses between PDU1

and PDU2 are negligible. Therefore, the curve-fitting for both detection units was performed simultaneously for each particle size with both data sets (PDU1 and PDU2) by a comprehensive analysis, which allows to combine two data sets in one common, single curve-fitting procedure. In the following, this procedure is referred to as combined curve-fitting. During this combined curve-fitting procedure, the variable $A_{scan}$ was linked for both PDUs by determining one $A_{scan}$ value for PDU1 and PDU2 simultaneously. Thus, only one value for $A_{scan}$ per measured particle size was

obtained (see Fig. 4).

For the evaluation of the measurement with PSL particles of 108 nm in size, a different approach was chosen because losses between PDU1 and PDU2 seemed reasonable due to the particle beam divergence (Huffman et al., 2005). Therefore, the evaluation was carried out without the combined curve-fitting procedure and thus, individually for the measurements at PDU1 and PDU2. Due to the mathematical relation between the variables $r_{eff,L}$ and $A_{scan}$ during curve-fitting, it was not possible to

determine both variables at the same time. Therefore, $r_{eff,L}$ was calculated separately and kept constant during the curve-fitting. Considering the size-dependence of the scattered light intensity based on Mie scattering, $r_{eff,L,108nm}$ was estimated for the measurement with PSL particles of a size of 108 nm adopting suitable software routines following Bohren and Huffman (1998). The value of $r_{eff,L,218nm}$, determined for the measurements of particles with sizes of 218 nm, was used as base for the estimation. The result of the calculations showed, that a particle of 108 nm scatters the same amount of light as a particle of

218 nm, when it is closer to the focus by a factor of 0.955. Thus, $r_{eff,L,108nm} = 0.955 \cdot r_{eff,L,218nm}$ was used as curve-fit constant for the evaluation of the measurement with PSL particles of 108 nm (see Sect. S4.1.1 in the supplement). Since this calculation is based on a Gaussian laser beam profile, it can only be seen as an approximation, since especially the outer parts of the laser beam might deviate from a Gaussian profile due to diffraction and reflection in the laser beam setup.

In addition to the particle detection efficiency for PSL particles, the optical particle detection efficiencies of particle counting at both PDUs were determined according to Eq. (4) for AN particles between 91 nm and 814 nm in size (see Sect. S3 in the supplement). Besides the singly charged, the doubly charged particles have to be considered when using a DMA for size selection out of a polydisperse aerosol. For this, a newly developed, iterative method was adopted and is described in detail in Sect. S4.2 in the supplement. Briefly, the curve-fit function of Eq. (4) was extended by a second term for the doubly charged

particles and two weighing factors to account for the fractions of the particle charges. As for the measurements with PSL particles, the parameters $r_{eff,L}$, $\sigma$, $x_0$, and $A_{scan}$ could be determined by a combined curve-fitting procedure (exceptions see Sect. 4.2 in the supplement).





Simultaneously to the measurements with AN particles at the detection units PDU1 and PDU2 of the ERICA-LAMS, the mean mass concentration of AN was determined with the ERICA-AMS. Similar to Liu et al. (2007), the detection efficiency of the ERICA-AMS, based on particle mass, was measured. As a reference, we used the CPC to obtain the mean particle number concentration and calculated the input mass concentration. The afterwards applied curve-fitting evaluation method also

accounts for the doubly charged particle fraction and is described in detail in Sect. S4.2 in the supplement. By the curve-fitting procedure, the parameters $r_{eff,V}$ (effective vaporizer radius), $\sigma$, $x_0$, and $A_{scan}$ could be determined (see Sect. S4.2 in the supplement for definitions and exceptions). All these parameters, $r_{eff,L}$, $r_{eff,V}$, $\sigma$, $x_0$, and $A_{scan}$, are essential for adjustment procedures of the instrument and to interpret the obtained laboratory and field mass spectra. Furthermore, the determined parameters are used in Sects. 3.3.2 and 3.3.3 to characterize the particle beam. Overall, they serve as a means for the evaluation

of the performance of the instrument.

### 3.3.2    Results of the particle beam characterization

The parameters $r_{eff,L}$, $r_{eff,V}$, $\sigma$, $x_0$, and $A_{scan}$ were determined by the curve-fitting functions (Eq. (5) and Eqs. (S14) and (S16) in the supplement) and are thus in the dimension relative to the ADL position $x_{pos}$ as read out on the micrometer adjustment screw (see Sect. S1.2 in the supplement). Below, the parameters were rescaled, using the intercept theorem, to the

dimension of the particle beam at the specific position (PDU1, PDU2, ablation spot, and ERICA-AMS vaporizer).

The curve-fittings yield the standard deviation $\sigma$, which is proportional to the particle beam $\frac{1}{\sqrt{e}}$-radius at each detector (PDU or vaporizer). The particle beam diameter $w_{part}$ is defined as $2\sigma$, i.e., the $\frac{1}{\sqrt{e}}$-diameter of the Gaussian distribution function. In Fig. 5, $w_{part}$ is displayed as function of the particle size $d_{va}$ at various locations within the instrument. The particle beam diameter $w_{part}$ is approximately 0.1 mm at PDU1, and 0.2 mm at PDU2 for particle sizes above 400 nm. For PSL particles of

108 nm in size, the $w_{part}$ values are 5 times (7 times) wider at PDU1 (PDU2). The measurements with the OPC for larger diameters indicate a trend for $w_{part}$ from 0.10 mm to 0.18 mm. For AN particles of 335 nm in size, a minimum of $w_{part}$ was found, as the corresponding values for $w_{part}$ at PDU1 and PDU2 are 0.04 mm and 0.03 mm, respectively. At the vaporizer, the largest value for $w_{part}$ of 2.2 mm was measured for AN particles of 91 nm in size, which is narrower than the width of the vaporizers' physical cross-sectional diameter of 3.8 mm. Thus, by adjusting the ADL properly, all investigated AN particles

larger than 91 nm can be collected by the vaporizer. The overall curve shapes at each PDU describe a "V", where the smaller and the larger particles show a larger $w_{part}$ than particles of 335 nm in size. Smaller particles can be deflected by collisions with residual gas molecules and larger particles are over-focused by the ADL due to their inertia (Zhang et al., 2002; Peck et al., 2016). Considering the geometry of the instrument, also $w_{part}$ at the ablation spot and at the ERICA-AMS vaporizer can be extrapolated from the respective $w_{part}$ for AN at PDU2. The longer travel distance for the particles and the particle beam

divergence (Huffman et al., 2005) results in a 3.3-fold broader $w_{part}$ for AN particles at the vaporizer than at PDU2. The calculation yields a maximum $w_{part}$ of 0.48 mm at the ablation spot, a value which is approximately two times the ablation laser beam diameter $w_{0,dia}$ (see overlap parameter determination below in this section), and $w_{part}$ of 1.07 mm at the vaporizer (both for AN particles of 548 nm in size).

In the next step, we focus on the overlap of the particle beam with the detection laser focus. Considering an optical laser beam

diameter $w_{0,dia}$ of 60 μm of the PDUs (see Sect. 3.1), the particle beam diameter $w_{part}$ is a factor 2 to 3 wider (PSL, $d_{va} > 400$ nm). However, the laser intensity of a Gaussian beam provides intensities larger than zero also for radial distances above $w_0$ and the scattered light might be sufficient for particles to be detected. The maximum distance from the laser axis where particles can be detected is represented by the parameter $r_{eff,L}$ and not $w_0$. Fig. 6 shows the effective laser beam radius $r_{eff,L}$ and $r_{eff,V}$ as a function of the particle size $d_{va}$. Overall, for PSL particles, $r_{eff,L}$ is between 0.1 mm and 0.4 mm. The



shape of the curve of the effective laser beam radius depends on the response function of the scattered light intensity as a function of size, where an increase to larger sizes was expected. For the measurements with PSL particles of 108 nm and AN particles of 91 nm and 138 nm in size, this is inevitable, since the values of $r_{eff}$ are calculated based on the Mie scattering (see Sect. S4.1 in the supplement). For larger particles, or the measurements with the OPC as reference device, an increase of $r_{eff,L}$

with particle size would be expected. Due to the fact that the OPC measurements were performed with various PMT threshold values (see Sect. S3 in the supplement), $r_{eff,L}$ appears lower than the CPC reference measurements and thus, $r_{eff,L}$ for particle sizes above 834 nm is underestimated in Fig. 6. The AN measurement results do not agree with the results of the measurements with PSL particles, possibly due to a non-spherical shape and a different refractive index of AN as compared to that of PSL. The vaporizer width determined by the ADL position scans, i.e., $r_{eff,V}$, agrees with the vaporizer's physical dimension of 1.9

mm radius.

To determine the overlap of the particle beam with the detection laser beam, the particle beam diameter $w_{part}$ is compared to the effective laser diameter $d_{eff,L} = 2r_{eff,L}$. Therefore, the overlap parameter $S_{detect,L} = {w_{part}}/{d_{eff,L}}$ was calculated for different particle sizes at the PDUs as the maximum possible overlap of $w_{part}$ and $d_{eff,L}$ for each measurement at lens position

$x_{pos} = x_0$. The parameter $S_{detect,V} = {w_{part}}/{d_{eff,V}}$ (with $d_{eff,V} = 2r_{eff,V}$) expresses the overlap of the particle beam with the effective vaporizer width. Both are shown in Fig. 7. The gray horizontal line marks an overlap parameter of 1. All investigated particle sizes below that line are detected sufficiently well within $1\sigma$ of the particle beam width. That is the case, within their uncertainties, for all measurements except for PSL particles of 108 nm in size. The reason for this is a large $w_{part}$ for the smallest particles resulting from a large particle divergence caused by the small particle inertia for this size (Zhang et

al., 2002). The values of $S_{detect,L}$ of the measurements with the OPC are overestimated, since the resulting values of $r_{eff,L}$ are underestimated, due to the varying threshold during the measurements (see Sect. S3 in the supplement). However, the values are below a ratio of 1. It has to be remarked that a value above 1 does not indicate an impossible particle detection by the PDUs, but just a reduced detection efficiency. As shown in Sect. 3.2 the PDUs can detect particles in a size range between 80 nm and 5145 nm.

An overlap parameter $S_{ablation}$ can also be determined for the overlap of the particle beam and the ablation laser spot by dividing the particle beam diameter $w_{part}$, exemplarily for AN particles, at the ablation laser spot (see brown curve in Fig. 5) by the determined optical laser beam waist $w_{0,dia}$ of 250 µm (see Sect. 3.1; $S_{ablation} = {w_{part}}/{w_{0,dia}}$). In Fig. 8, $S_{ablation}$ is plotted versus the particle size $d_{va}$. The calculated fraction of the illuminated area of the UV ablation laser spot is between

0.23 (at $d_{va}$ = 335 nm) and up to 1.91 (at $d_{va}$ = 548 nm). Although the particle beam is larger than the ablation laser beam waist diameter for most particle sizes, it is possible to ablate particles and measure them with the mass spectrometer. This indicates again, that $w_{0,dia}$ is not the most meaningful measure for the overlap. It also leads to the conclusion that particles can experience largely different laser intensities depending on the position of the particle within the ablation laser beam. However, $S_{ablation}$ smaller than 1 indicates that $1\sigma$ of the particle beam is within the $w_{0,dia}$ of the ablation laser spot. Nevertheless, field

measurements with ambient aerosol show that also particles of sizes between 80 nm and 5145 nm can be ablated and detected by the MCPs (see Sects. 3.4 and 4).

All the data shown for the parameters $S_{detect,L}$, $S_{detect,V}$, and $S_{ablation}$ are the maximum possible values of the respective particle sizes obtained when performing the ADL adjustment separately for each particle size.



### 3.3.3    Results of the optical particle detection efficiency and the particle mass detection efficiency

We determined the optical detection efficiencies for PSL and AN particles at PDU1 and PDU2, and the particle mass detection efficiency for AN particles at the ERICA-AMS vaporizer for two cases: largest possible, i.e., the maximum, detection efficiency $DE_{max}$ and the detection efficiency for the set ADL position ($x_{pos}$ = 10.55 mm) during the deployment in

Kathmandu, Nepal (KTM), $DE_{KTM}$. Both, $DE_{max}$ and $DE_{KTM}$, combine the optical detection efficiency measurements with PSL and AN particles described in Sect. 3.3.1. Section S4.5 in the supplement provides a listing of all relevant equations.

The parameter $DE_{max}$ was determined for each measurement. For this, the determined set of parameters ($r_{eff,L}$, $r_{eff,V}$, $\sigma$, $x_0$, and $A_{scan}$) of each curve-fitting, was re-inserted in the respective Eqs. (5), (S14), or (S16). For the maximum possible detection efficiency $DE_{max}$, the variable $x_{pos}$ equals the modal value of the ADL position scan $x_0$, thereby compensating for the size-

dependent particle beam shift (see Sect. S4.6 in the supplement). To obtain the $DE_{max}$ values in practice, the ADL has to be readjusted for each particle size.

Fig. 9 presents the largest possible, i.e., the maximum, detection efficiency $DE_{max}$ at ADL position $x_0$ as a function of the particle size $d_{va}$. The values of $DE_{max}$ for PSL particles with particle sizes larger than 200 nm is above 0.60, reaching the value of 1 for particle sizes of 834 nm at PDU1. The parameter $d_{50}$ is typically used to characterize the detection limits of

single particle counting devices. The parameter $d_{50}$ is defined as 50 % of the maximum $DE_{max}$ value, as it is for $A_{scan}$, discussed in Sect. 3.3.2. Here, the low $d_{50}$ value of the optical particle detection is between the particle sizes 108 nm and 218 nm. The upper $d_{50}$ value lies slightly above a particle size of 3150 nm. Interpolations or extrapolations for the measurements with PSL particles are used to estimate the $d_{50}$ values. We found 180 nm as the lower and 3170 nm as the upper $d_{50}$ value. At PDU2, the $DE_{max}$ is lower, which can be explained by the broader particle beam at PDU2 compared to PDU1.

The curve progression of the particle measurements up to particle sizes of 1000 nm follows the expected response function of the light scattering, especially the decreasing $DE_{max}$ at small particle sizes. The decreasing $DE_{max}$ values for large particles and be explained by the reduced transmission of the ADL due to particles losses by inertial impaction.

The $DE_{max}$ values found for the measurements at the ERICA-AMS vaporizer are not comparable in absolute terms with the $DE_{max}$ values found for the AN measurements at PDU1 and PDU2, since the measurements at the position of the ERICA-

AMS vaporizer are analogous to an IE calibration measurement (Drewnick et al., 2005). During this IE calibration, among other losses, the transmission losses in the ADL are compensated. However, this measurement demonstrates that the decreasing $DE_{max}$ for smaller sizes at the PDUs are not caused by losses in the ADL, but the inability to detect small particles by adopted optical means. No $d_{50}$ value could be determined for the measurements on the vaporizer. Even though the data point at 91 nm indicates a lower $d_{50}$ cut-off, we assume that the particle size range in which the ERICA-AMS can measure is between

~120 nm and 3.5 µm, as specified by Xu et al. (2017) for the ADL type used here.

Due to the size-dependent particle beam shift, and thus the $DE_{max}$ for various particle sizes is found at various lens settings, a compromise for all particle sizes has to be found to adjust the ADL. To choose the optimum ADL position, AN particles with various sizes were measured with the ERICA-AMS at different ADL positions. The position that yields the highest mass concentration signal as compromise for all sizes is defined as the best ADL position. We found $x_{pos}$ = 10.55 mm as the

optimum ADL position, which was subsequently applied during the field deployment in Kathmandu, Nepal (KTM). Fig. 10 shows the optical detection efficiency during field deployment in KTM $DE_{KTM}$ as a function of the particle size $d_{va}$ at this specific ADL position. The calculations of the parameter $DE_{KTM}$ are based on Eq. (5), (S14), or (S16) and are shown in Sect. S4.5 in the supplement. Here, besides $x_{pos}$ = 10.55 mm, all other parameter values of the singly charged fraction were adopted from the curve-fitting results of the individual measurements. In Fig. 10a, the detection efficiency $DE_{KTM}$ of PSL particles is

plotted as a function of the particle size $d_{va}$. The graph shows an increase with particle size until a maximum for $DE_{KTM}$ of 0.74 for a particle size of 410 nm. By interpolation, the lower $d_{50}$ values are 190 nm at PDU1 and 160 nm at PDU2. As upper $d_{50}$ values we found 745 nm at PDU1 and 750 nm at PDU2. Furthermore, $d_{50}$ is pronounced differently for particles with





optical properties other than PSL such as AN. Except for the measurement with particle sizes of 213 nm at PDU1, all AN particle measurements (Fig. 10b) result in a $DE_{KTM}$ larger than 0.40 and reach their maximum here for particle sizes of 335 nm (PDU2) and 548 nm (PDU1), both having values around 0.86. Here, $d_{50}$ solely can be determined for the measurement with AN particles at PDU1 to 270 nm. For the measurements at the vaporizer, no $d_{50}$ values can be determined, because the results

are above 50 % of their maximum $DE_{KTM}$ values over the entire size range. The $DE_{KTM}$ at the vaporizer is 1 due to the normalization by the IE calibration, as explained above (see Sect. 3.6.2).

The measurements demonstrated in this section have shown that detection efficiency varies with particle size and type. The efficiency of the optical detection strongly depends on the adjustment of the instrument as well as the optical and the aerodynamic properties of the particle. The AMS part instead shows a fairly stable efficiency around 1 for the examined size

range after calibration with AN particles of 483 nm in size. This is highly desirable to ensure the quantitative measurement of the AMS.

### 3.4 Ablation efficiency

Another relevant parameter to describe the performance of a single particle laser ablation mass spectrometer is the ablation

efficiency $AE$. The definition of $AE$ (see Eq. (6)), also called hit rate, is the number of acquired spectra $N_{spectra}$, i.e., particles successfully ionized by the ablation laser and recorded by the oscilloscope, divided by the number of laser shots $N_{shots}$, i.e., attempts to ablate particles (Su et al., 2004),

$$AE = \frac{N_{spectra}}{N_{shots}} \tag{6}$$

This definition is largely independent from ambient particle number concentration and the idle time of the laser, but rather

reflects the adjustment of the instrument. For each particle for which a laser shot is triggered, the aerodynamic particle size is determined by the TC. With ERICA-LAMS, $AE$ values of up to 1 (not shown) could be achieved in the laboratory for PSL particles of a certain size after optimizing the PMT thresholds and the pulse generator multiplier value for the corresponding particle size. To assess on the smallest detectable particle size, the detection units PDU1 and PDU2 were optimized for the following experiment for PSL particles of 218 nm size.

To determine the ablation efficiency for ambient aerosol, ambient air from outside the laboratory was sampled. Only spectra of particles with diameters in the range of calibration (see Sect. 3.2) were considered. The ablation laser was adjusted to maximum ablation efficiency for ambient aerosol, by varying the pulse generator multiplier (see Sect. 2.4) and adjusting the dichroitic mirror DM1 (Fig. 2). The average ablation laser pulse energy was 3.2 mJ. Fig. 10 shows the ablation efficiency $AE$ of the described experiment as a function of the particle size $d_{va}$. Furthermore, $N_{spectra}$ and $N_{shots}$ are plotted as a function

of particle size. In the size range from 100 nm to 1000 nm, $AE$ values of more than 10 % are achieved. At the particle sizes between 200 nm and 300 nm, at approximately 230 nm, a maximum of 0.52 was found. The reason for the maximum at this particular particle size might be the selected optimization in the adjustment of the detection and ablation units. Particles get detected by the PDU as soon as their scattered light is sufficiently intense. This might be earlier for larger particles due to the higher $r_{eff,L}$ and thus the timing might not be optimal for all particle sizes. In addition, a large particle beam divergence (see

Sect. S4.6 in the supplement) can lead to a low $AE$ for small particles ($d_{va} < 200$nm) as well as for large ones ($d_{va} > 400$ nm). This curve progression reflects the experimentally determined particle beam width $w_{part}$ and the overlap parameter $S_{ablation}$, (see Fig. 8 in Sect. 3.3.2). Furthermore, $AE$ is less than unity over all sizes, which may be due to the ionization efficiency of particle components in the ablation process. Beside the particle size, $AE$ also depends on the particle shape and the chemical composition of the particle (Su et al., 2004) as well as on the laser intensity of the ablation laser (Brands et al., 2011).



### 3.5 Single particle mass spectra measured by the ERICA-LAMS

#### 3.5.1 Exemplary single particle mass spectra from laboratory tests

To study mass spectra of different chemical compounds, solutions of sodium chloride (NaCl), ammonium nitrate (AN; $NH_4NO_3$), benz[a]anthracene (BaA; $C_{18}H_{12}$), and a gold sphere suspension were nebulized. Details on the experimental setup, as well as on the properties of the studied particles are provided in Sect. S3 in the supplement. If not mentioned separately, all mass spectra were processed by the evaluation software CRISP (Klimach, 2012). During this processing, the mass-to-charge ratio ($m/z$) of all spectra is calibrated and each peak area is integrated over 25 signal acquisition samples before and after the determined $m/z$ peak center. In the resulting so-called stick spectra, a stick reflects the ion peak area in units of mV·sample of the specific $m/z$. To determine the ion peak area threshold of the ERICA-LAMS, i.e., minimum peak that can be detected, the data set of the first field campaign (see Sect. 4) was used. The ion peak area threshold is defined as the ion peak area at $m/z$, which are usually unoccupied ($m/z$ 2 to $m/z$ 6 for cations, $m/z$ 2 to $m/z$ 11 for anions), below which 99% of the baseline noise is present (Köllner et al., 2017). The result for cations and anions is an ion peak area threshold value of 7 mV·sample.

As an example, Fig. 12a presents a bipolar ion mass spectrum of a single sodium chloride particle as detected by ERICA-LAMS during laboratory measurements. Other pure substance spectra are shown in Fig. 12b for a single AN particle. The spectral patterns detected by the ERICA-LAMS are comparable and in good agreement with results produced by other established single particle mass spectrometers, e.g., ALABAMA (Brands et al., 2011; Köllner et al., 2017), ATOFMS (Gard et al., 1997; Gross et al., 2000; Liu et al., 2000), and a modified LAAPTOF (Ramisetty et al., 2018). Also for ambient stratospheric particles, Schneider et al. (2021) have shown that spectra from ERICA-LAMS and ALABAMA are comparable.

It is noteworthy that an important prerequisite for the later application of ERICA during airborne measurements was the capability to detect the presence of gold particles in the sampled aerosols. Gold can be used as a marker for self-contamination. By plating the sampling inlet with gold, it can safely be assumed that if gold-containing particles are found, it indicates that they have removed material from the inlet (Dragoneas et al., 2021). To test the instrument's capability of measuring gold particles, dispersions of gold spheres ($d_{va}$ = 3860 nm) were used. A typical bipolar spectrum is displayed in Fig. 12c. In addition to the signal on $m/z$ 197 from the Au$^+$-cation, the peak of Au$_2^+$-cation on $m/z$ 394 was consistently present, providing a good indication that actual gold particles were detected, even in the absence of an isotopic pattern or specific anion signal. The Na$^+$-, K$^+$-, and Ca$^+$-signals in the spectra can be attributed to the residual buffer solution of the gold particle dispersion. The identification of particle types for which the evidence is based on hardly ionizable substances, such as gold, is only possible, if the content of well ionizable substances is moderate (Reilly et al., 2000), since otherwise no Au signal might be obtained.

We further investigated BaA particles, as BaA has been identified as a component of soot (Lima et al., 2005). A characteristic example of their mass spectra is shown in Fig. 12d. Therein, the C$_n$ and the C$_n$H$_m$ pattern is clearly visible in both the cation and the anion spectra being indicative of polycyclic aromatic hydrocarbons (PAH; e.g., Hinz et al. (1999)). Also, the molecular peak at $m/z$ 228 appears in the spectrum. This observation is consistent with the typical performance of mass spectrometers employing lasers with a wavelength of 266 nm, which result in less fragmentation as compared to those with a wavelength of 193 nm (Thomson et al., 1997). The four examples shown here demonstrate that the ERICA-LAMS provides valid single particle mass spectra that are comparable to those of other instruments in the literature.

#### 3.5.2 Mass spectral resolution of ERICA-LAMS

The mass spectral resolution $R_{MS}$ is a measure for the mass separation performance of the mass spectrometer and is defined as $R_{MS} = \frac{M}{\Delta M}$. The parameter $\Delta M$ is defined as the FWHM of $M$, i.e., the $m/z$ value. Thus, a higher value of $R_{MS}$ indicates a better separation of the $\frac{m}{z}$ peaks in the mass spectra. Appropriate separation is particularly necessary for the identification of

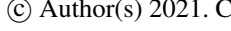



neighboring nominal masses like $m/z$ 39 and $m/z$ 40 (for K$^+$ and Ca$^+$) as well as for signals caused by isotopes, e.g., elements such as tin and lead. In Fig. 13, details of two different raw cation spectra from two ambient aerosol particles are presented. Here, the output voltage signal of the digitizer is displayed as a function of the digitizer sample number (1.6 ns per sample). The particles of the presented spectra were recorded during the StratoClim campaign (July and August 2017) at ground level at the airport of Kathmandu, Nepal. The signal intensities correspond to the isotopic abundance of tin (Fig. 13a) and lead (Fig. 13b). The occurrence of both species can be expected in a polluted environment as in Kathmandu, Nepal. Out of these mass spectra, $R_{MS}$ of the ERICA-LAMS can be estimated to 200 for cations at $m/z$ 120 (Fig. 13a) and 700 at $m/z$ 200 (Fig. 13b). For anion spectra we found a $R_{MS}$ of about 600 at both, $m/z$ 100 and $m/z$ 200. The $R_{MS}$ values of other single particle mass spectrometers are comparable to the here presented ones. Brands (2009) states for the ALABAMA a resolution of 200 for cations of $m/z$ 108 and of 600 for anions of $m/z$ 120. The resolution of the A-ATOFMS (at $m/z$ 100) is for cations 500 and for anions 800 (Pratt et al., 2009). Without any specific $m/z$ value, Gemayel et al. (2016) state for the LAAPTOF a $R_{MS}$ of above 600 for both polarities.

### 3.6 ERICA-AMS performance

#### 3.6.1 Mass spectral resolution of the ERICA-AMS and data preparation

The ERICA-AMS mainly adopts elements of the commercial AMS from Aerodyne (see Sect. 2.1). The observed mass resolution of 800 at $m/z$ 200 during ambient aerosol sampling (see Sect. S5 in the supplement) is comparable with that of commercial C-ToF-MS instruments (Drewnick et al., 2005). The conversion of the ion flight time to a $m/z$ is done using predefined calibration peaks. We use the peaks for CH$^+$, O$_2^+$, SO$_2^+$, $^{182}$W$^+$, $^{184}$W$^+$, and $^{186}$W$^+$, species for which the exact $m/z$ ratio is known and which occur in every spectrum, due to their existence in the vacuum background or outgassing of the heated tungsten filament. The wide range of covered $m/z$ values allows to fit a 3-parameter time-of-flight to $m/z$ relation, which is then valid for the whole spectrum. We decided not to use the common Ar$^+$ peak, because in measurements shortly after evacuating the chamber, the residual organic peak at the same nominal mass of $m/z$ 40 can disturb the determination of the peak center. The software integrates the signal at each particular $m/z$ ratio to generate a stick spectrum. The signal occurring between the $m/z$ peaks is used to estimate a baseline, which is subtracted during this integration. Stick spectra are generated for measurements with open and closed shutter (see Fig. 14a) to subtract the instrument background signal from the aerosol measurement signal, in order to obtain the aerosol contribution only (see Fig. 14b). The difference between the total and the background signal results in the aerosol signal. The open-closed cycle is set to 10 seconds (see Sect. 2.5). A so-called fragmentation table is used to attribute the individual $m/z$-peaks to certain species (e.g., air, organic, nitrate, sulfate, ammonium, and chloride; Allan et al. (2004)). The fragmentation table can be manually adapted to compensate for instrument specific deviations. Along with the particles, a small fraction of the gaseous components are measured, which still exhibit the most dominant peaks for N$_2$, O$_2$, and Ar in the mass spectrum (see Fig. 14b). A more detailed description on the evaluation procedure can be found in e.g., Allan et al. (2004) and Fröhlich et al. (2013).

#### 3.6.2 ERICA-AMS ionization efficiency

By means of a calibration with a test aerosol of AN, the IE can be determined and the peak areas obtained from integration can be converted into a quantitative measure of the aerosol mass concentration of the atmosphere. In order to determine the IE of the ERICA-AMS, in a first step the average signal of a single ion must be measured. This is done by considering single mass spectrum extractions. The assumption is that a rarely occupied $m/z$ signal has a very low probability to experience the arrival of two ions in the same extraction. The peak area of these $m/z$ signals, averaged over multiple events where the signal is above the noise threshold, then represents the average single ion signal (SIS). The SIS is given in units of mV·ns and depends

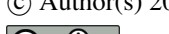



on multiple factors; mostly the type and condition of the MCP detector, the applied high-voltages and the resulting field strengths, the temperature, and the gain of the signal amplifier. After voltage adjustment of the MCP we obtain a SIS of around 0.8 mV·ns.

The IE is determined with AN particles applying Setup B as described in Sect. S3 in the supplement (Fig. S7). The so created

mono-disperse aerosol is sampled by the instrument as well as by a CPC for reference. This mass-based approach is similar to the one described in Drewnick et al. (2005). This method considers the transmission efficiency through the ADL and the possible losses due to particle beam divergence. As a reference zero, a measurement through a filter is performed. The IE calibration factor in "Tofware" is then adjusted so that the nitrate signal equals the nitrate mass load determined by the CPC. To calculate the mass load from the CPC data, several corrections have to be applied. For instance, doubly charged particles

of a larger size are also transmitted through the DMA due to the same electrical mobility, which will also contribute to the mass load. To reduce this effect, we choose a rather large particle size of 483 nm for the calibrations, so that the corresponding larger sized particles of 814 nm are not generated by the nebulizer in a high quantity. By measuring the concentration of singly charged 814 nm and calculating the charge ratio generated by the neutralizer according to Tigges et al. (2015), we correct for the effect of doubly charged 814 nm particles (see Sect. S4.3 in the supplement). In addition the Jayne shape factor has to be

applied (Jayne et al., 2000). The IE is usually given for nitrate and is strongly dependent on the flux of electrons for ionization. The ERICA achieves an IE of 2000 ions per pg, or $2.05 \cdot 10^{-7}$ ions per molecule. This is lower than reported for comparable instruments (e.g., Canagaratna et al. (2007)), partly  due to operation at a lower filament emission current of 1.6 mA. Other test aerosol species can be used to determine a species dependent relative ionization efficiency (RIE). The RIE of ammonium $RIE_{NH4}$ and the RIE for sulfate $RIE_{SO4}$ were determined by independent measurements of AN particles and ammonium sulfate

particles according to Canagaratna et al. (2007). We calculated an averaged $RIE_{NH4}$ to 4.4 and $RIE_{SO4}$ to 0.97. The default RIE values of the organic compounds ($RIE_{org} = 1.4$), for chloride ($RIE_{Chl} = 1.3$ and for nitrate ($RIE_{NO3} = 1.1$) were adopted from Canagaratna et al. (2007).

With the IE and RIE values the ion count signal can be converted into an aerosol mass. Together with the known flow into the instrument ($\Phi_{ERICA} = 1.48$ cm³ s⁻¹) the mass concentration of the particulate matter is calculated. Due to the installed constant

pressure inlet (Molleker et al., 2020), which keeps the pressure in the ADL constant, the volumetric flow into the instrument increases with decreasing ambient pressure. With the assumption of a stable instrument temperature, this leads to a constant mass flow or normal flow (normal temperature and pressure, NTP, 20°C, 1013 hPa). Thus, the dimension of the measurement result is mass per normal volume.

### 3.6.3    ERICA-AMS detection limits

Several methods can be used to determine the detection limit (DL) for the species measured by an AMS as described by Drewnick et al. (2009). One approach is the calculation based on the ion counting statistics during a measurement with the shutter closed (closed signal), denoted as $DL_{stat}$. The most common way is a measurement of the signal noise during a measurement of filtered air, denoted as $DL_{filter}$. Especially during in-flight measurements, this filter-based method cannot be representative for the whole flight due to changing vacuum, temperature, and instrument background conditions. For field

measurements we thus calculate a detection limit $DL_{spline}$ out of the closed signal after applying a spline-based detrending method comparable to Schulz et al. (2018) and Reitz (2011). In each case DL is defined as three times the standard deviation of the respective signal. The detection limits of all species are given in Table 1 for each method. The statistical approach as well as the filter-based method are based on a long-term filter measurement in the lab, while $DL_{spline}$ was determined from the measurements during the StratoClim 2017 campaign. The differences are reasonable, because $DL_{stat}$ does not consider

interferences with other species, especially water and air, whereas $DL_{spline}$ was measured under different conditions regarding pumping time and consequently instrument background. The detection limits are slightly higher than reported for other



airborne instruments (e.g., Schulz et al. (2018)), due to a different time basis, but also a rather strong airbeam signal in our instrument (see Sect. 3.6.4).

### 3.6.4 ERICA-AMS airbeam and water signal

The ADL is supposed to focus particles into a narrow beam into the vacuum chamber, while the air molecules are strongly diverging after the end of the lens. However, some of the air is also propagating towards the ion source and generates ions at $m/z$-ratios of 14 ($N^+$), 16 ($O^+$), 28 ($N_2^+$), 32 ($O_2^+$), 40 ($Ar^+$), and 44 ($CO_2^+$) as well as the corresponding isotopes. This signal, so called "airbeam" signal, can on one hand be used for diagnostic purposes, but on the other hand introduces uncertainties in measuring particle signals at the corresponding $m/z$. A small airbeam signal is thus desirable, e.g., to reduce the detection limit of aerosol species. In the ERICA-AMS we experienced a rather strong airbeam signal of around $2.9 \cdot 10^6$ ions $s^{-1}$ (see Fig. 14). This is larger than reported by Canagaratna et al. (2007) (1.5 to $2.5 \cdot 10^6$ ions $s^{-1}$), with a 5-fold higher IE value at the same time. We found out that the reason lies in the assembly of ERICA. Since the front part of the instrument was optimized for laser-ablation mass spectrometry, a rather large conical skimmer with an inner diameter of 1.9 mm was built in after the ADL for the separation of air and particles. While this causes no problem for the laser-ablation part, it leads to a substantial transfer of air molecules towards the following stages of the vacuum chamber. For improvement, we implemented a newly designed skimmer with an opening of 1 mm and a channel of 21.5 mm length in order to reduce the airbeam signal by a factor of 6.7 to $4.4 \cdot 10^5$ ions $s^{-1}$. Since this skimmer was implemented in 2019, earlier campaigns, like StratoClim 2017, were conducted with the large airbeam signal. Additionally, interferences of particle signals with the signal of residual water influence the detection limit of ammonium. Here, especially the background water vapor in the vacuum plays a role. We experience an intense water signal of $2.5 \cdot 10^6$ up to $1 \cdot 10^7$ ions $s^{-1}$ depending on instrument temperature and pumping time. This signal occurs independently of the shutter position and does thus not directly relate to the airbeam streaming into the instrument, but to the background vacuum conditions.

### 4 First aircraft borne measurements

The first field deployment of the ERICA was during two aircraft field campaigns as part of the StratoClim project. The main objective of the StratoClim project was to produce more reliable predictions of regional and global climate change through a better understanding of key microphysical, chemical, and dynamical processes in the upper troposphere and lower stratosphere (UTLS) of the Asian monsoon Rex et al. (2016); http://stratoclim.org, last access 30.08.2021). During the two aircraft field campaigns (43 flight hours), over 150,000 single-particle mass spectra were recorded and the ERICA-AMS provided reliable data for about 31.2 hours. By means of a satellite communication link to the operators (Dragoneas et al., 2021), the time of data losses could be kept low with 29 minutes for the ERICA-AMS and 39 minutes for the ERICA-LAMS. The first aircraft campaign took place in Kalamata, Greece, in August and September 2016 and the second in Kathmandu, Nepal, in July and August 2017. The high-altitude research aircraft M-55 *Geophysica* served as platform for these campaigns. With this platform it was possible to reach altitudes up to 20 km. It was the first time that bipolar single particle mass spectra were measured at altitudes above 16 km. Also, the ERICA-AMS was the first AMS type mass spectrometer that was successfully deployed to measure at such high altitudes. The analyses of the research flight data presented in this study serve to provide a proof of concept for ERICA, as well as to document its operational reliability and performance, without the purpose to provide details on the results connected with the scientific objectives. Detailed results from the aircraft field campaigns can be found, for example, in Höpfner et al. (2019), Schneider et al. (2021), and Appel et al. (2021). In the following, data examples from the second aircraft campaign of StratoClim 2017 in Kathmandu (KTM) are shown.





A selected bipolar single particle mass spectrum containing heavy metal signatures is presented in Fig. 15. The mass spectrum shows signals of light metals like sodium, magnesium, aluminum, and calcium, showing that the ERICA-LAMS is able to identify metals by their isotopic patterns. Furthermore, sulfate fragment ions and heavy metal ions of chromium, iron, molybdenum, and tungsten are present. The identification of iron, molybdenum, and tungsten was done by comparing the

signal intensity patterns with those of the natural abundance of the isotopes of the elements. The presence of molybdenum could be confirmed by signals for MoO$^+$, which has the same isotopic ratio as Mo$^+$. This particular mass spectrum was recorded at an altitude of ~20 km (a.m.s.l.) on 29.07.2017. Attributing this single particle to a certain source is difficult. However, an anthropogenic source as an exhaust of an aircraft engine, in which tungsten-molybdenum-alloys are in use (Guan et al., 2011), is conceivable due to its heavy metal signals.

We use the ablation efficiency $AE$ (see Sect. 3.4 for definition and limitations of $AE$) as a function of altitude to determine whether the ERICA-LAMS can measure over the entire sampled altitude range. The parameter $AE$ is instrument specific and independent of both the aircraft residence time and ambient particle number concentration. Fig. 16 shows the $AE$ vertical profile for the entire second aircraft campaign in 500 m bins. Here, the $AE$ values are between 0.1 and 0.3 over the entire altitude range. At maximum altitude, $AE$ is 0.24. These results demonstrate that single particle mass spectra can be recorded

both on the ground and at altitudes up to more than 20 km. Variations in $AE$ values may be due to differences in aerosol composition, size, and shape at different altitudes (Su et al., 2004; Brands et al., 2011). In addition to $AE$, the number of recorded single particle mass spectra $N_{spectra}$ and the number of ablation laser shots $N_{shots}$ also show that mass spectra can be recorded in all sampled altitude ranges (up to 20.5 km; Fig. 16). However, $N_{spectra}$ and $N_{shots}$ depend on the residence time of the aircraft at the respective flight altitude, which was long at altitudes above 15 km and also below 5 km.

After demonstrating that it is possible to measure with the ERICA at flight altitudes up to about 20 km, in the following we show that aerosol species known in the literature can be identified with both, the ERICA-LAMS and the ERICA-AMS. The evaluation of the data was carried out separately for the ERICA-LAMS and the ERICA-AMS. For the ERICA-AMS, the species reported in Sect. 3.6.1 were quantified. To determine specific particle types of the single particles, the ERICA-LAMS data set was processed with the software CRISP (Klimach, 2012) using the k-means clustering algorithm as described in Roth

et al. (2016). In this processing, all single particle mass spectra were pre-sorted into a predefined number of so-called clusters and then manually combined into meaningful particle types. In this way, two particle types (in addition to other particle types not included in this publication) well described in the literature were found: A meteoric material containing (e.g., Schneider et al. (2021)) and an elemental carbon (EC) containing particle type (e.g., Pratt and Prather (2010)). In the following, we focus on the aerosol composition at high altitudes (> 10 km) considering particulate sulfate and the meteoric material containing

particle type.

The sulfate particle type measured by the ERICA-AMS is a non-refractory species (Canagaratna et al., 2007) and consists mainly of pure sulfuric acid in the stratosphere (Murphy et al., 2014). The mass fraction is the calculated fraction of the mass concentration of the sulfate species over the total mass concentration determined by the ERICA-AMS for each altitude bin. In Fig. 17, the vertical profile of the sulfate mass fraction of the research flight of 04.08.2017 is depicted. The profile shows an

enhancement at altitudes starting at 17.5 km. In 20 km altitude, the sulfate mass fraction is 1. This result can be expected due to the proximity of the Junge-layer, where the aerosol particles mainly consist of pure sulfuric acid (Junge and Manson, 1961; Murphy et al., 2006b).

To identify the sulfate-containing single particle spectra (ERICA-LAMS), the data set of the research flight of 04.08.2017 was filtered for single particle spectra that contained sulfate marker signals at $m/z$ -96 (SO$_4^-$) or $m/z$ -97 (HSO$_4^-$) or both markers.

Fig. 17 shows the vertical profile of the particle number fraction of the sulfate containing single particles. A particle number fraction is the fraction of a particle type out of all mass spectra recorded in the respective altitude bin (bin size 500 m). In the vertical profile, a large number fraction of about 0.6 of the sulfate-containing single particles can be seen between 10 and 17 km (ERICA-LAMS) that increases with higher altitudes up to a maximum value of 1.



As identified and described by Murphy et al. (1998) Cziczo et al. (2001), the meteoric material containing particle type is characterized by a high abundance of magnesium (Mg$^+$, isotopes at $m/z$ 24, $m/z$ 25, and $m/z$ 26) and iron (Fe$^+$, isotopes at $m/z$ 56 and $m/z$ 54) signals in the cation spectrum and of sulfate (HSO$_4^-$ at $m/z$ -97) in the anion spectrum. The occurrence of the described characteristic signals in the single particle mass spectra of the ERICA-LAMS and the dominant presence of

the meteoric material containing particle type at high altitudes (> 17 km) were already published by Schneider et al. (2021). The mean spectrum can be found in Sect. S6 in the supplement. Fig. 17 exemplarily shows the abundance of meteoric material in the vertical profile of the research flight on 04.08.2017 in the particle number fraction of the meteoric material containing particle type. The particle number fraction is larger than 0.6 above 19.5 km and reaches its maximum of 0.8 at the maximum flight altitude of the research flight. The increase in particle number fraction of the described meteoric particle type at high

altitudes is also described for measurements with other mass spectrometers, like the PALMS and the ALABAMA (Murphy et al. (2014) and Schneider et al. (2021)). Furthermore, similar particle number fraction values of up to 0.6 were also reported for a similar particle type recorded in the mid-latitude stratosphere by Murphy et al. (2014). The demonstrated results of the meteoric material containing particle type can be considered as indication of the reliable operation of the ERICA-LAMS at high altitudes such as up to 20 km.

The measurements of the two instrument parts, ERICA-LAMS and ERICA-AMS, were evaluated separately and the derived results complement each other. Pure sulfuric acid cannot be ablated with the frequency quadrupled Nd:YAG laser (wavelength 266 nm) used in the ERICA-LAMS, because light of this wavelength is not efficiently absorbed by the particles (Murphy, 2007). Vice versa, the meteoric particles consists of refractory components that can be detected by the ERICA-LAMS, but not by the ERICA-AMS. The analyses presented here as examples show that the ERICA can be used to measure aerosol

components, such as sulfuric acid and meteoric material, that are significantly present in the stratosphere by means of the two complementary measurement methods. The results can also be used to show that the aerosol composition between 10 km to 17 km differs from the aerosol composition above 17 km. For this, the mass fraction of sulfate (ERICA-AMS) and the number fraction of sulfate-containing single particle spectra (ERICA-LAMS) were examined. Below 17 km, the number fraction of sulfate-containing single particle spectra is stable around 0.6 and the mass fraction of the sulfate less than 0.2. This could be

indicative for an internal mixing state of the measured aerosol particles, where the sulfate species within the single particles is assumed as predominantly refractory compound, since the mass fraction of the sulfate species is low compared to the number fraction of sulfate-containing particles. The reason is that the ERICA-AMS only can measure non-refractory substances. Above 17 km, the composition is more complex. With increasing altitude, the sulfate mass fraction and the particle number fraction of sulfate-containing single particles increase up to 1. The change in mass fraction is strong compared to the number fraction

of sulfate-containing single particles. Therefore, it can be assumed that the non-refractory content increases. Since the ERICA-LAMS is not able to detect pure (non-refractory) sulfuric acid, no distinct determination of the mixing state can be obtained. Here, an internal or an external mixing state but also a combination of both states can be present. In a conceivable internal mixing state, the non-refractory sulfuric acid has deposited on a particulate core, generating a coated particle or the sulfuric acid acts as a condensation nucleus for other substances. Additional pure sulfuric acid particles lead to an external mixing

state.

As described above, the EC particle type was identified using the k-means clustering for the data set. The EC particle type is characterized by an C$_n^+$ pattern in the cation and an C$_n^-$ pattern in the anion spectrum (e.g., Hinz et al. (2005)). Fig. 18a shows the mean spectrum of the recorded EC particle type mass spectra (total number 389) during the StratoClim research flight of

08.08.20217. Here, the described signal pattern is evident in both polarities. Fig. 18b displays the vertical distribution of the particle number fraction of all EC-containing particles in the research flight on 08.08.2017 (vertical bin size 500 m). As expected, the particle number fraction of EC is enhanced in the lowest 6 km with a value of around 0.05. EC is created as primary aerosol by combustion processes as part of soot at low altitudes (Turpin et al., 1991; Seinfeld and Pandis, 2016).

 

Combustion is a common source of air pollution in Nepal (Saud and Paudel, 2018; Sadavarte et al., 2019). Field measurements with the established single particle mass spectrometer A-ATOFMS that is comparable to the ERICA were conducted in the USA. Pratt and Prather (2010) found a stable EC particle number fraction of also around 0.05 in the altitude range of 1 to 6 km. This comparison with the A-ATOFMS shows that the ERICA provides credible results at low altitudes. We observed

another enhancement of the EC particle number fraction in the altitude range between 7 and 15 km and assume that the occurrence of EC-containing particles in this altitude range can be caused either by local emitters, such as aircraft (Liu et al., 2017), or by vertical transport, such as the convective outflow of the Asian monsoon (Garny and Randel, 2016). Above 16 km, the EC particle number fraction is very low, ranging around 0.01.

Pure soot is a refractory compound and, consequently, cannot be detected by the ERICA-AMS (Canagaratna et al., 2007). On

the other hand, the ERICA-AMS is capable of providing quantitative mass concentration of the non-refractory components of ambient aerosol and thus is well suited for the identification of particle layers by quantitative means. The total ERICA-AMS mass concentration $C_{total}$ is defined as the sum over all non-refractory aerosol species. Fig. 18c depicts the vertical profile of $C_{total}$ for the research flight on 08.08.2017. An enhancement in the total mass concentration is clearly evident for altitudes from ground level to approximately 3.5 km and can be associated with anthropogenic emissions at ground. This layer can be

seen as the particle boundary layer, similar to the definition used by Schulz et al. (2018). In the particle boundary layer, we found during the flight (monsoon season measurement) a maximum $C_{total}$ of 6.9 µg m$^{-3}$ at an altitude of 2 km. At ground level, a $C_{total}$ of 4.8 µg m$^{-3}$ was found for this flight. Pre-monsoon season PM$_{2.5}$ filter measurements (April 2015) in the Kathmandu valley show typical $C_{total}$ values between 30.0 and 207.4 µg m$^{-3}$ (Islam et al., 2020) at ground level. Due to particle scavenging processes, $C_{total}$ is lower during the monsoon season (Hyvärinen et al., 2011). The second enhancement

(at altitudes between 15.5 and 19.5 km) with a maximum of 2.8 µg m$^{-3}$ can be associated to the ATAL (e.g., Vernier et al. (2011)). In the free troposphere (at altitudes between 4 and 15 km), $C_{total}$ goes down to approximately 1 µg m$^{-3}$. The results from the non-refractory $C_{total}$ can be discussed together with the particle number fraction of the refractory EC particle type to provide complementary information about the sampled aerosol particles. Within the particle boundary layer, as measured by the ERICA-AMS, $C_{total}$ decreases whereas the EC particle number fraction is stable, as in the free troposphere. This indicates

within the limitations of the applied methods that the composition of the sampled aerosol is well mixed within the particle boundary layer and in the free troposphere, although $C_{total}$ changes. Thus, the EC particle number fraction cannot be used to define the particle boundary layer. In the ATAL, EC particles seem to play a minor role in the composition of the aerosol, while for the convective outflow levels the data suggest an increase in EC as result of detrainment. (This StratoClim flight on 08.08.2017 was performed at a time of high convective activity and in the presence of large cloud systems above the Himalayan

foothills.) Overall, the studies presented here confirm that the ERICA can be adopted for aircraft missions from ground level up to an altitude of 20 km and operate reliably under demanding field conditions. A more comprehensive evaluation of the collected data will be conducted in further studies.

As an example that the ERICA-LAMS provides single particle size information, Fig. 19 shows the size distribution of EC-containing particles for the research flight on 04.08.2017 consisting of three modes. The first at the edge of the small particle

sizes below 200 nm, the second between a particle size of around 300 nm and 1700 nm with a maximum particle number fraction of 0.08 at 800 nm, and the third between 1700 nm and 2600 nm with a maximum of 0.17.

## 5   Summary and outlook

In this study we present a novel aerosol mass spectrometer combining a laser ablation technique (ERICA-LAMS; quadrupled

Nd:YAG laser at λ=266 nm) with a vaporization and electron impact ionization technique (ERICA-AMS; vaporizer operated at a temperature of 600 °C, electron impact energy of 70 eV). These techniques are implemented in two consecutive instrument





stages that are connected in series within a common vacuum chamber. The use of a common vacuum chamber and other components for both measurement techniques, minimizes weight and volume of the instrument. The resulting compact dimensions enable the instrument to be deployed on aircraft, ground stations, and mobile laboratories. By that, the same aerosol sample can be investigated with two different physical methods. The chemical characterization of single particles is achieved

by recording bipolar mass spectra with a B-ToF-MS. For the non-refractory components, the cations are detected with a C-ToF-MS. By deploying both methods, complementary chemical information can be obtained. By means of the laser ablation, single particles consisting of refractory or non-refractory components, are qualitatively analyzed, while the flash vaporization and electron impact ionization technique provides quantitative information on the non-refractory components (i.e., particulate sulfate, nitrate, ammonium, organics, and chloride) of small particle ensembles.

Comprehensive laboratory measurements with PSL and AN test aerosol were conducted to characterize the key instrumental parameters. Focused laser beams of the PDUs and the ablation laser beams as well as the particle beam were investigated. In order to determine the particle beam characteristic parameters, ADL position scans with particles of various sizes were performed. The parameters presented in this publication are: the PDU and ablation laser beam waist radii ($w_{0,dia}$), the particle beam width ($w_{part}$), the effective detection radius of the PDUs ($r_{eff,L}$) and of the vaporizer ($r_{eff,V}$), the particle beam overlap

parameters ($S_{detect,L}$, $S_{detect,V}$, and $S_{ablation}$), and the transmission efficiency of the ADL ($A_{scan}$), each as function of particle size. Extensive information about the beam characteristics were obtained and show the performance of the ERICA. Here, 1σ overlap of the particle beam with the detection laser spot for particle sizes between 213 nm and 3150 nm was found. The installed ADL is described in the literature (Peck et al., 2016; Xu et al., 2017) and covers a particle size range of ~120 nm to 3.5 µm ($d_{50}$). We found that the particle beam hits the vaporizer completely even at sizes as low as 91 nm. The evaluation of

the particle beam shift resulted in two cases of the optical particle detection efficiency, due to a non-concentric focusing of all particle sizes: the maximum optical detection efficiency ($DE_{max}$) that theoretically can be achieved and the optical detection efficiency during the field campaign in Kathmandu ($DE_{KTM}$). The characterization shows that $DE_{max}$ at the PDUs reaches a value up to 1.00 compared to a reference instrument in a laboratory setup and shows an optical detectable size range of 180 nm to 3170 nm ($d_{50}$) for PSL particles. During the field campaign in Nepal the optical particle detection efficiency $DE_{KTM}$ reached

up to 0.86. As $d_{50}$ values for the $DE_{KTM}$ 190 nm and 745 nm can be stated for PSL particles (at PDU1). Particle time-of-flight calibration was performed for particle sizes between 80 nm and 5145 nm. Furthermore, the particle time-of-flight calibration agrees well with the measurements performed with AN particles. The evaluation of scattered light intensities for particle size determination is also conceivable, but not implemented yet.

The capabilities of the ERICA were tested in field and laboratory experiments. After the adjustment preparation procedure as

conducted before any field campaign, a ground-based field experiment was conducted to determine the size resolved ablation efficiency of the ERICA-LAMS. The result was a maximum $AE$ of 0.52 for a particle size of around 230 nm. The outcome of this experiment reflect the results of the particle beam characterization measurements. In addition, we measured pure chemical substances from solutions or suspensions in order to validate that ERICA-LAMS raw mass spectra can be m/z calibrated by the software CRISP correctly. Beside sodium chloride, ammonium nitrate, and benz[a]anthracene, gold spheres were sampled.

All substances could be identified by their specific marker peaks in the mass spectra after CRISP processing. Furthermore, mass spectra resolution $R_{MS}$ values of 200 for $m/z$ 120, 700 for $m/z$ 200 (both cations) and of about 600 for the anion spectra were determined and are comparable to similar single particle mass spectrometers. For the ERICA-AMS, $R_{MS}$ was determined by the evaluation software "Tofware" to be 800 for m/z 200 that is also comparable to other C-ToF-MSes. The conversion of the ion time of flight into a mass spectrum is based on six predefined calibration peaks. A major difference from a commercial

AMS instrument is that the ERICA AMS features a shutter instead of a chopper. By means of the shutter, the background signal (shutter closed) can be determined and then subtracted from the "shutter open" signal. The fragmentation table implemented in "Tofware" allows the determination of various species, such as organic, nitrate, sulfate, ammonium, and chloride. By means of an IE calibration, the determined sample signal can be turned into an aerosol mass concentration. The



IE calibration procedure was conducted with monodisperse AN particles using a CPC as reference device and yielded $2.05 \cdot 10^{-7}$ ions per molecule. For the detection limits, results for five aerosol particle species were obtained and presented for three different methods. Also, for the StratoClim 2017 campaign a valid airbeam signal of $2.9 \cdot 10^6$ ions s$^{-1}$ and a water signal between $2.5 \cdot 10^6$ and $1 \cdot 10^7$ ions s$^{-1}$ were found. Subsequent modification of a skimmer reduced the airbeam by a factor of 6.7 for future

instrument deployments. The losses due to particles ablated and hence not contributing to ERICA-AMS signal were determined to be low and within the AMS's measurements uncertainties of 30 % for most atmospheric conditions. However, for low particle concentrations the losses have to be considered, but they are hard to quantify. Therefore, the operation of the ERICA-LAMS part would need to be paused, at least intermittently, to enable undisturbed quantitative measurements by the ERICA-AMS. This procedure can be implemented into the automated mode. With a similar mode it would be possible to investigate

the fraction of charged ambient particles by switching the HV switch on and off in defined intervals.

The two aircraft field campaigns as part of the StratoClim project in 2016 and 2017, were the first field deployments of the ERICA. This was the first time an AMS type mass spectrometer was deployed above 16 km, as well as the first bipolar single particle mass spectra were recorded at these altitudes. Mass spectra examples from high altitudes presented here agree with spectra presented in the literature and show that ERICA delivered reasonable data even under field conditions during

autonomous operation aboard a research aircraft. For the ERICA-LAMS, the meteoric material containing particle type, and for the ERICA-AMS, the sulfate species are used for a proof-of-concept of the operation at stratospheric altitudes. For low altitudes, down to ground level, the EC particle type and total mass concentration serve as examples of the capabilities of the ERICA-LAMS and ERICA-AMS, respectively. The vertical profiles of these species and additionally of the *AE* show a reasonable instrument performance over the entire altitude range from ground level up to 20 km. In this study, we also show

that ERICA-LAMS and ERICA-AMS can provide complementary information about the sampled aerosol. Some limitations of one ionization method can be partially compensated by the other. We estimated the mixing states in and a few km below the UTLS and assume that the particles are externally and internally mixed.

Although the ERICA-LAMS and ERICA-AMS combination was developed for the aircraft deployment within the ATAL and
the combination has been shown to perform reliably in field campaigns, in the future modifications could be made to the instrument to address other scientific questions. One modification might be the implementation of another laser type such as an excimer laser for measurements in the lower stratosphere (Murphy et al., 2007). While this is possible for ERICA as well, space and weight limitations inherent in the implementation prevented the use of an excimer laser setup on the M-55 *Geophysica*. However, the light at the longer ablation laser wavelength generates less fragmentation in the mass spectra
(Thomson et al., 1997). Furthermore, the mass spectra recorded with ERICA are in a higher degree comparable with instruments like the A-ATOFMS (Gard et al., 1997) and the ALABAMA (Brands et al., 2011), which operate also with an ablation laser at a wavelength of 266 nm. In another upcoming development, an additional single particle mode for the ERICA-AMS will be added, which will be based on optical particle detection. As with LAMS, a single particle is optically detected by the PDUs and by means of the TC the point in time is calculated when the particle hits the vaporizer. For the same point in
time, a data acquisition card is triggered and, similar to the procedure with a light scattering probe on the AMS (Cross et al., 2007; Freutel, 2012), the single particle mass spectrum is recorded. In this way it is possible to quantify the non-refractory components of a single particle. In addition, the size information of the measured single particle is obtained by means of the particle flight time between the two PDUs. Here, a future characterization of interest is the ablation laser's effect to the particles that are only partly ablated and the residuals reach the vaporizer of the ERICA-AMS. For this purpose, a method has to be
developed to ensure the linkage of the results to the very same particle. Such a procedure needs more implementations and further laboratory studies.

The presented examples of field measurements showed that the instrument has already been successfully operated during the aircraft campaign of the StratoClim project. The evaluation of the data is ongoing and will be presented in further publications.



Furthermore, the ERICA was successfully deployed during the ND-MAX/ECLIF-2 (NASA/DLR-Multidisciplinary Airborne eXperiments/Emission and CLimate Impact of alternative Fuel; Voigt et al. (2021)) field campaign in January to February 2018 (Schneider et al., 2021) and during the ACCLIP (Asian summer monsoon Chemical and CLimate Impact Project) test phase in January and February 2020. The main campaign will be set up in July to August 2022 based in South Korea

(https://www.eol.ucar.edu/field_projects/acclip, last access 30.08.2021).

**Data availability**

Data can be accessed by contacting the corresponding author Stephan Borrmann (stephan.borrmann@mpic.de).

**Authors contributions**

SB provided the instrumental concept and an initial design in his ERC Advanced Research Grant proposal. SB, FD, and JS initiated the instrumental design and accompanied its development and characterization. FH and TK designed the detection units. OA, TB, AD, AH, and SM developed the instrument. OA, AD, AH, and SM performed the described measurements in the field and in the lab. OA and AH evaluated the data. The lens scan evaluation method was developed by TK. HC initiated and accompanied the implementation of the HV switch and the electric shielding of the ion optic as an essential improvement.

AH, together with SB, OA, AD, FK, and SM drafted the manuscript. All co-authors provided detailed comments on the manuscript.

**Competing interests**

The authors have the following competing interests: Johannes Schneider is associate editor of AMT.

**Acknowledgements**

We gratefully thank the workshops of the Max Planck Institute for Chemistry and of the Institute for Physics of the Atmosphere (Mainz University) and Tofwerk AG, in particular C. Gurk, H. Schreiber, B. Meckel, D. Gottert, S. Best, J. Sody, and U. Rohner, for the essential support. The help of M. Cubison for customizing "Tofware" is gratefully acknowledged. Special thanks are due to W. Xu and P. Croteau from Aerodyne Research, Inc. for the specification measurements of the ADL

deployed. We would like to express our gratitude to F. Stroh for his extraordinary commitment to the realization of the field campaigns and to M. Rex for managing the entire StratoClim project. Our special thanks are extended to the crew of MDB (Myasishchev Design Bureau) and the M-55 *Geophysica* pilots. This work was financially supported by the Max Planck Society and the European Research Council under the European Union's Seventh Framework Program (FP/2007-2013)/ERC Grant Agreement No.321040 (EXCATRO). The StratoClim project was funded by the EU (FP7/2007–2018 Grant No. 603557)

and supported by the German Federal Ministry of Education and Research (BMBF) under the joint ROMIC-project SPITFIRE (01LG1205A). We extend our sincere thanks to the Greek government authorities and Kalamata International Airport, as well as the Nepalese government authorities, research institutions and Tribhuvan Airport, as well as the German Embassy, for their extraordinary support and hospitality that made the StratoClim field campaigns and our research possible.



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





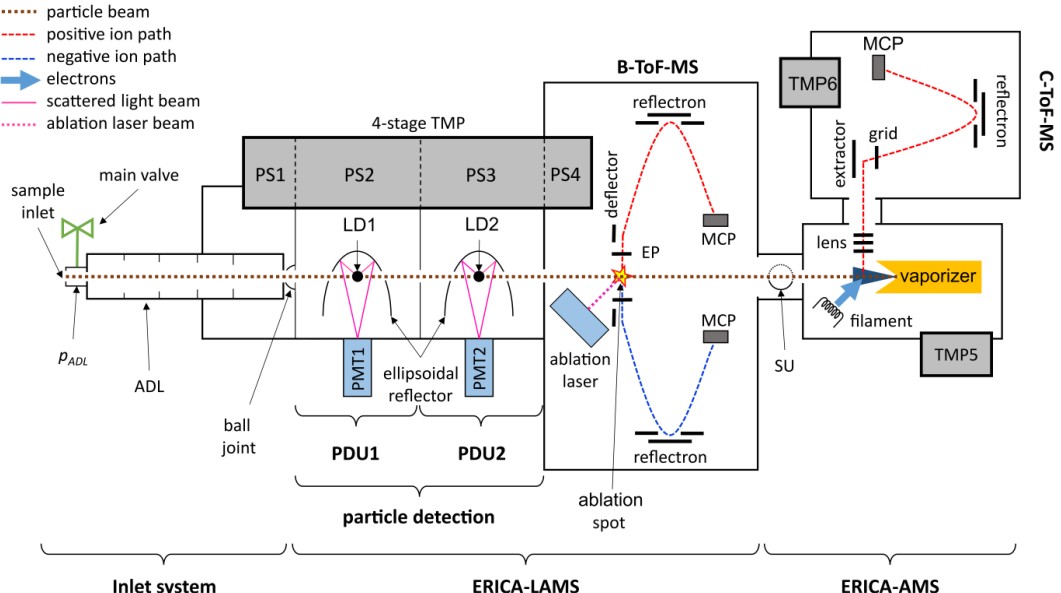

**Fig. 1: Overview of the ERICA setup. (ADL – aerodynamic lens, LD – laser diode, EP – extraction plates, MCP – micro-channel plate, PDU – particle detection units, PMT – photomultiplier tubes, PS – pumping stage, SU – shutter unit, TMP – turbo molecular pump). The additional backing pump for the TMPs is not shown. The detection laser beams and the ablation laser beam enter the vacuum chamber perpendicularly to the plane of drawing. The constant pressure inlet (not shown) is located upstream of the main valve.**

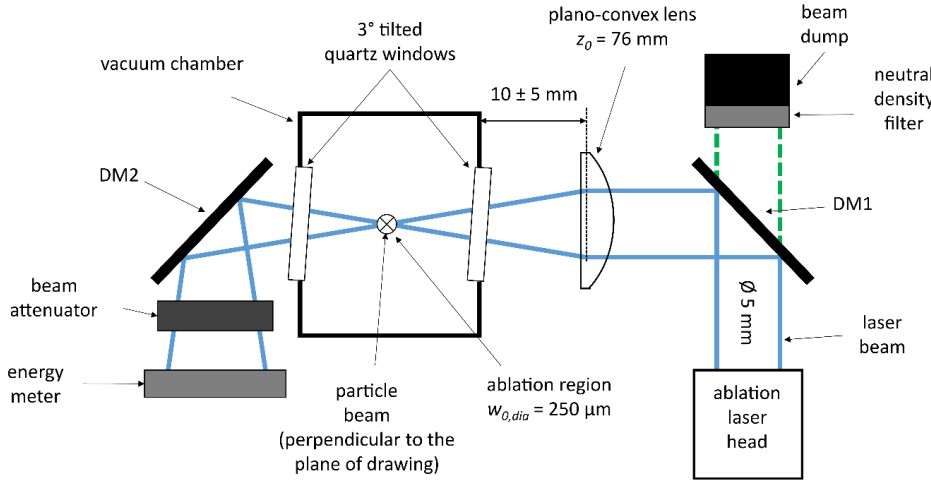

**Fig. 2: Schematic of the ablation laser unit of the ERICA-LAMS and corresponding optical dimensions ($z_0$: focal length; $w_{0,dia}$: laser beam focus $1/e^2$-diameter). The particle beam is pointing perpendicularly to the plane of the drawing. The dichroitic mirrors are labelled as DM1 and DM2.**



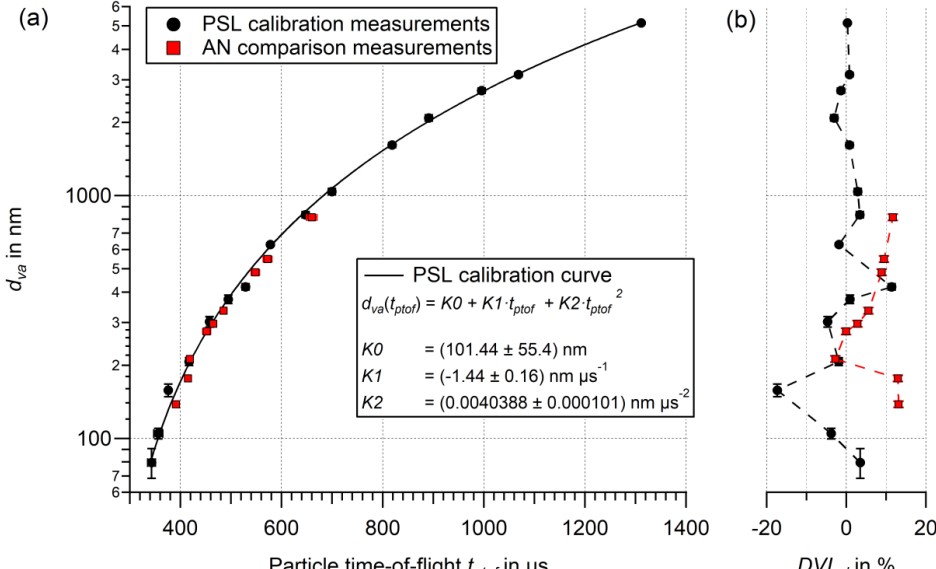

**Fig. 3: (a) particle time-of-flight calibration curve ($d_{va}$ as a function of $t_{ptof}$, continuous line) of PSL particles (black markers). For comparison of AN measurements to the calibration curve, the particle size of the measured AN particles is depicted as a function of the measured $t_{ptof}$ (red markers). (b) relative deviation of the NIST particle size standard measurements (black markers) and AN comparison measurements (red markers) from the calibration curve $DVI_{rel}$ according to Eq.**

**(3) as function of $d_{va}$ (black markers). The uncertainty of PSL particle size is given by NIST certificates and converted to $d_{va}$. The uncertainty of AN particle size $d_{va}$ is estimated to be 3 % (Hings, 2006). These uncertainties for PSL and AN particle sizes are the same for Fig. 3 and all Figs. 5 to 10. The uncertainty of particle flight time is calculated from $1\sigma$ (from histogram curve-fitting). The error bars are, in some cases, smaller than the symbol. $K0$, $K1$, $K2$ are parameters from the polynomial function**

**used for the particle time-of-flight calibration.**

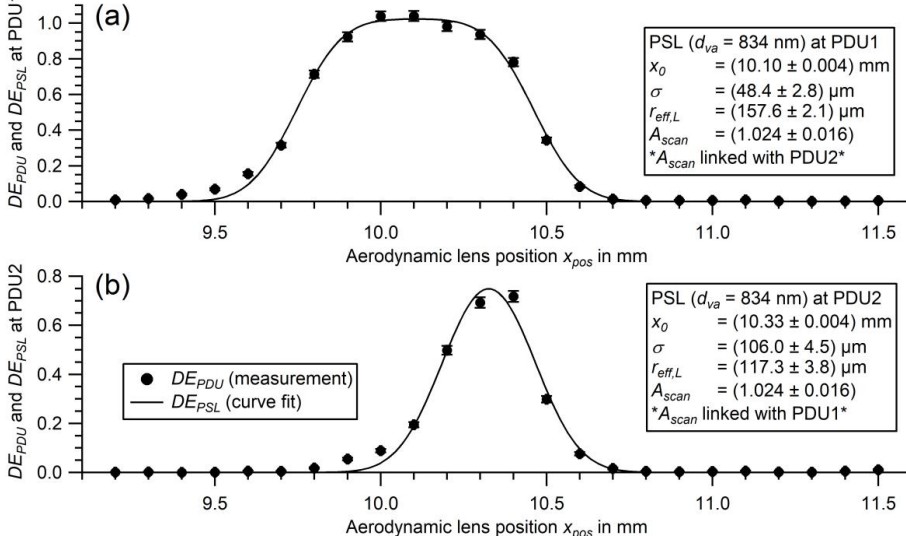

**Fig. 4: Scan of the ADL position ($x_{pos}$) with PSL particles with a size of $d_{va}$ = 834 nm perpendicular to the laser beam at PDU1 (a) and PDU 2 (b). Displayed are the $DE_{PDU}$ values of the measurement (markers) according to Eq. (4) and the curve-fit ($DE_{PSL}$; line) according to Eq. (5). The results of the curve-fits are shown in the box. The values of $\sigma$ and $r_{eff,L}$ were rescaled according to the**

**instrument's geometry (see Sect. S1.2 in the supplement), using the intercept theorem, for further evaluation. The uncertainty of the detection efficiency is based on counting statistics. The uncertainty of the lens position results from reading errors at the micrometer screw. The error bars are, in almost all cases, smaller than the symbol.**





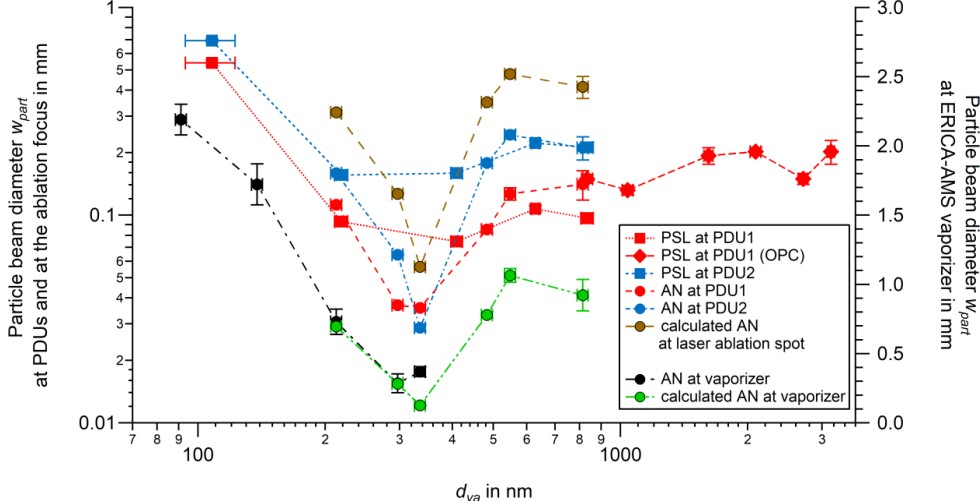

**Fig. 5:** The particle beam diameter $w_{part}$ ($\frac{1}{\sqrt{e}}$-diameter) as a function of particle size $d_{va}$ for PSL (squares) and AN (circles) particles measured at the detection units PDU1 (red, left ordinate) and PDU2 (blue, left ordinate), and for AN particles measured at the ERICA-AMS vaporizer (right ordinate, black). The reference values for number concentrations were either obtained from the experimental setup with the CPC or the OPC (Setup B or C, respectively, see Fig. S7 in the supplement). The AN particle beam diameter at the ablation spot (brown, left ordinate) and the ERICA-AMS vaporizer (green, right ordinate) were calculated by extrapolation of the measurement at PDU2. The uncertainties of the particle beam diameters result from the curve-fittings (one standard deviation). The error bars are, in some cases, smaller than the symbol.

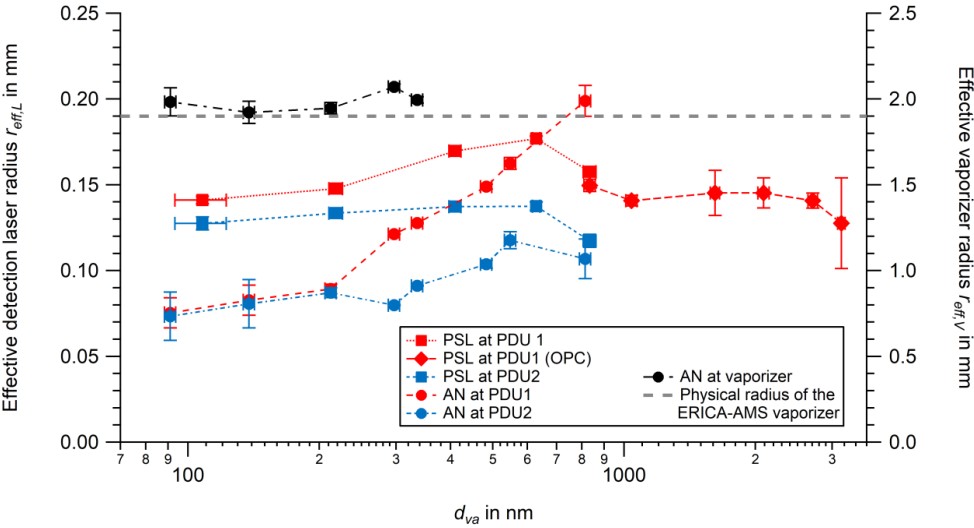

**Fig. 6:** The effective detection laser radius $r_{eff,L}$ as a function of particle size $d_{va}$ determined for PDU1 (red, left ordinate) and PDU2 (blue, left ordinate) with PSL (squares) and AN (circles) particles, and the effective vaporizer radius $r_{eff,V}$ as a function of particle size $d_{va}$ for the ERICA-AMS vaporizer (right ordinate, black) determined with AN particles. CPC and OPC measurements as for Fig. 5. The physical vaporizer radius is marked by a dashed gray line. The uncertainties of the effective radii result from the curve-fittings (one standard deviation). The uncertainty of $r_{eff,L}$ for the PSL measurement with particle size of 108 nm was estimated to be 0.002 mm (PDU1) and 0.004 mm (PDU2) and the uncertainties of $r_{eff,L}$ for the AN measurements with particle sizes of 138 nm and 91 nm are conservatively estimated to be 0.009 mm at PDU1 and 0.014 mm at PDU2. These values are the approximated maximum uncertainties of $r_{eff,L}$ in the considered size range of 213 nm to 814 nm at PDU1 and PDU2. For the measurement with AN particles of 91 nm in diameter, the uncertainty of $r_{eff,V}$ was estimated to be 0.08 mm. The error bars are, in some cases, smaller than the symbol.





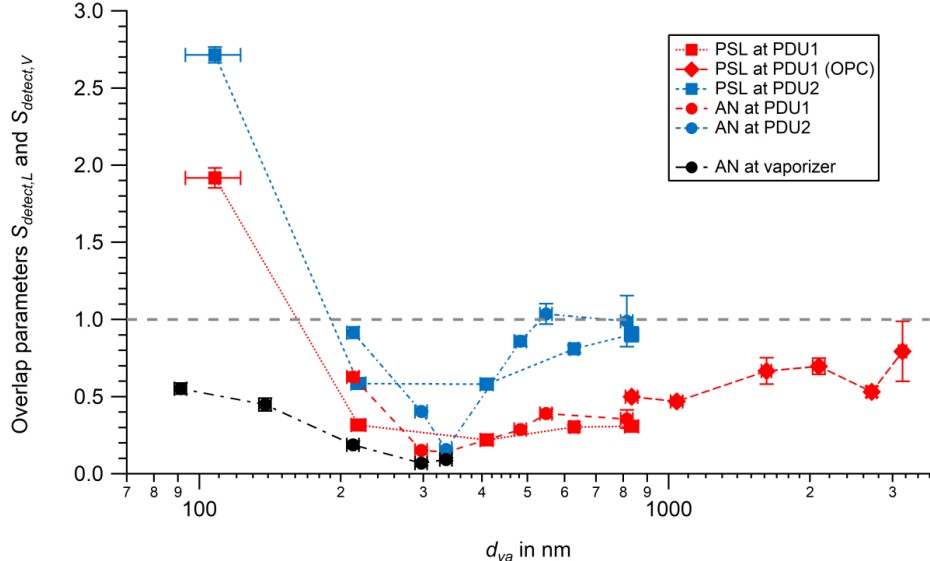

**Fig. 7: The overlap parameters $S_{detect,L}$ and $S_{detect,V}$ as a function of particle size $d_{va}$ for PSL (squares) and AN (circles) particles measured at PDU1 (red) and PDU2 (blue), and for AN particles measured at the ERICA-AMS vaporizer (black). CPC and OPC measurements as for Fig. 5. The gray horizontal dashed line illustrates where the ratio equals 1. The uncertainties of $S_{detect,L}$ and $S_{detect,V}$ result from the curve-fitting values (one standard deviation). The error bars are, in some cases, smaller than the symbol.**

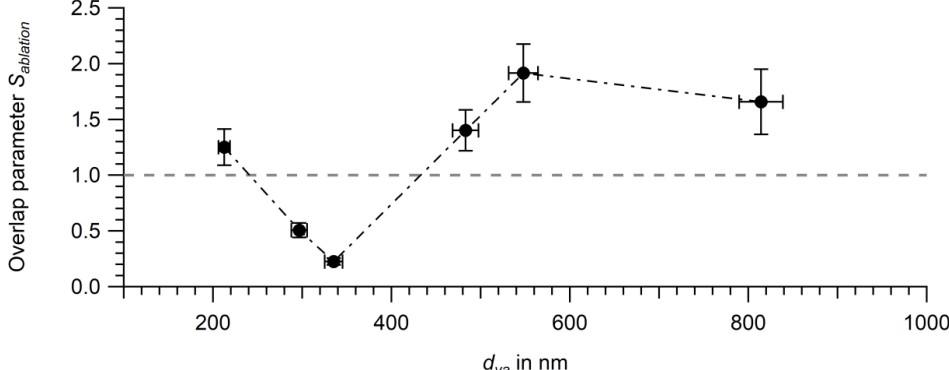

**Fig. 8: The overlap parameter $S_{ablation}$ as a function of particle size ($d_{va}$) for AN particles at the ablation spot. The gray horizontal dashed line illustrates where the ratio equals 1. The uncertainties of $S_{ablation}$ result from the curve-fitting values (one standard deviation). The error bars are, in some cases, smaller than the symbol.**





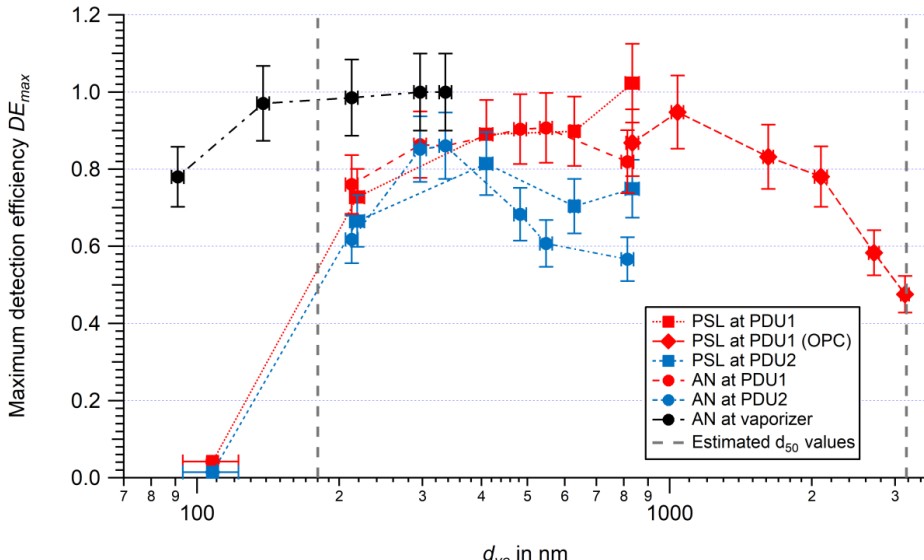

**Fig. 9: Maximum detection efficiency $DE_{max}$ as a function of particle size $d_{va}$ for PSL (squares) and AN (circles) particles measured at PDU1 (red) and PDU2 (blue), and for AN particles measured at the ERICA-AMS vaporizer (black). CPC and OPC measurements as for Fig. 5. The estimated $d_{50}$ values of the optical detection are shown as gray vertical dashed lines , whereas the $d_{50}$ values of the AMS measurement lie outside the applied particle range. The uncertainties of $DE_{max}$ reflect the conservatively estimated value of 10 %. The error bars are in some cases smaller than the symbol.**

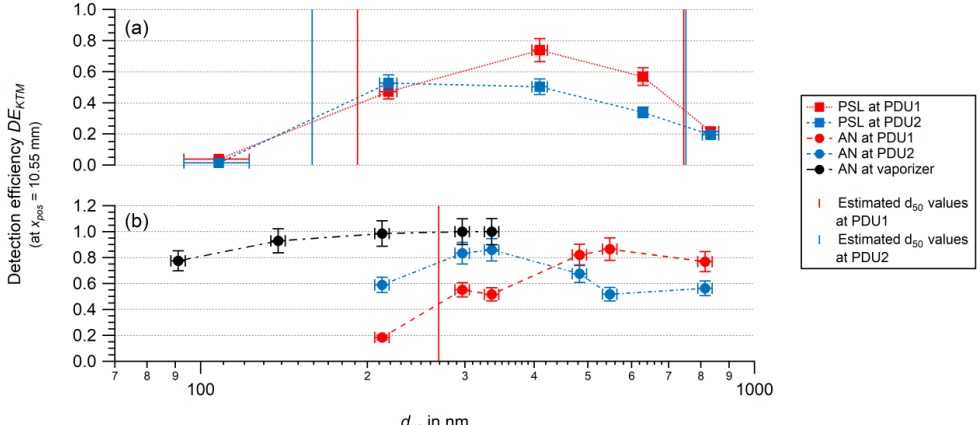

**Fig. 10: Detection efficiency $DE_{KTM}$ as function of particle size $d_{va}$ experimentally determined for PSL (squares, panel a) and AN (circles, panel b) particles measured at the detection units PDU1 (red) and PDU2 (blue), and the ERICA-AMS vaporizer (black) for the ADL setting during field deployment in Kathmandu, Nepal. The estimated $d_{50}$ (50 % of the maximum) values are shown as vertical lines (PDU1: red; PDU2: blue). The uncertainties of $DE_{KTM}$ reflect the conservatively estimated value of 10 %. The error bars are in some cases smaller than the symbol.**





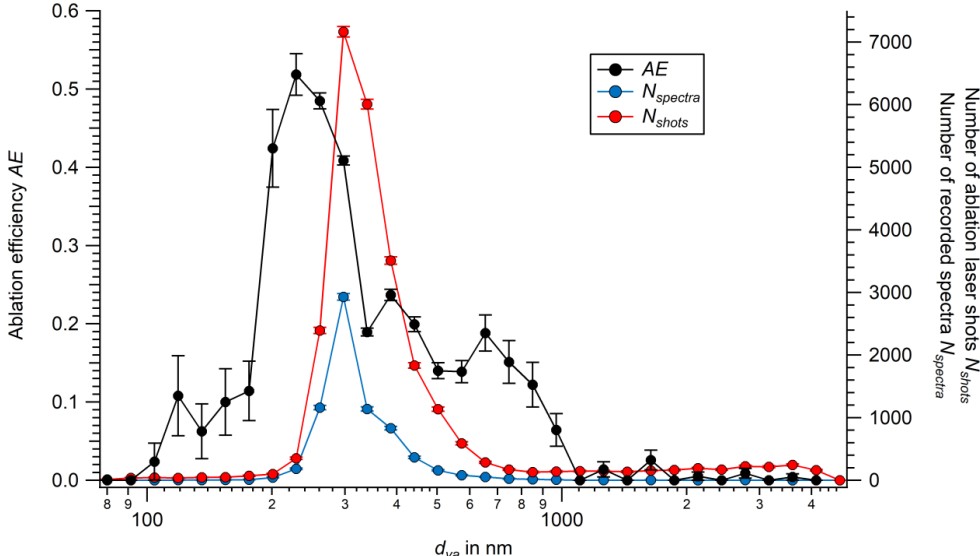

**Fig. 11:** The ablation efficiency $AE$ (black, left ordinate), the number of spectra $N_{spectra}$ (blue, right ordinate), and the number of detected particles, i.e., ablation laser shots $N_{shots}$ (red, right ordinate) as a function of particle size $d_{va}$ (logarithmic bin size) for ambient urban aerosol. Only the spectra with size information within the calibrated size range were processed (see Sect. 3.2). Uncertainties of $AE$, $N_{shots}$, and $N_{spectra}$ are based on counting statistics. The error bars are in some cases smaller than the symbol.





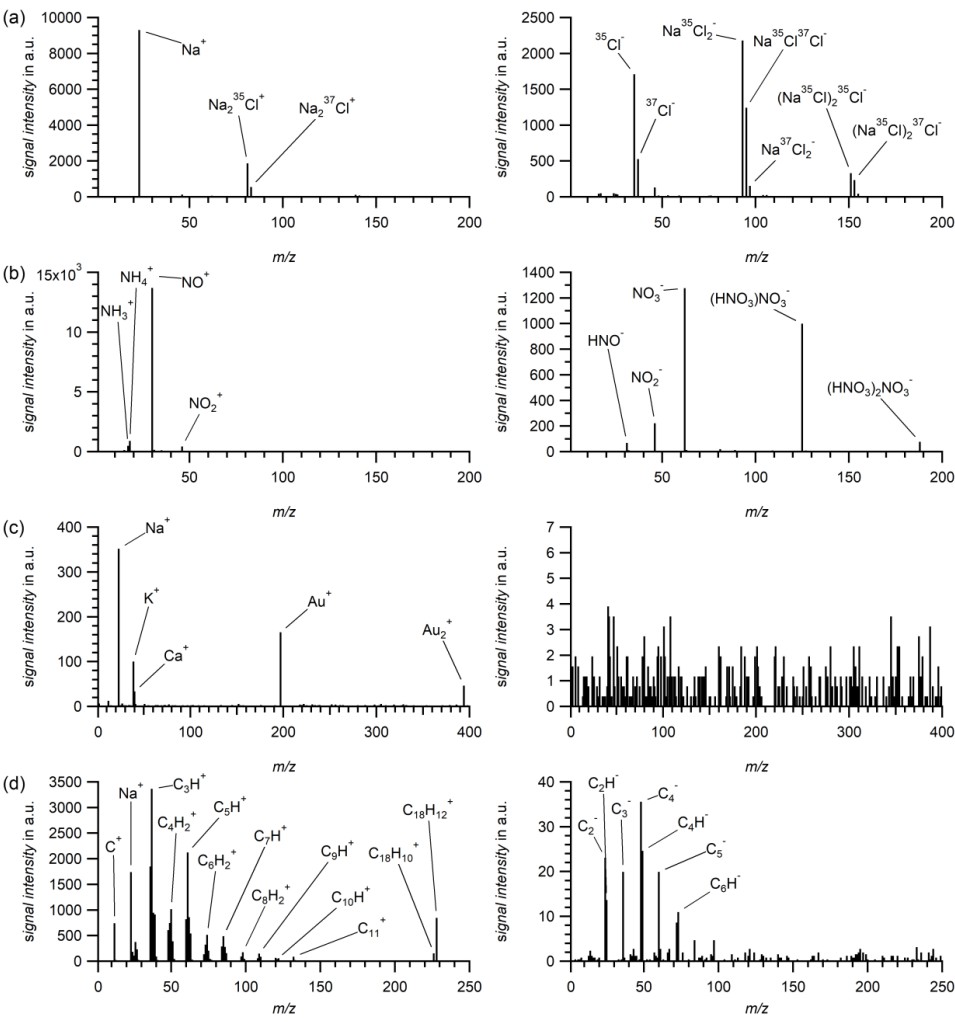

Fig. 12: Exemplary stick mass spectra ($m/z$) of laboratory generated particles as measured by ERICA-LAMS. Left: Cations, right: Anions. (a) NaCl, (b) AN, (c) gold spheres, (d) benz[a]anthracene (BaA).

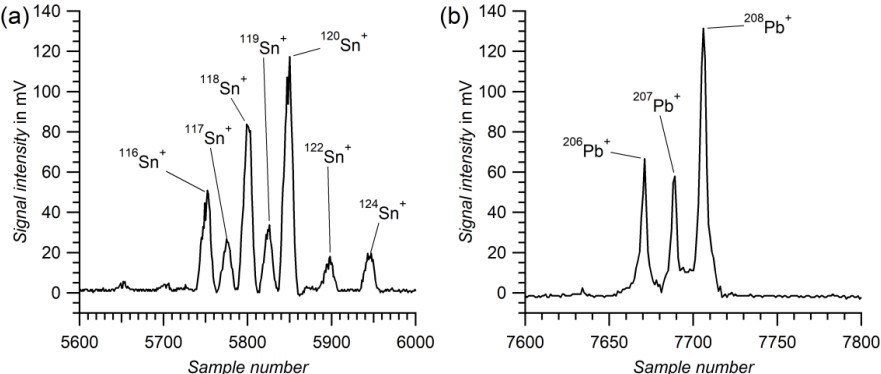

5   Fig. 13: Details of cation raw spectra (voltage output versus sample number of the digitizer, 1.6 ns per sample) of two ambient single particles at the airport of Kathmandu, Nepal. (a) Tin isotopic pattern ($d_{va}$ = 277 nm). (b) Lead isotopic pattern ($d_{va}$ = 311 nm).





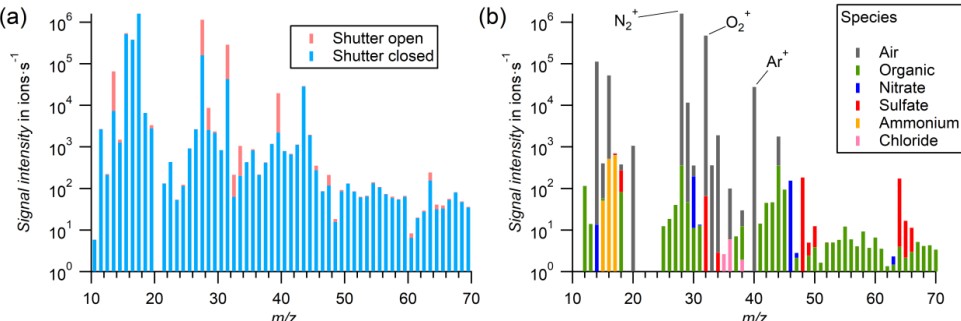

**Fig. 14: Example of an ambient aerosol average spectrum collected during the field campaign in Kathmandu, Nepal (averaged over the entire campaign period). (a) The integrated signal intensities at open (red) and closed (blue) shutter position. The "shutter closed" signal overlays the "shutter open" signal. (b) The calculated difference of open-closed from the left spectrum. Cumulative species**
5 **(air, organic, nitrate, sulfate, ammonium, and chloride) colored according to their fraction in the applied fragmentation table.**

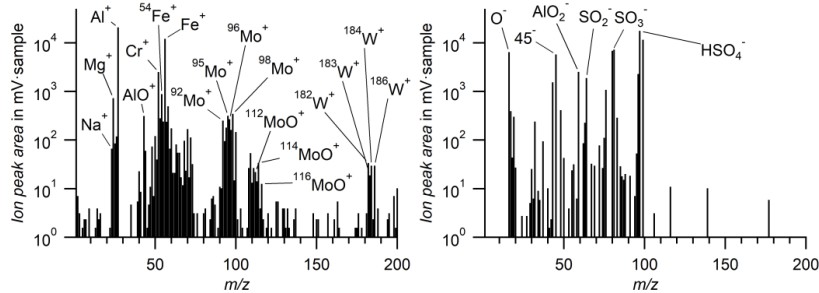

**Fig. 15: Exemplary single particle spectrum recorded during StratoClim 2017 demonstrates the feasibility of identifying metallic isotopes. Left: Cations, right: Anions. This heavy metal and sulfate-containing particle was measured at an altitude of 20402 m (29.07.2017, 06:09:34 UTC, $d_{va}$ = 602 nm). Note that the y-axis is logarithmic, in contrast to the spectra shown in Fig. 12.**





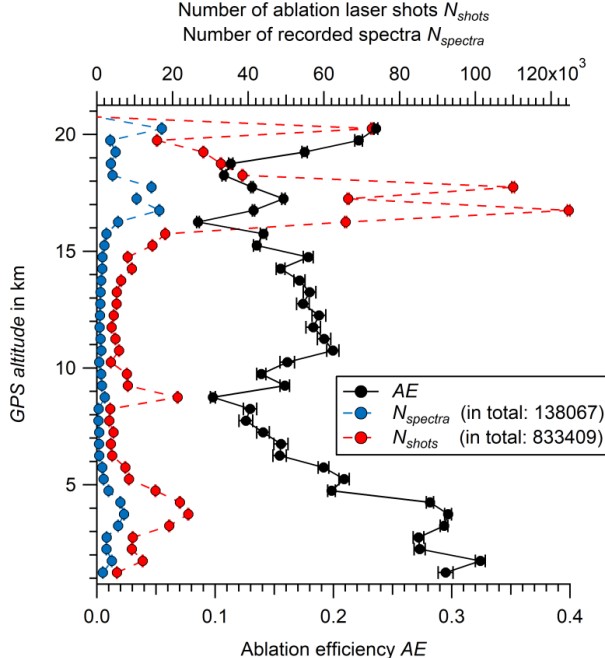

**Fig. 16: Vertical profile of the ablation efficiency $AE$ (black, bottom abscissa), the number of recorded spectra $N_{spectra}$ (blue, top abscissa), and number of ablation laser shots $N_{shots}$ (red, top abscissa) for the entire second aircraft campaign in 500 m bins. Uncertainties of $AE$, $N_{spectra}$, and $N_{shots}$ are based on counting statistics. The error bars are in some cases smaller than the symbol.**

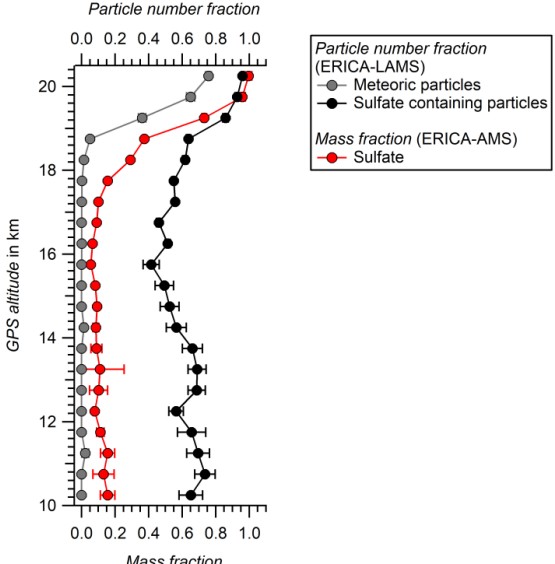

**Fig. 17: Vertical profile (flight on 04.08.2017) of the particle number fraction of meteoric material (gray) and sulfate-containing (black) single particles (ERICA-LAMS) and the mass fraction of sulfate (red; ERICA-AMS). The vertical resolution is in altitude bins of 500 m. The uncertainties of the particle number fraction are calculated from counting statistics. The uncertainty of the mass fraction is based on the background measurement and was propagated for the mass fraction. The error bars are in some cases smaller than the symbol.**





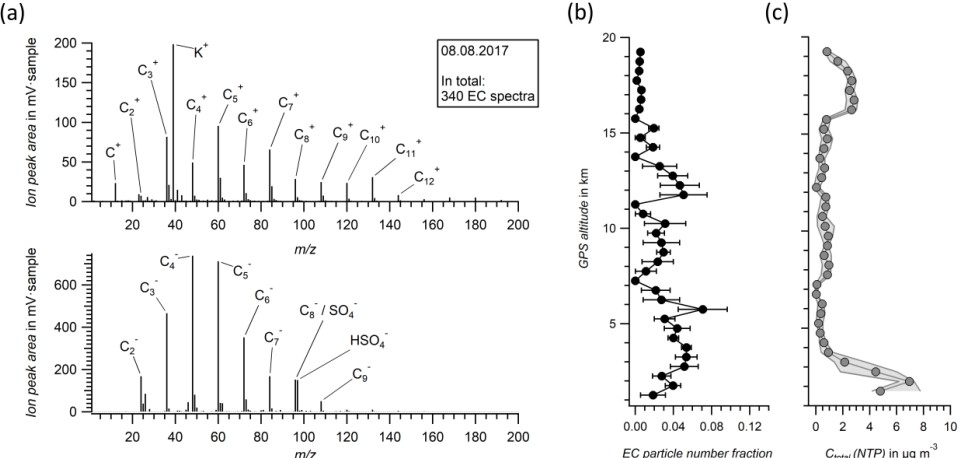

**Fig. 18: Data from the research flight on 08.08.2017 during StratoClim, Nepal. The vertical resolution is in altitude bins of 500 m. (a) The mean mass spectrum of 340 EC-containing single particles. (b) The vertical profile of the particle number fraction of EC-containing single particles (ERICA-LAMS). The uncertainty of the particle number fraction is calculated from counting statistics.**

**The error bars are in some cases smaller than the symbol. (c) The vertical profile of the median total mass concentration $C_{total}$ (NTP; ERICA-AMS). The interquartile ranges of the median total mass concentration $C_{total}$ is shaded in gray.**

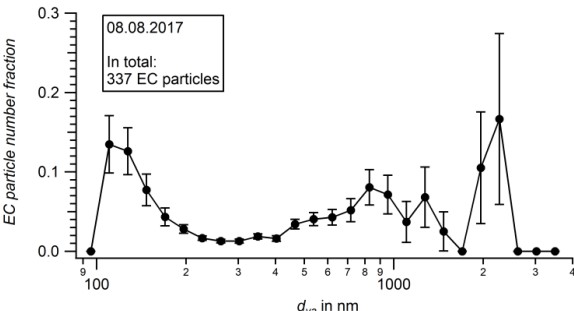

**Fig. 19: Particle number fraction of the EC-containing particle type as a function of particle size $d_{va}$ (logarithmic bin size) recorded during a research flight during the second aircraft field campaign of StratoClim on 08.08.2017, where 340 single particles were**
10 **identified as EC- containing particles. Only the spectra with size information within the calibrated size range were processed (in total: 337). Below a particle size of 100 nm and above 2400 nm, no EC-containing particles were observed. The uncertainties are calculated from counting statistics.**

**Table 1: Detection limits of the species measured by the ERICA-AMS determined with several methods. $DL_{stat}$ and $DL_{filter}$ measured under lab conditions, $DL_{spline}$ measured during StratoClim field campaign. The limits are given for one measurement cycle (10s) and are expected to reduce with longer averaging times $t$ proportionally to $1/\sqrt{t}$.**

| species | $DL_{stat}$ in µg m⁻³ | $DL_{filter}$ in µg m⁻³ | $DL_{spline}$ in µg m⁻³ |
|---|---|---|---|
| chloride | 0.13 | 0.24 | 0.090 |
| ammonium | 0.050 | 0.40 | 0.73 |
| nitrate | 0.11 | 0.12 | 0.12 |
| organic | 0.18 | 0.52 | 0.50 |
| sulfate | 0.0037 | 0.060 | 0.13 |