# Peer review of "Design, characterization, and first field deployment of a novel aircraft-based aerosol mass spectrometer combining the laser ablation and flash vaporization techniques"

_Atmospheric Measurement Techniques, 2021_

## Referee Comment (RC1)

**Review of "Design, characterization, and first field deployment of a novel aircraft-based aerosol mass spectrometer combining the laser ablation and flash vaporization techniques"**

**General comments**

In this study, the authors present a novel mass spectrometer ERICA (ERC Instrument for Chemical composition of Aerosols), which combines two ionization techniques, i.e., laser ablation and the flash vaporization with electron impact ionization. Given the complementary strengths of the techniques, ERICA allows for in-situ and real time measurements of size and chemical composition of the aerosol particles, provides qualitatively information of almost all the particulate components and the quantitative information of the non-refractory components. The authors have done comprehensive laboratory and ground-based field measurements to characterise this instrument and tried to demonstrate its improved chemical characterization capability. As shown in the manuscript, such a hybrid instrument with compact and light-weight design is good for aircraft measurement. This study would be quite useful for atmospheric science research, especially in the mass spectrometry community. However, the presentation is not very well structured and not clear enough in current version, which needs to be improved. In addition, the authors should do more literature research on single particle mass spectrometry (SPMS) and aerosol mass spectrometer (AMS) to make correct statements. Therefore, I recommend it to be published after major revisions.

**Major Comments:**

**1. Several confusion/wrong statements on these two complementary techniques need to be revised.**

1) Please note that the SPMS uses laser for desorption and ionization, while AMS uses vaporization followed by electron impact ionization. "vaporized" (P3L22) needs to change to "desorbed". Please distinguish these two ionization techniques in a clearer way throughout the manuscript. In addition, SPMS and AMS use different way to determine particle size $d_{va}$. The authors miscited some references in section 2.1, Page (P) 3 Line (L) 20. Please correct.

2) Limited repetition rate of ablation laser is only one of the reasons for the low detections, but not the main one. There are several other influencing factors on the low detection efficiency and detailed discussions on such topic. Please refer to and cite the corresponding SPMS publications, e.g., from the most related instrument ALABAMA, and revise accordingly, e.g., P2 L30-32 & P6 L13-14.

3) The authors should be very cautious when compare ERICA-LAMS with ERICA-AMS.

For example, in section 4 the authors compare the number fraction of sulfate containing particle with the mass fraction of sulfate and discuss the difference (P21 L23-30 & Fig. 17). However, the reasons for the difference are not convincing. Please reconsider the explanations.

In Fig. 17 the sum of the number fractions of meteoric and sulfate containing particles are larger than 1 at higher altitude. This is confusing and needs more explanation. Apparently, the methods to obtain these two particle types are not the same: the meteoric type is based on k-means clustering, while the sulfate containing particle type is very likely based on the maker peaks' intensities (please describe). Consider modifying Fig. 17 or add detailed descriptions in the figure caption.

The discussion on total mass concentration (measured by ERICA-AMS) and EC-containing particles (ERICA-LAMS) cannot come to the conclusion that "the sampled aerosol is well mixed within the particle boundary layer and in the free troposphere", also cannot show the complementary strength. Please reshape the statements.

**2. Presentation quality needs to be improved.**

1) Citation formats: Please pay attention to the formats between Author et al. (year) and (Author 1 et al., year; Author 2 et al., year; Author 3 et al., year; ...) and use them properly. Please revise the citation format throughout the manuscript and keep consistency.

E.g., P1 L35 "(See for example Fuzzi et al. (2015))" should be changed to (Fuzzi et al., 2015); P2 L10: Change "(e.g., in Froyd et al. (2019))" to (Froyd et al., 2019).

2) Section 2 Instrument description: I would suggest refining the descriptions of ERICA-LAMS and EIRCA-AMS modules, since most of them have been well described in SMPS and AMS papers. Please emphasize the difference, e.g., the shutter unit (SU) needs more descriptions. Consider combining 2.3 and 2.4. Pleas simplify the headers.

3) Section 3 Instrument characterization: This section is very important and with comprehensive information, but the key points are buried. It would be very hard for the readers to follow since the LAMS and AMS information is mixed in an unclear way. I would highly suggest rewriting this section by considering the following points.

Please separate the characterization of LAMS and AMS first and then discuss complementary features, and also revise the corresponding figures. Besides, move some detailed descriptions, regarding e.g., calibration (e.g., particle size cal in LAMS; AMS IE and RIE cal), instrument alignment (e.g., ADL position scan), in the supporting information, since they are very well described in other publications or user's manual. An example of restructuring: 3.1 Particle beam characterization; 3.2 ERICA-LAMS characterization (Laser beam; Optical detection efficiency; Hit rate; LAMS mass spectra); 3.3 EIRCA-AMS characterization (Collection efficiency; Detection limit; AMS mass spectra; Mass concentration), and 3.4 Overall performance comparison (sensitivity, size, spectra, etc).

Please keep the terminology same as the ones commonly used in SPMS and AMS communities, respectively, e.g., use "hit rate" instead of "ablation efficiency"; use "collection efficiency" instead of "detection efficiency".

4) Section 4: The authors only describe the sizes of EC-containing particles in the last paragraph in this section, which is not strong enough. Please give the information of chemical resolved size distributions obtained by both ERICA-LAMS and ERICA-AMS, and add more discussion accordingly.

5) Figures:

Consider moving some to the SI, e.g., Fig. 3 and 4, and combing some, e.g., Fig 7 and 8, Fig 8 and 9.

In Fig. 5, 6, and 7, the solid squares, diamonds and circles with the same colour are not easy to distinguish. Please modify them in a clearer way.

Error bars: Since this sentence "The error bars are in some cases smaller than the symbol" is shown in most of the figure captions (Fig. 3, 5, 6, 7, 8, 9, 10, 11, 16, 17, and 18), I suggest that put the corresponding values in SI. The following question is that in the laboratory how many repeated experiments have been done to generate one data point?

Mass spectra: The x and y scales, as well as the axis labels, are inconsistent among all the spectra. E.g., for y axis, in Fig. 12 it is "signal intensity in a.u." in linear mode, while in Fig. 15 and 18, it is "ion peak area in mV. sample" in log mode. Please try to keep consistency. Please normalize the spectra to the total ion intensity and keep the same scales (both x and y) for consistency.

**3**. The advantages of this hybrid instrument are not very well demonstrated, not only due to the poor manuscript structure, but also lacking discussion on complementary results. Please try to improve. Besides, in addition to the compact size, are there any other big advantages of using such a hybrid instrument compared to deploying SPMS and AMS instruments in parallel? Please state the differences.

4. For the current configuration of the LAMS module, it is hard to believe that PSL particles with smaller size < 200 nm can be detected. Several statements on the PSL 80 nm and 108 nm with the corresponding data shown in the Figures 3, 5, 6, 7, 9, 10, and 11 are not valid. Please consider modifying or removing accordingly.

**Minor Comments:**

**P1L23-25:** Please change 3170 nm to 3.17 μm or change 3.5 μm to 3500 nm to keep consistency and revise throughout the manuscript.

**P2 L10:** Change "(e.g., in Froyd et al. (2019))" to (Froyd et al., 2019; Author 2 et al., year...), and please add more corresponding references. Lots of quantification work has been done by using ATOFMS and other reference instruments like OPC, AMS, and so on.

**P3L25:** Please cite the corresponding publications.

**P3L32:** Please use 3.17 μm to keep consistency.

**P3L32-33:** It would be more helpful to mention the transmission efficiency of the ADL instead.

**P3L36-38:** Please describe the difference between shutter and chopper.

**P4L25:** The full name of LAAPTOF should be "Laser Ablation Aerosol Particle Time-Of-Fight mass spectrometer" rather than "…spectrometry".

**P4L26:** Please change the dot in "5·10$^2$ cm³ s$^{-1}$ " to multiplication symbol "5×10$^2$ cm³ s$^{-1}$ " and revise the others throughout the manuscript.

**P15L28:** Fig 10 should be Fig 11.

**P16L11:** Please give the reason for choosing these peaks.

**P16L34:** Please assign the peak at m/z 228.

**P19L27:** The left half of the bracket is missing.

**P20L28-30:** The reader would expect the following focus on meteoric and EC containing types rather than particulate sulfate, which is a compound. Please reshape this sentence to make the transition smoothly.

**P20L31:** Incorrect statement. Please revise.

**P20L35:** Please clarify that when only considering the non-refractory species, the sulfate mass fraction is 1 at 20 km.

**P21L32-35:** Please add references to support the assumption.

**Fig.1:** Please add TMP 1 to 4 in the figure or point out their positions. Please add the distances between LD1, LD2, ablation spot, shutter unit, vaporizer, etc.

**Fig.2:** Consider rescale some sizes/distances. E.g., the distance between convex lens and the quartz window (10 mm) should be twice the size of the ablation laser beam (5 mm). This can be easily done.

**Fig.3 caption:** (b) is not clear, please reshape the sentence; (3) is confusing, please rewrite.

**Fig.6:** Please use the same scales for the left and right Y-axes.

**Fig.12:** Please clarify that whether the stick spectra are for individual particles or the averaged ones? If averaged, please give the total number of the spectra for averaging. Please normalize the spectra, e.g., to the total ion intensity, and keep the same scales (both x and y) for consistency. E.g., m/z can be fixed from 0 up to 250 amu. for each spectrum. This can be applied to the special case of gold particles too, only need to illustrate the $Au_2^+$ additionally.

**Fig.9 and 10**: Please combine them. Please remove the AN measured by AMS and put it in a separate figure.

**Fig.13:** Please give the definition of the "sample number".

**Fig.14:** (a) It is hard to see the signal difference between shutter open and closed. Please consider a better way to demonstrate. (b) The calculated difference does not agree with the left spectrum. E.g., the bars are apparently not at the same positions between two plots; the most intensive peak m/z $28^+$ (labelled $N_2^+$) is even a bit higher than the corresponding one in (a), as well as the m/z 32+, 40+, etc. The labels of $N_2$ and $O_2$ are confusing, since the peaks also contain the organic and sulfate fragments, respectively. Please modify them with a clearer way.

---

## Referee Comment (RC2)

**Review of *Design, characterization, and first field deployment of a novel aircraft-based aerosol mass spectrometer combining the laser ablation and flash vaporization techniques* by Hünig et al.**

Anonymous Reviewer

October 2021

**1  Summary**

In this work, Hünig et al. describe, for the first time, the design and characterization of ERICA. At the time of this review, ERICA is a unique instrument, but it does combine two well-known methods: (1) single-particle mass spectrometry using laser ablation to (partially) vaporize single particles and ionize their constituents, and (2) an AMS-style instrument that flash vaporizes the non-refractory component of aerosol using a hot tungsten filament and creates ions using electron impact. Method 1 will be referred to as ERICA-LAMS, and Method 2 will be referred to as ERICA-AMS, per the authors' designation. ERICA LAMS uses two time-of-flight mass spectrometers to analyze the positive and negative ions from a single particle; ERICA-AMS uses a compact-time-of-flight mass spectrometer to analyze positive ions. Both ERICA-LAMS and ERICA-AMS share a common aerosol focusing inlet (AFI), which is pressure-controlled and has been written about in a separate publications (Molleker et al., 2020). After exiting the AFI, the particles are sized by measuring the particle time-of-flight between two particle detection units (PDU1 and PDU2). Optical sizing was experimentally achieved for PSL between 80 nm and 5.145 $\mu m$ Particles detection by PDU2 triggers a 266-nm quadrupled Nd:YAG ablation laser to fire (max repetition rate 8 hz$^{-1}$, $\sim$ 4 mJ/pulse). Particles that are not detected by PDU2 or are missed by the ablation laser are collected $\sim$55 cm from the exit the AFI, and $\sim$30.1 cm downstream from the ablation laser spot.

The authors give much attention in the paper to the particle beam diameter and the effective laser / vaporizer diameters. All are fitted parameters, which are fitted to a convolution of two functions–a top hat function for the effective laser / vaporizer width and a 2D Gaussian function for the particle beam width. In ERICA-LAMS, the particle beam width ranges from $\sim$30-40 $\mu m$ for 335 nm

AN particles to $\sim$100-200 $\mu m$ for all particles >400 nm to >500 $\mu m$ for 103 nm PSL. For particles $\geq$ 208 nm, the particle beam diameters are smaller than the effective laser diameters in PDU1 and PDU2. For ERICA-AMS, particles with diameters > 91 nm have particle beams smaller than the effective diameter of the vaporizer, which, unlike the effective laser diameters, is similar to the physical dimensions of the vaporizer (3.8 mm).

The most userful meaasured parameters in the paper are the detection efficiency (DE) and the abation efficiency (AE). The former measures the number of particles detected by the PDUs compared to a separate measurement of particles counts by a CPC or OPC; the latter is the number of particles that has mass spectra divided by the number of particles that trigger PDU2. The DE analysis shows that, under ideal conditions (*e.g.,* idea beam position, which changes as a function of size), the DE for PSL is above 0.6 for particles $\geq$ 208 nm; however, for real-world particles the DE is generally lower across all sizes measured. Finally, the AE for real-world urban particles was presented. The AE has a maximum value of 0.52 @ 218 nm; however, the authors also found that the AE is a steep function of size, and hovers around 10-20% for particles below $\sim$ 200 nm and above $\sim$ 300 nm.

The paper finished with some example laboratory particles, as well as some example particles and science from the first aircraft deployment.

Overall, this paper is very well written and very well thought out. The scope of the paper also fits very well within the scope Aerosol Measurement Techniques. At the time of this review, ERICA is a completely unique instrument; thus, a detailed description and characterization paper is well-timed and necessary for future publications. This reviewer only has a few comments, which are outlined below.

**2 General Comments**

1. Section 3.1–It is unclear to this reviewer if the "razor blade" is integrated into the system like the "knife edge" in the PALMS instrument. If so, it is also unclear if ERICA uses the knife-edge to re-position the papers during flight, where they might have moved due to vibrations from the aircraft.

2. Section 3.3: It is unclear to the reviewer if the "effective laser radius" being much larger than the physical dimensions of the laser is supported by Mie theory (as was done for 108 nm particles). Is this true? Is this akin to a "scattering cross section?" If so, the authors should support that with some calculations in the supplemental. Otherwise, the authors risk comparing the physical beam diameters to a laser diameter that is fitted (as opposed to measured) and perhaps physically unrealistic.

3. Fig.10: I am slightly confused how it is possible that PDU2 can have higher values than PDU1. Can the authors comment on this?

4. Section 3.4: Because ERICA has both an optical DE for PDU2 and an AE,

it would be helpful for the authors to explicitly show a DE for ablation. This wold help the readers understand biases in ERICA number fractions etc.

5. Section 4: Towards the end of the paper, the authors compare ERICA-LAMS data to ERICA-AMS data on the same plot. This caused this reviewer to of biases between the measurements that should be addressed before having a combined interpretation of the LAMS and AMS results. The major bias, as understood by this reviewer from the figures in this paper, is that the "number fraction" will be highly dependent on the size and composition of the particles present. These should somehow be weighted accordingly–by internal DE curves or by normalizing to external quantitative measurements. No discussion of this correction is present in the current manuscript–this reviewer strongly suggests that the authors address that in this manuscript, as it will affect all future work from this instrument.

**3   Minor Comments**

1. P2L40: Since each paper should stand on its own–a brief description of the Dragoneas paper should be described here. That way the reader does not have to download a separate paper to fully understand your methods.

2. P3L24: "A large fraction" here is largely meaningless without some general numbers or statistics.

3. P3L34: Is the lens and geometry in ERICA the same as the lens in XU et al.?

4. P8L29: At what aerosol concentration (number and volume / mass), does ERICA-LAMS affect $\sim 30\%$ of the particles? This should be spelled out for the reader? I assume it could affect some areas of the Upper Troposphere.

5. P10L18: That the aerodynamic diameters of AN are similar to PSL suggest that they are spherical and of similar density. This not entirely surprising because AN is notoriously difficult to effloresce; however, the authors state that effective laser radius for AN do not match PSL because the AN are non-spherical. Can you reconcile these two statements?

6. P13L32: I'm not sure that "$w_{0,dia}$" is not the most meaningful measurement for overlap. Unlike the signal in PDU1 and PDU2, the intensity of the ablation laser will be essential to the interpretation of the mass spectra–especially for large or coated particles. Thus, a measure of the overlap between the particle beams and where the ablation laser is sufficiently powerful is indeed important to report.

7. P13L35: I don't think I saw any evidence that the 80 nm and 5145 nm particles were ablated and detected by the MCP. Is this true? If so,

perhaps a $AE_{max}$ could be shown for PSL particles much like $DE_{max}$ was?

8. P18L21: This reviewer is not an AMS expert–but, as written, it sounds like all RIEs are relative to the nitrate IE. So, why does nitrate have an RIE of 1.1?

9. P19L9: As written, it is unclear if it is most desirous to have a "small air beam sample" over no air beam sample.

10. P20L34: Can an estimate of the UT and LT altitude / altitude ranges be added to Fig. 17?

11. P22L25: It seems to this reviewer that different removal rates of EC and $C_{total}$ suggests that the particles are not well mixed–because they would then be removed at the same rates.

12. P22L36: Are these EC particles from coagulation? They seem quite high to be primary particles.

13. P23L5: The authors often differentiate the EREICA-AMS data by say "the non-refractory components." This is misleading because ERICA-LAMS also measure the non-refractory components.

14. Figures: It is really hard, especially with the errors bars to differentiate the filled circles from the filled squares. Perhaps switch to filled and open squares?

15. Figure 10: Using 50% of the max is a bit strange in this plot–it results in PDU1 having larger D50s than PDU2, which is counter-intuitive given that PDU2 has better detection efficiencies.

16. Figure 11: Can you make the right side of this plot a log-scale (and also possibly the left?). It is hard to see if you're getting spectra for any particles below $\sim 120$ nm or above $\sim 1$ $\mu m$.

17. Figure 12: Why do you have a large $Na^+$ peak in your PAH spectra? Is your mass scale possibly off?

**4   Technical Comments**

- P1l11: What does "ERC" stand for?

- P1L15: Perhaps "*The same* aerosol sample can be sampled with both methods *simultaneously*?

- P1L20,25,26: The acronyms ADL, B-ToF-MS an C-ToF-MS are defined here, but are not used again in the abstract. The abstract should generally stand alone, and therefore these acronyms can be omitted, but need to be defined at their first use in the main section of the paper.

- P1L36: You probably can delete the comma after "anthropogenic-".

- P2L19: Perhaps use "e.g.," instead of "beside others by."

- P13L33: This reviewer is not sure "However" is the right word here–this statement does not seem to be related to the previous sentence.

- P16L10: It is hard to understand ion peak threshold as currently described. It might be easier to understand by splitting this statement up into two or more sentences.

- P19L19: You can probably delete "especially" in this line.

- P22L25: The statement "within the limitations of the applied method" is parenthetical and needs commas around it.

**References**

Molleker, S., Helleis, F., Klimach, T., Appel, O., Clemen, H.-C., Dragoneas, A., Gurk, C., Hünig, A., Köllner, F., Rubach, F., Schulz, C., Schneider, J., and Borrmann, S.: Application of an O-ring pinch device as a constant-pressure inlet (CPI) for airborne sampling, Atmospheric Measurement Techniques, 13, 3651–3660, https://doi.org/10.5194/amt-13-3651-2020, URL https://amt.copernicus.org/articles/13/3651/2020/, 2020.

---

## Referee Comment (RC3)

The authors present the design and development of a mass spectrometry system for comprehensive measurement of aerosol composition, in which two commonly used techniques, single particle mass spectrometry (SPMS) and aerosol mass spectrometry (AMS) are combined in a single tandem instrument. The manuscript represents a substantial body of work that required considerable expertise in instrument design including differential pumped vacuum systems, optical particle detection and time-of-flight mass spectrometry (TOFMS). A substantial amount of data is presented to evaluate the instrument design. The subject matter is very suitable for this journal but some important issues need to be addressed in the content if this manuscript is to be used as an instrument characterisation reference for future publications.

**Major Comments**

Both instrument use TOFMS as an analyser. This should be introduced and the benefits explained. They both also use aerodynamic lens inlet. The main difference is with the ionisation techniques employed to achieve the desired measurement. The pros and cons to each technique and the consequences on the data should be developed in the introduction. Both techniques are hard ionisation that causes intense fragmentation that has to be dealt with in the data analysis. In the case of laser desorption ionisation (SPMS), this renders the measurements inherently non-quantitative for molecular ion species. The thermal desorption ionisation method used in the AMS method is only quantitative with careful calibration. The authors present some details of the mass calibration in terms or the relative ionisation efficiencies (RIE) of nitrate, sulphate, and ammonium using the same method used for the Aerodyne AMS family of instruments. This is where my first major concern with the work arises.

In various places throughout the document the authors state the ERICA-AMS is 'similar' in design to the Aerodyne AMS, but the similarity is not described nor are the differences. In fact, no detailed description of the vaporiser, ioniser and ion extraction optics is given. The Thermal Desorption ionisation technique (TDI) is not well understood and Quantitative nature of the Aerodyne AMS instrument is underpinned by a large body of publications and method development (See Jimenez 2016 and references therein). If the authors wish to convey these characteristics onto their instrument, they need demonstrate equivalence in the design, particularly regarding the geometry of the ionisation source and the incident particle beam.

This leads to the second point of major concern with this manuscript regarding the measurement/calculation the particle beam width. The method description is extremely difficult to follow in the current version of the document and it is impossible to get any sense of the error in the calculation. This needs to be addressed. The authors use a method in which the particle beam is tracked across optical detection system which is kept static, in a very similar method to that presented in Marsden 2016 (not cited here) with the LAAPTOF single particle mass spectrometer, an instrument with many common features to the ERICA LAMS. The results are quite different regarding the ratio of particle beam and detection laser beam width compared to the LAAPTOF. This may be due to a superior quality aerodynamic lens, but the result should be discussed with respect to LAAPTOF and other instrument design as this is an important factor in instrument design.

Finally, I have concerns about the dynamic range of the ion detection system in ERICA LAMS. The A/D has only 8bits if vertical dynamic range which equates to 3 orders of magnitude within spectrum signal. This is insufficient in the reviewers experience and will either produce excessive saturation of intense ion signals or the complete loss of minor signals depending on the gain setting. Can the authors comment on this in section 3.5.2?

**Minor Comments**

- Take care to make accurate definitions upfront in the introduction, and then stick to those definition throughout the document.
- Please check the correct use of commas throughout the document and avoid excessive paragraph length.
- The writing style changes part way through the document which is rather odd.

**Introduction**

Page 1  ln 35    Chemical composition measurements can provide…

   Ln39    Comma after 'in situ' not required

Page 2, Ln 1    Define the 'pulsed laser technique' as 'single particle mass spectrometry (SPMS)'

Page2, Line 5    the correct term is 'Thermal Desorption (TD)' and should be used throughout the document.

Page2, Ln8    This sentence is a little muddled. Maybe replace 'previous' with 'former'?

Page2, Ln10    Froyd et al. (2019) demonstrates a method for quantifying particle classes, not absolute mass concentrations of specific ions. There is an important distinction.

Page2, Ln 11    Consider starting a new paragraph

Page2, Ln 30    Perhaps introduce the term 'tandem measurement'

Page2, Ln31    Replace 'repetition rate' with the term 'temporal resolution'

Page2, Ln37    'Tandem Instrument'?

**Instrument Description**

- I brief principal of operation required before getting into the detail. Both techniques are sampling to same particle beam with the ERICA AMS at the end of the particle path. The LDI is requires optical detection to size particles and trigger the pulsed laser part way along the path.

Page3, Ln12    More effort should be made to describe Fig1.

Page3, Ln12.    Define LAMS and AMS in the introduction or consider changing to Laser desorption ionisation (LDI) and Thermal desorption Ionisation (TDI) therefor highlight the actual distinction between the two techniques.

Page3, Ln14    Why is a constant pressure inlet required? Should this have already been introduced as part of the challenges of aircraft measurement?

Page 3, Ln23    The term 'ion extraction'  instead of acceleration would be more appropriate

Page3, Ln25    Some particles are partially vaporised. What happens to particle fragment and partly ablated material?

| | |
|---|---|
| Page3, Ln28 | Un-ablated particles do not pass through the B-TOF-MS section because they are not extracted. |
| Page3, Ln31 | use 'extracted' instead of 'injected. |
| Page3, Ln31 | C-TOF-MS has not been properly introduced. |
| Page3, Ln31, | You have to be more specific than 'Detectable particle size' as that would appear to conflict the next sentence. Do you mean you get composition measurement from that size range? |
| Page3, Ln33 | Xu 2017 describes the ACSM – please state that. Is it valid to assume the detectable particle size range is the same as the ACSM? This requires some discussion. |
| Page 3, Ln39 | Consider putting the final paragraph of this section as part of the introduction. |
| Page4, Ln 30 | Are the vacuum pressures measured or calculated? A schematic of the vacuum system would be helpful. |
| Page5, Ln15 | How is the vacuum seal achieved on a movable assembly? |
| Page 5, Ln20 | How do you know that the system collects 75% of the scattered light. Has this been modelled or measured? |
| Page6, Ln10 | What shape beam profile is produced by the pulsed laser system. Is there variation in the power density with respect to position on the particle beam axis? |
| Page6, Ln29 | 8bits the effective dynamic range including the noise? This equates to around 3 orders of magnitude. |
| Page6, Ln30 | The positive and negative ion signals are measured by separate detection systems. Whilst having different gain on each channel is beneficial, it does not actually increase the dynamic range of the A/D, nor the dynamic range within the spectra. This is misleading. |
| Section 2.5 | The writing style changes to prose, which is rather odd. |
| Page 8, Ln1 | Replace 'serial configuration' with 'tandem configuration' |
| Section 2.6 | Is the data for 5% reduction in particle mass on the AMS with LAMS switched on actually presented in this paper? Where? |
| Section 3.1 | The detection laser beam waist (250um) is much smaller than particle beam, but much larger that the particle diameters. Particles can encounter very different laser fluence depending on their trajectory through the Gaussian profile, therefore the effective irradiance encountered cannot be calculated by diciding the laser power by the beam area. See Marsden et al 2018. |

**References**

Jose L. Jimenez, Manjula R. Canagaratna, Frank Drewnick, James D. Allan, M. Rami Alfarra, Ann M. Middlebrook, Jay G. Slowik, Qi Zhang, Hugh Coe, John T. Jayne & Douglas R. Worsnop (2016) Comment on "The effects of molecular weight and thermal decomposition on the sensitivity of a thermal desorption aerosol mass spectrometer", Aerosol Science and Technology, 50:9, i-xv, DOI: 10.1080/02786826.2016.1205728

Marsden, N., Flynn, M. J., Taylor, J. W., Allan, J. D., and Coe, H.: Evaluating the influence of laser wavelength and detection stage geometry on optical detection efficiency in a single-particle mass spectrometer, Atmos. Meas. Tech., 9, 6051–6068, https://doi.org/10.5194/amt-9-6051-2016, 2016.

---

## Author Comment (AC1)

AMT-2021-271

**Design, characterization, and first field deployment of a novel aircraft-based aerosol mass spectrometer combining the laser ablation and flash vaporization techniques**

Hünig et al.

**Replies to the comments by Anonymous Referee #1**

General Reply:

We very gratefully acknowledge the detailed, diligent and careful review provided by Referee #1. This review significantly helped us to improve the manuscript.

The reviewer comments are written in this font style and color.

Our answers are written in this font style and color.

Changes to the revised version of the manuscript are printed in red.

**Review of "Design, characterization, and first field deployment of a novel aircraft-based aerosol mass spectrometer combining the laser ablation and flash vaporization techniques"**

**General comments**

In this study, the authors present a novel mass spectrometer ERICA (ERC Instrument for Chemical composition of Aerosols), which combines two ionization techniques, i.e., laser ablation and the flash vaporization with electron impact ionization. Given the complementary strengths of the techniques, ERICA allows for in-situ and real time measurements of size and chemical composition of the aerosol particles, provides qualitatively information of almost all the particulate components and the quantitative information of the non-refractory components. The authors have done comprehensive laboratory and ground-based field measurements to characterise this instrument and tried to demonstrate its improved chemical characterization capability. As shown in the manuscript, such a hybrid instrument with compact and light-weight design is good for aircraft measurement. This study would be quite useful for atmospheric science research, especially in the mass spectrometry community. However, the presentation is not very well structured and not clear enough in current version, which needs to be improved. In addition, the authors should do more literature research on single particle mass spectrometry (SPMS) and aerosol mass spectrometer (AMS) to make correct statements. Therefore, I recommend it to be published after major revisions.

**Major Comments:**

**1. Several confusion/wrong statements on these two complementary techniques need to be revised.**

1) Please note that the SPMS uses laser for desorption and ionization, while AMS uses vaporization followed by electron impact ionization. "vaporized" (P3L22) needs to change to "desorbed". Please distinguish these two ionization techniques in a clearer way throughout the manuscript. In addition, SPMS and AMS use different way to determine particle size $d_{va}$. The authors miscited some references in section 2.1, Page (P) 3 Line (L) 20. Please correct.

(Numbers of pages, lines and sections refer to the submitted manuscript for review)

– "vaporized" (P3L22) was corrected to "desorbed".

– Furthermore, the termini „LDI" (Laser Desorption and Ionization) for SPMS and „ TD-EI " (Thermal Desorption and Electron impact Ionization) for AMS were implemented to distinguish both methods in a clearer way.

– Correction of citation:
The references Jimenez et al. (2003a), Jimenez et al. (2003b), and DeCarlo et al. (2004) refer to the definition of the vacuum aerodynamic diameter $d_{va}$. The reference Hinds (1999) was removed. The reference for sizing by means of a calibration in LAMS, Brands et al. (2011), was added. In total:

"The time elapsing between the two light scattering signals is used to derive its vacuum aerodynamic diameter $d_{va}$ (Hinds (1999), Jimenez et al. (2003b), Jimenez et al. (2003a), and DeCarlo et al. (2004)) by involving a calibration (see Sect. 3.2) and to determine the point in time the particle reaches the ablation spot of the ERICA-LAMS.

was changed to

"The time elapsing between the two light scattering signals is used to derive the particles vacuum aerodynamic diameter $d_{va}$ (for definition see: Jimenez et al., 2003a, b; DeCarlo et al., 2004) by involving a calibration (Brands et al., 2011)"

2) Limited repetition rate of ablation laser is only one of the reasons for the low detections, but not the main one. There are several other influencing factors on the low detection efficiency and detailed discussions on such topic. Please refer to and cite the corresponding SPMS publications, e.g., from the most related instrument ALABAMA, and revise accordingly, e.g., P2 L30-32 & P6 L13-14.

(Numbers of pages and lines refer to the submitted manuscript for review)

P2 L30-32:

"Also, since the repetition rate of high-power UV ablation lasers limits the number of particle detections per second, the addition of a thermal vaporization and electron impact ionization unit largely enhances the data yield for the particle analysis."

was changed to (Numbers of sections refer to the revised manuscript):

„ Since, beside other reasons (see Sect. 2.3), the temporal resolution of the ablation laser, limits the number of particles detected (e.g., Su et al., 2004). The addition of a TD-EI unit largely enhances the data yield for the particle analysis by complementary information."

P6 L13-14:

"This maximum repetition rate imposes a limit to the number of particles analyzed per time unit, which affects the spatial resolution for measurements from a fast flying aircraft."

 was changed to:

"Beside other reasons, the maximum repetition rate of the ablation laser, particle losses in the ADL, the particle beam divergence, particle beam and laser beam alignment, the focusing width of the particle beam, the ionization efficiency of the particle components, and the sensitivity of the optical detection units limit the number of particles analyzed (Su et al., 2004; Zelenyuk and Imre, 2005; Brands et al., 2011; Marsden et al., 2016; Clemen et al., 2020), which affects the spatial resolution for measurements from a fast flying aircraft."

3) The authors should be very cautious when compare ERICA-LAMS with ERICA-AMS.

For example, in section 4 the authors compare the number fraction of sulfate containing particle with the mass fraction of sulfate and discuss the difference (P21 L23-30 & Fig. 17). However, the reasons for the difference are not convincing. Please reconsider the explanations.

(Numbers of pages, lines, sections, and figures refer to the manuscript submitted for review.)

Section 4 is not intended to be a comparison to highlight the differences of the ERICA-LAMS and the ERICA-AMS. Here, the possibility of obtaining complementary information and that this information can be merged is demonstrated. Therefore, not the differences are discussed here. In order to prevent the reader's expectation of a discussion on the differences, Fig. 17 was separated into 3 panels. See also our reply to RC2.

P20L37: To explain the high sulfate mass fraction value of 1 in 20 km altitude, following sentence was added: " Since no other species, such as nitrate or organics, were observed by the ERICA-AMS in significant amounts at  this altitude, the convective and radiatively driven vertical transport within the Asian Monsoon Anticyclone (AMA; Ploeger et al., 2015) does not play as much of a role here anymore, as further below."

P21L24: We revised the following text passage and removed the misleading statement about the internal mixing state:

"The results can also be used to show that the aerosol composition between 10 km to 17 km differs from the aerosol composition above 17 km. For this, the mass fraction of sulfate (ERICA-AMS) and the number fraction of sulfate-containing single particle spectra (ERICA-LAMS) were examined. Below 17 km, the number fraction of sulfate-containing single particle spectra is stable around 0.6 and the mass fraction of the sulfate less than 0.2. This could be indicative for an internal mixing state of the measured aerosol particles, where the sulfate species within the single particles is assumed as predominantly refractory compound, since the mass fraction of the sulfate species is low compared to the number fraction

of sulfate-containing particles. The reason is that the ERICA-AMS only can measure non-refractory substances. Above 17 km, the composition is more complex. With increasing altitude, the sulfate mass fraction and the particle number fraction of sulfate-containing single particles increase up to 1. The change in mass fraction is strong compared to the number fraction of sulfate-containing single particles. Therefore, it can be assumed that the non-refractory content increases. Since the ERICA-LAMS is not able to detect pure (non-refractory) sulfuric acid, no distinct determination of the mixing state can be obtained. Here, an internal or an external mixing state but also a combination of both states can be present. In a conceivable internal mixing state, the non-refractory sulfuric acid has deposited on a particulate core, generating a coated particle or the sulfuric acid acts as a condensation nucleus for other substances. Additional pure sulfuric acid particles lead to an external mixing state."

Was changed to:

"The results can also be used to show that the aerosol composition and mixing state between 10 km to 17 km differ from those above 17 km. For this, the mass fraction of sulfate (ERICA-AMS) and the number fraction of sulfate-containing single particle spectra (ERICA-LAMS) were examined (Fig. 15). Below 17 km, the number fraction of sulfate-containing single particle spectra is stable around 0.6 and the mass fraction of sulfate in the non-refractory aerosol is less than 0.2. This indicates that many particles contain sulfate, but typically only in a small mass fraction (about 1/3 on average), because they are internally mixed with nitrate and organics. Above 17 km, with increasing altitude, the sulfate mass fraction and the particle number fraction of sulfate-containing single particles both increase up to 1. The observed change in the mass fraction is stronger, compared to the increase in the number fraction of sulfate-containing single particles. Since the two measurement methods provide not only different views on the aerosol, but also have different limitations, this observation must be interpreted with care. A possible interpretation for the increasing sulfate mass fraction could be that within the internally mixed aerosol of particles containing a refractory core, e.g. of meteoric dust, and a sulfuric acid coating (Murphy et al., 2014), the coating grows as a consequence of further condensation. However, since the ERICA-LAMS is not capable of measuring pure sulfuric acid particles (Murphy, 2007), it is also possible that partial external mixing of the internally mixed particles with sulfuric acid particles causes this observation."

In Fig. 17 the sum of the number fractions of meteoric and sulfate containing particles are larger than 1 at higher altitude. This is confusing and needs more explanation. Apparently, the methods to obtain these two particle types are not the same: the meteoric type is based on k-means clustering, while the sulfate containing particle type is very likely based on the maker peaks' intensities (please describe). Consider modifying Fig. 17 or add detailed descriptions in the figure caption.

(Numbers of figures refer to the manuscript submitted for review.)

As the reviewer noticed, two different methods are used to determine the sulfate-containing (marker method) and the meteoric material -containing particle type (k-means). Both methods are briefly explained in the text. Since basically all "meteoric" particles are included in the "sulfate-containing" particles, the "meteoric" particles represent a subset of the sulfate-containing particles. Therefore, a summation of both particle number fractions is not meaningful. For better understanding and to avoid misinterpretation, Fig. 17 was divided into 3 panels and the description of the sulfate-containing single particles (measured by the ERICA-LAMS) was placed before the description of the mass fraction (measured by ERICA-AMS).

"To identify the sulfate-containing single particle spectra (ERICA-LAMS), the data set of the research flight of 04.08.2017 was filtered for single particle spectra that contained sulfate marker signals at $m/z$ -96 ($SO_4^-$) or $m/z$ -97 ($HSO_4^-$) or both markers."

Was changed to:

"To identify the sulfate-containing particle type, the ERICA-LAMS data set was filtered for single particle spectra that contained sulfate marker signals at $m/z$ -96 ($SO_4^-$) or $m/z$ -97 ($HSO_4^-$) or both markers. Since these sulfate marker signals are also found in the meteoric material containing particle spectra, by this approach, the "meteoric material containing particle type is a subtype of the sulfate-containing particle type."

The discussion on total mass concentration (measured by ERICA-AMS) and EC-containing particles (ERICA-LAMS) cannot come to the conclusion that "the sampled aerosol is well mixed within the particle boundary layer and in the free troposphere", also cannot show the complementary strength. Please reshape the statements.

The paragraph was revised (see also reply to RC2)

"This indicates within the limitations of the applied methods that the composition of the sampled aerosol is well mixed within the particle boundary layer and in the free troposphere, although $C_{total}$ changes. Thus, the EC particle number fraction cannot be used to define the particle boundary layer. In the ATAL, EC particles seem to play a minor role in the composition of the aerosol, while for the convective outflow levels the data suggest an increase in EC as result of detrainment."

was changed to:

"This indicates, within the limitations of the applied methods, that the EC particle type is well mixed within the boundary layer and in the free troposphere, although $C_{total}$ changes. In the ATAL ($> 16$ km), EC particles seem to play a minor role in the composition of the aerosol, while for the convective outflow levels ($< 16$ km), the data suggest an increase of the EC particle number fraction as result of detrainment."

**2. Presentation quality needs to be improved.**

1) Citation formats: Please pay attention to the formats between Author et al. (year) and (Author 1 et al., year; Author 2 et al., year; Author 3 et al., year; …) and use them properly. Please revise the citation format throughout the manuscript and keep consistency.

E.g., P1 L35 "(See for example Fuzzi et al. (2015))" should be changed to (Fuzzi et al., 2015); P2 L10: Change "(e.g., in Froyd et al. (2019))" to (Froyd et al., 2019).

The format was revised over the entire manuscript. The 'e.g.' was used to indicate that this reference is one example of many possible other references.

2) Section 2 Instrument description: I would suggest refining the descriptions of ERICA-LAMS and EIRCA-AMS modules, since most of them have been well described in SMPS and AMS papers. Please emphasize the difference, e.g., the shutter unit (SU) needs more descriptions. Consider combining 2.3 and 2.4. Pleas simplify the headers.

(Numbers of sections refer to the manuscript submitted for review.)

Sections 2.3 and 2.4 were combined and the headers were simplified.

The instrument description is already kept to a minimum. The ERICA-LAMS is published here for the first time and we feel it should be explained in more detail. Some readers may not be very familiar with *both* techniques, as one reviewer actually indicated. And here we hope our description may be useful. The ERICA-AMS is an adopted Aerodyne AMS, but the actual settings such as vaporizer temperature, emission current, etc. are of interest for other AMS users. Although the information content regarding the ERICA-AMS has not been further reduced, the amount of text regarding the ERICA-AMS is now about half of the text regarding the ERICA-LAMS.

The major difference of the ERICA-AMS to the Aerodyne AMS, the use of the shutter unit instead of a chopper, was emphasized. Furthermore, it was highlighted that without a chopper, no size information can be obtained by the ERICA-AMS.

The difference was described in P7 L21-33 (Numbers of pages and lines refer to the manuscript submitted for review). However, the corresponding paragraph was revised.

"For quantitative aerosol composition measurements, the background signal, which originates from air molecules and residual vapor molecules inside the chamber, has to be considered and is subtracted from the aerosol sampling signal. For this purpose, in the commercial Aerodyne AMS (Canagaratna et al., 2007) the particle beam is periodically blocked by a chopper inside the low vacuum stage. By means of the chopper it is also possible to distinguish between different vacuum aerodynamic particle sizes, as the particle flight time duration between passing the (open) chopper and arriving at the vaporizer is size dependent. However, this flight time duration -and the corresponding flight distance between chopper and vaporizer- need to be long enough to achieve such size-resolved sampling. For ERICA-AMS the distance from the shutter to the vaporizer is very short. This would not be the case if we had placed a chopper directly behind the ball joint of the ADL. However, by periodically blocking the particle beam with a chopper at this position, the detection frequency of ERICA-LAMS would have been reduced accordingly. Thus, we decided to use a simple shutter device instead of the chopper. It consists of a C-shaped profile made of metal and is mounted on the shaft of a high-vacuum magnetically-coupled feed-through (Pfeiffer Vacuum GmbH, Germany). The shaft periodically rotates the C-profile by 90° into and back out of the particle beam axis. In this way, the particle stream to the vaporizer is blocked and permitted, respectively, for adjustable time periods."

Was changed to:

"For quantitative aerosol composition measurements, the background signal, which originates from air molecules and residual vapor molecules inside the chamber, has to be subtracted from the aerosol sampling signal. For this purpose, the SU is used to periodically block the particle beam. The SU consists of a C-shaped surface made of metal, which is mounted on the shaft of a high-vacuum magnetically-coupled feed-through (Pfeiffer Vacuum GmbH, Germany). The shaft periodically rotates the shutter by 90° into and back out of the particle beam path. In this way, the particle stream to the vaporizer is blocked

and permitted, respectively, for adjustable time periods. In the commercial Aerodyne AMS (Canagaratna et al., 2007), the particle beam is periodically blocked by a chopper inside the low vacuum stage. By means of the chopper it is possible to distinguish between different vacuum aerodynamic particle sizes, as the particle flight elapsed from its pass through the chopper until its arrival at the vaporizer is size-dependent. The distance between the chopper and the vaporizer and the corresponding flight time need to be long enough to achieve such size-resolved sampling. In the design of the ERICA-AMS, the distance from the shutter to the vaporizer is very short. This would not be the case, if a chopper was mounted directly behind the ball joint of the ADL. However, by periodically blocking the particle beam with a chopper at this position, the detection frequency of ERICA-LAMS would have been reduced accordingly. Thus, a simple shutter has been implemented and the particle size information can only be provided by the PDU of the ERICA-LAMS (see Sect. S4 in the supplement)."

3) Section 3 Instrument characterization: This section is very important and with comprehensive information, but the key points are buried. It would be very hard for the readers to follow since the LAMS and AMS information is mixed in an unclear way. I would highly suggest rewriting this section by considering the following points.

Please separate the characterization of LAMS and AMS first and then discuss complementary features, and also revise the corresponding figures. Besides, move some detailed descriptions, regarding e.g., calibration (e.g., particle size cal in LAMS; AMS IE and RIE cal), instrument alignment (e.g., ADL position scan), in the supporting information, since they are very well described in other publications or user's manual. An example of restructuring: 3.1 Particle beam characterization; 3.2 ERICA-LAMS characterization (Laser beam; Optical detection efficiency; Hit rate; LAMS mass spectra); 3.3 EIRCA-AMS characterization (Collection efficiency; Detection limit; AMS mass spectra; Mass concentration), and 3.4 Overall performance comparison (sensitivity, size, spectra, etc).

The particle time-of-flight calibration (particle size calibration) of the ERICA-LAMS was shifted to the supplement, since the approach with a polynomial fit is described in Brands et al. (2011).

The AMS IE and RIE sections were kept in the main part, since they are instrument specific and of interest for further publications. Also, the values differ from other AMSes. Thus, a presentation in the main part is reasonable.

(Numbers of figures refer to the manuscript submitted for review.)

Fig. 4 (example for the ALS position scan) was moved to the supplement, since the methodology of the measurement (including a figure) is described in Molleker et al. (2020). The basics of the methodology to determine the optical particle detection efficiency and the particle mass detection efficiency in our view should be better presented in the main text. Details of the complex determination procedure can be found in the supplement (Sect. S5, revised manuscript). It has to be emphasized that the ADL position scans are not only used for alignment, but also to determine the parameters for the particle and detection laser beam characteristics and, finally, the parameters $DE_{max}$ and $DE_{KTM}$.

(Numbers of sections refer to the revised manuscript)

Following the reviewer's suggestion (for which we are quite grateful) Section 3 was restructured like this:

Please keep the terminology same as the ones commonly used in SPMS and AMS communities, respectively, e.g., use "hit rate" instead of "ablation efficiency"; use "collection efficiency" instead of "detection efficiency".

(Numbers of sections and equations refer to the revised manuscript)

The term ‚ablation efficiency (AE)' was replaced by the term ‚hit rate (HR)', since this is the more common term in the community and do not exclude other efficiencies as ionization and ion extraction efficiency. The definition is given by Eq. (5) and is the same as used by, e.g., Brands et al. (2011) (termed ablation efficiency), Su et al. (2004) (termed hit rate), and Gemayel et al. (2016) (termed hit rate).

The term ‚detection efficiency' varies within the SPMS literature: In Gemayel et al. (2016) this term is used as the overall detection efficiency: A product of the hit rate and the 'scattering efficiency (SE)'. The latter term is defined as the here used optical detection efficiency $DE_{PDU}$ (Eq. (1)), related to one of the

particles detected at one of the detection lasers. In Marsden et al. (2016), the symbol $E_{detect}$ is used. In Molleker et al. (2020), the term ‚detection efficiency' is used without an abbreviation, but with the same definition as in the manuscript here. In Brands et al. (2011), the detection efficiency of the ALABAMA refers to the number of particles detected at both detection units within a given time interval and whose sizes were successfully determined. In Clemen et al. (2020), the detection efficiency of the ALABAMA also refers to the number of particles detected at both detection units within a given time interval for the measurements at the optimal fixed position of the aerodynamic lens system, whereas the detection efficiency for the ADL scans, just like in this study, refers to the individual detection lasers. Finally, we kept the term ‚detection efficiency' (in the manuscript for clarification with the adjective ‚optical').

Furthermore, the detection efficiencies (optical detection efficiency measured at the PDUs $DE_{PDU}$ and the particle mass detection efficiency measured at the ERICA-AMS vaporizer $DE_{vaporizer}$) are defined by Eq. (1) and in supplement Eq. (S16), respectively and also, the curve fit functions (Eq. (2), (S15), and (S17)). The combination to $DE_{max}$ and $DE_{KTM}$ is described in Sect. S5.6.

The term ‚collection efficiency' is not applicable for measurements with an optical device, since the particles are not „collected" literally. However, it is applicable for the ERICA-AMS, since at the vaporizer the particles get in a sense „collected". The definition of $DE_{vaporizer}$ (Eq. (S16)) is very similar to the definition of the ‚collection efficiency (CE)' used in the AMS community. However, to keep consistency and not to confuse the reader, we keep the term ‚detection efficiency' also for the ‚collection efficiency' of the ERICA-AMS. This is also one of the reasons, why we provide the equations, from which the terms become clearer. To consider the fact that $DE_{vaporizer}$ and CE are defined in the same way, the text paragraph (P12L1, submitted manuscript for review) has been adapted: „ Simultaneously to the measurements with AN particles at the detection units PDU1 and PDU2 of the ERICA-LAMS, the mean mass concentration of AN was measured with the ERICA-AMS, similar to the approach described in Liu et al. (2007). The efficiency with which particle mass concentrations were measured with the ERICA-AMS was determined. While this quantity is equivalent to the 'collection efficiency' (CE; e.g., Canagaratna et al., 2007; Matthew et al., 2008; Drewnick et al., 2015) in AMS measurements, we define it as 'particle mass detection efficiency' for consistency with the ERICA-LAMS discussion. "

4) Section 4: The authors only describe the sizes of EC-containing particles in the last paragraph in this section, which is not strong enough. Please give the information of chemical resolved size distributions obtained by both ERICA-LAMS and ERICA-AMS, and add more discussion accordingly.

(Numbers of sections refer to the revised manuscript)

Readers may expect size distributions from the ERICA-AMS, because these commonly are provided by the Aerodyne AMS. As mentioned above, no size information can be obtained by the ERICA-AMS, due to the lack of a chopper. (The corresponding paragraph in Sect. 2.4 was revised in order to emphasize the difference to the commercial Aerodyne AMS.) Therefore, no size distribution can be shown for the ERICA-AMS.

The EC particle type (single particle data) is just an example that size information from the ERICA-LAMS is evaluable and is meant as "proof-of-concept". However, the original Fig. 19 and its discussion was shifted into the supplement, since the ability to provide size information is already shown in Fig. 11 (submitted manuscript for review), where the size dependency of the hit rate is shown for an ambient

measurement. The determination and evaluation of particle types other than EC and the evaluation of particle size distributions during this field campaign is beyond the scope of this manuscript and will be part of a forthcoming publication.

In the manuscript

"As an example that the ERICA-LAMS provides single particle size information, Fig. 19 shows the size distribution of EC-containing particles for the research flight on 04.08.2017 consisting of three modes. The first at the edge of the small particle sizes below 200 nm, the second between a particle size of around 300 nm and 1700 nm with a maximum particle number fraction of 0.08 at 800 nm, and the third between 1700 nm and 2600 nm with a maximum of 0.17."

was replaced by:

„An example for single particle information, which ERICA-LAMS is capable of delivering, is provided in Sect. S8 of the supplement. Due to the lack of a chopper, no particle size information can be determined by the ERICA-AMS."

5) Figures:

Consider moving some to the supplement, e.g., Fig. 3 and 4, and combing some, e.g., Fig 7 and 8, Fig 8 and 9.

(Numbers of sections and figures refer to the manuscript submitted for review.)

Fig. 3 (and the corresponding Sect. 3.2) was moved to the supplement (see reply to major comment 2.3 above).

Fig. 4 was moved to the supplement, since such types of graphs are presented already in the literature. For example, Molleker et al. (2020) show a graph measured by the ERICA in the main part.

Fig. 7 and 8 were combined.

The combination of Fig. 8 and 9 is not meaningful. In case the reviewer meant Fig. 9 and 10: In order not to overload a graph and keep two different types of detection efficiencies separate, both figures were kept separately.

In Fig. 5, 6, and 7, the solid squares, diamonds and circles with the same colour are not easy to distinguish. Please modify them in a clearer way.

Solid markers were changed to non-filled markers. In addition, the marker size was enlarged to make the error bars visible.

Error bars: Since this sentence "The error bars are in some cases smaller than the symbol" is shown in most of the figure captions (Fig. 3, 5, 6, 7, 8, 9, 10, 11, 16, 17, and 18), I suggest that put the corresponding values in SI. The following question is that in the laboratory how many repeated experiments have been done to generate one data point?

(Numbers of equations and sections refer to the manuscript submitted for review.)

We changed the solid markers to non-filled markers to make the error bars visible. Determination of uncertainties is now described in the figure captions.

For one $DE_{PDU}$ data point (see Eq. (4)), a single run of 30 seconds was performed (see Sect. 3.3.1). Since the measurement of $DE_{vaporizer}$ (see Eq. (S15) in the supplement) was simultaneously performed, the measuring time is the same as for the measurements of one $DE_{PDU}$ data point.

More details on the applied methods can be found in Sect. 3.3 and in the supplement Sect. S2, S3, and S4.

Mass spectra: The x and y scales, as well as the axis labels, are inconsistent among all the spectra. E.g., for y axis, in Fig. 12 it is "signal intensity in a.u." in linear mode, while in Fig. 15 and 18, it is "ion peak area in mV. sample" in log mode. Please try to keep consistency. Please normalize the spectra to the total ion intensity and keep the same scales (both x and y) for consistency.

(Number of figures refer to the manuscript submitted for review.)

Fig. 12: The axes were changed to log scale and the labels were changed to „ion peak area in mV·sample". The abscissas were changed to maximum m/z 250 (gold particle up to m/z 400). Note: BaA and gold particle spectra were swapped.

The spectra show single particle spectra, on which the ion marker threshold can be applied. Thus, a normalization is not appropriate.

3. The advantages of this hybrid instrument are not very well demonstrated, not only due to the poor manuscript structure, but also lacking discussion on complementary results. Please try to improve. Besides, in addition to the compact size, are there any other big advantages of using such a hybrid instrument compared to deploying SPMS and AMS instruments in parallel? Please state the differences.

(Numbers of pages, lines, and sections refer to the manuscript submitted for review.)

- Section 3 of the manuscript was restructured as the reviewer suggested. By that, the instrument presentation was improved and the instrumental design should be much clearer now.
- The discussion on complementary results is part of Sect. 4. For better understanding, this section was revised.
- The instrument was designed initially for the mobile field deployment aboard the high-altitude research aircraft *Geophysica*. Here, valid for all (high-altitude) research aircraft, weight and space for the payload is limited. In addition, field deployments with research aircraft at high altitudes are rare, so as much information as possible (with as many instruments as possible) should be collected. Thus, a compact design is crucial for implementation on such aircraft and therefore a combination of two measurement methods into one apparatus a major advantage.

  „The final design of the compact instrument was implemented into an aircraft rack (Dragoneas et

al., 2022) of 60 cm x 74 cm x 140 cm (height x width x length) with a total weight of 200 kg. Such a compact and light-weight design is essential for aircraft implementation, especially aboard a high-altitude aircraft."

was changed (and on request from Reviewer #3 shifted to Sect. 1):

„ Furthermore, the mechanical components of ERICA are designed to operate under the demanding conditions like thermal stress and vibrations aboard an aircraft. The final design of the compact instrument was implemented into an aircraft rack (Dragoneas et al., 2022) of 60 cm x 74 cm x 140 cm (height x width x length) with a total weight of 200 kg. In addition, field deployments with research aircraft at high altitudes are rare, so as much information as possible − with as many instruments as possible −should be collected. Thus, a compact design is crucial for implementation on such aircraft and therefore a combination of two measurement methods into one apparatus is a major advantage."

– In the outlook (Sect. 5) on P24L33, a future mode for the ERICA is presented. This mode is only possible with a serial linkage of a LAMS and an AMS, like it is in ERICA. The paragraph was revised to highlight this unique feature as an advantage:

„For the same point in time, a data acquisition card is triggered and, similar to the procedure with a light scattering probe on the AMS (Cross et al., 2007; Freutel, 2012), the single particle mass spectrum is recorded. In this way it is possible to quantify the non-refractory components of a single particle. In addition, the size information of the measured single particle is obtained by means of the particle flight time between the two PDUs. Here, a future characterization of interest is the ablation laser's effect to the particles that are only partly ablated and the residuals reach the vaporizer of the ERICA-AMS. For this purpose, a method has to be developed to ensure the linkage of the results to the very same particle. Such a procedure needs more implementations and further laboratory studies."

was changed to

„For the same point in time, the data acquisition card is triggered and the single particle mass spectrum is recorded. For the ERICA this mode is called optically triggered AMS (OT-AMS) mode. With the method of the OT-AMS mode, it is possible to quantify the non-refractory components of single particles when the ablation laser is in idle mode. This method is similar to the procedure with a light scattering probe on the AMS (Cross et al., 2007; Freutel et al., 2013). In addition, the size information of the measured single particle is obtained by means of the particle flight time between the two PDUs. One possible future investigation by means of the OT-AMS mode is the ablation laser's effect on the particles that are only partly ablated and where the residuals reach the vaporizer of the ERICA-AMS. This investigation is only possible with the unique feature, the serial configuration of SMPS and AMS, as in the OT-AMS mode. A method has to be developed to ensure the linkage of the results to the very same particle. Such a procedure needs more implementations and further laboratory studies. "

4. For the current configuration of the LAMS module, it is hard to believe that PSL particles with smaller size < 200 nm can be detected. Several statements on the PSL 80 nm and 108 nm with the corresponding data shown in the Figures 3, 5, 6, 7, 9, 10, and 11 are not valid. Please consider modifying or removing accordingly.

(Numbers of figures refer to the revised manuscript)

We don't understand which statements are regarded not to be valid and why particles smaller than 200 nm diameter should not be detected. The authors are aware, and this is also described in the manuscript or was measured by us, that the detection efficiency decreases significantly below 200 nm. However, the following arguments support that the detection efficiency for PSL particles of sizes 80nm and 108 nm is non-zero:

– Fig. S9 in the supplement shows the histograms of the PSL calibration measurements, which demonstrate the ability of the ERICA, to optically detect particles of sizes in a range between 80 nm to 5145 nm.
– Fig. S21 in the supplement shows the size distribution from a research flight during the second aircraft field campaign of StratoClim on 08.08.2017. Here, mass spectra from particles in a size range between 100 nm and 3700 nm were obtained.
– Fig. 8 was revised and shows the number of ablation laser shots and the number of recorded spectra now in log scale to highlight that ambient particles in the size rage of 80 nm to 4000 nm can be optically detected. Also, particles below 200 nm were ablated during this experiment. However, their hit rate and the numbers are low (HR: 2 to 11 %; 1 to 8 spectra).

**Minor Comments:**

**P1L23-25**: Please change 3170 nm to 3.17 µm or change 3.5 µm to 3500 nm to keep consistency and revise throughout the manuscript.

Done. Particle sizes are given now in ‚nm' (throughout the manuscript).

**P2 L10:** Change "(e.g., in Froyd et al. (2019))" to (Froyd et al., 2019; Author 2 et al., year...), and please add more corresponding references. Lots of quantification work has been done by using ATOFMS and other reference instruments like OPC, AMS, and so on.

"Within certain limitations this may become possible, if the data of other instruments are included in the analysis (e.g., in Froyd et al. (2019)).

was changed to:

„Within certain limitations this may become possible, if the data of other instruments are included in the analysis (e.g., Ault et al., 2009; Healy et al., 2012; Gunsch et al., 2018; Köllner et al., 2021)."

**P3L25**: Please cite the corresponding publications.

Since for ERICA the „large fraction" is an assumption, we changed the text as follows:

„A large fraction of the particles is not ablated by laser pulses, either because the laser pulses miss the particles, or because the particles are too small for the optical detection. However, even most particles amenable for laser ablation, which pass through the ablation region, remain undestroyed, because the laser is firing at a limited maximum repetition rate of 8 pulses per second."

was changed to (see also reply to RC2):

"It is assumed that a large fraction of the sampled particles will not generate a single particle spectrum. The major reasons for this effect are: First, the particles are not ablated, because the laser is firing at a limited maximum repetition rate of 8 pulses per second. During the idle time of the Nd:YAG laser, particles remain unablated, even if they are successfully detected by the units PDU1 and PDU2. This actually is by far the largest fraction of the sampled particles emerging from the ADL. If, for example, the ambient number density of particles with diameters above the optical detection limit is 100 $cm^{-3}_{Std}$, then, at most only 5.4 % (8 shots per second and sampling volumetric flow rate of 1.48 $cm^3$ $s^{-1}$) of the detectable particles are hit by the laser. Second, the particles are too small for optical detection. Third, particles for which the calculation of the trigger failed continue their travel towards the ERICA-AMS vaporizer. Fourth, particles that primarily consist of materials that are transparent at a UV wavelength of 266 nm, such as pure sulfuric acid, are hard to ablate (Murphy et al., 2007). We selected a UV laser with 266 nm wavelength due to smaller dimensions of the laser and the fact, that chemical substances show less fragmentation compared to ablation with shorter wavelengths (Thomson et al., 1997). In general, however, it is also possible to implement excimer lasers operating at shorter wavelength to ablate pure sulfuric acid droplets. Also, pure sulfuric acid is detected by the ERICA-AMS. Thus, even most particles amenable for laser ablation, which pass through the ablation region, remain undestroyed. Another reason why a spectrum is not triggered over a signal threshold for recording is a low number of generated ions during the LDI process."

**P3L32**: Please use 3.17 µm to keep consistency.

Done

**P3L32-33**: It would be more helpful to mention the transmission efficiency of the ADL instead.

(Numbers of sections refer to the manuscript submitted for review.)

The transmission efficiency of the deployed ADL as published by Xu et al. (2017) is mentioned two lines below. However, the term 'transmission efficiency' was not mentioned in the submitted manuscript.

"The detectable particle size range ($d_{va}$) of the ERICA-LAMS is between ~180 nm and 3170 nm (see Sect. 3.3.3). However, the signal-to-noise ratio of optical particle detection is sufficient for particle time-of-flight calibration between 80 nm and 5 µm (see Sect. 3.2). The detectable particle size range of the ERICA-AMS is assumed to be the same as published by Xu et al. (2017) for the deployed lens type.: ~120 nm to 3.5 µm."

was changed to (Numbers of sections and figures refer to the revised manuscript; see also reply to RC2 and RC3):

"The particle size range within the 50 % cut-off in detection efficiency ($d_{50}$) of the ERICA-LAMS is between 180 nm and 3170 nm (see Sect. 3.2.2). The signal-to-noise ratio of optical particle detection is sufficient for particle time-of-flight calibration between 80 nm and 5000 nm (see Sect. S4 in the supplement). For the ERICA-AMS, the detectable particle size range is determined by the transmission and focusing properties of the aerodynamic lens. For the ADL used in our instrument, Xu et al. (2017), who used this lens in combination with an ACSM (Aerosol Chemical Speciation Monitor), determined a transmission range from ~120 nm to 3500 nm. We assume that the detectable particle size range of the ERICA-AMS matches this transmission range."

**P3L36-38**: Please describe the difference between shutter and chopper.

The difference was described in P7 L21-33 (Numbers of pages and lines refer to the manuscript submitted for review). For changes, see reply on major comment 2.2 (above).

**P4L25**: The full name of LAAPTOF should be "Laser Ablation Aerosol Particle Time-Of-Fight mass spectrometer" rather than "…spectrometry".

Done

**P4L26**: Please change the dot in "5·10² cm³ s⁻¹ " to multiplication symbol "5×10² cm³ s⁻¹" and revise the others throughout the manuscript.

Done

**P15L28**: Fig 10 should be Fig 11.

Done

**P16L11**: Please give the reason for choosing these peaks.

„The ion peak area threshold is defined as the ion peak area at $m/z$, which are usually unoccupied ($m/z$ 2 to $m/z$ 6 for cations, $m/z$ 2 to $m/z$ 11 for anions), below which 99% of the baseline noise is present (Köllner et al., 2017)."

was changed to (see also reply to RC2):

„The ion peak area threshold is defined as the ion peak area at $m/z$, on which during ambient measurements typically no signals occur ($m/z$ 2 to $m/z$ 6 for cations, $m/z$ 2 to $m/z$ 11 for anions). To determine the ion peak area threshold, the normalized cumulative signal intensity distributions for each usually unoccupied $m/z$ were made and the overall 99 % threshold was determined (Köllner et al., 2017). Below this ion peak area threshold, 99% of the baseline noise is present (Köllner et al., 2017). The result for cations and anions is an ion peak area threshold value of 7 mV·sample."

**P16L34**: Please assign the peak at m/z 228.

Done

**P19L27**: The left half of the bracket is missing.

Added left half of the bracket

**P20L28-30**: The reader would expect the following focus on meteoric and EC containing types rather than particulate sulfate, which is a compound. Please reshape this sentence to make the transition smoothly.

The paragraphs were re-arranged and revised. The transition was smoothed as follows:

"In this way, two particle types (in addition to other particle types not included in this publication) well described in the literature were found: A meteoric material containing (e.g., Schneider et al. (2021)) and an elemental carbon (EC) containing particle type (e.g., Pratt and Prather (2010)). In the following, we focus on the aerosol composition at high altitudes (> 10 km) considering particulate sulfate and the meteoric material containing particle type."

Was changed to

„With this approach, two particle types (in addition to other particle types not included in this publication) well described in the literature were found: A meteoric material containing (e.g., Schneider et al., 2021) and an elemental carbon (EC) containing particle type (e.g., Pratt and Prather, 2010). To identify the sulfate-containing particle type, the ERICA-LAMS data set was filtered for single particle spectra that contained sulfate marker signals at $m/z$ -96 ($SO_4^-$) or $m/z$ -97 ($HSO_4^-$) or both markers. In the following, first, we focus on the aerosol composition at high altitudes (> 10 km), considering particulate sulfate as well as the meteoric material containing particle type.

**P20L31**: Incorrect statement. Please revise.

„The sulfate particle type measured by the ERICA-AMS is a non-refractory species (Canagaratna et al., 2007) and consists mainly of pure sulfuric acid in the stratosphere (Murphy et al., 2014)."

was changed to

"Non-refractory sulfate (Canagaratna et al., 2007) measured by the ERICA-AMS consists mainly of pure sulfuric acid in the stratosphere (Murphy et al., 2014)."

**P20L35**: Please clarify that when only considering the non-refractory species, the sulfate mass fraction is 1 at 20 km.

„In 20 km altitude, the sulfate mass fraction is 1."

was changed to

„In 20 km altitude, the non-refractory aerosol sulfate mass fraction is 1.

**P21L32-35**: Please add references to support the assumption.

We revised the text passage and added two references and highlighted our assumptions (changes see reply on major comment 1.3 above).

**Fig.1**: Please add TMP 1 to 4 in the figure or point out their positions. Please add the distances between LD1, LD2, ablation spot, shutter unit, vaporizer, etc.

(Number of figures refer to the manuscript submitted for review.)

The TMPs are now numbered from TMP1 to TMP3. TMP1 is a four-stage TMP with the numbered pumping stages PS1 to PS4.

The distances are provided in Fig. S3 in the supplement, since they are not further discussed

**Fig.2**: Consider rescale some sizes/distances. E.g., the distance between convex lens and the quartz window (10 mm) should be twice the size of the ablation laser beam (5 mm). This can be easily done.

After a bit of discussion, we decided to leave this figure as a not-to-scale-drawing, but at least we narrowed the laser beam.

**Fig.3 caption**: (b) is not clear, please reshape the sentence; (3) is confusing, please rewrite.

Done

**Fig.6**: Please use the same scales for the left and right Y-axes.

We prefer to leave the scaling as is. Scaling the left axis to 2.5, the details in presentation would get lost. Scaling the right axis to 0.25, the data points would be out of scale.

**Fig.12**: Please clarify that whether the stick spectra are for individual particles or the averaged ones? If averaged, please give the total number of the spectra for averaging. Please normalize the spectra, e.g., to the total ion intensity, and keep the same scales (both x and y) for consistency. E.g., m/z can be fixed from 0 up to 250 amu. for each spectrum. This can be applied to the special case of gold particles too, only need to illustrate the $Au_2^+$ additionally.

Caption changed to „Exemplary stick mass spectra ($m/z$) of four laboratory generated single particles as measured by ERICA-LAMS."

It is mentioned that the intensities are not normalized.

**Fig.9 and 10**: Please combine them. Please remove the AN measured by AMS and put it in a separate figure.

(Number of figures refer to the revised manuscript)

To clearly differentiate between maximum possible $DE_{max}$ and $DE_{KTM}$ during the aircraft campaign, we preferred to not merge the panels. However, the measurements at the ERICA-AMS were separated to a new figure (Fig. 12).

**Fig.13**: Please give the definition of the "sample number".

(Number of figures refer to the manuscript submitted for review.)

In Fig. 13 the raw spectrum is depicted. The abscissa was changed from "sample number" to the (to a raw spectrum of a TOF-MS) more intuitive term "ion flight time".

The sample number is the number of samples of the oscilloscope (Picoscope) during recording the single particle spectrum. The time resolution is set to 1.6 ns per sample. Thus, by multiplying the sample number by 1.6, the ion flight time (in ns) in the TOF-MS can be determined.

The caption was revised accordingly:

"Details of cation raw spectra (voltage output versus sample number of the digitizer, 1.6 ns per sample) of two ambient single particles at the airport of Kathmandu, Nepal. (a) Tin isotopic pattern ($d_{va}$ = 277 nm). (b) Lead isotopic pattern ($d_{va}$ = 311 nm)."

was changed to:

"Details of cation raw spectra (voltage output versus ion flight time in the B-ToF-MS) of two ambient single particles at the airport of Kathmandu, Nepal. (a) Tin isotopic pattern ($d_{va}$ = 277 nm). (b) Lead isotopic pattern ($d_{va}$ = 311 nm)."

**Fig.14**: (a) It is hard to see the signal difference between shutter open and closed. Please consider a better way to demonstrate. (b) The calculated difference does not agree with the left spectrum. E.g., the bars are apparently not at the same positions between two plots; the most intensive peak m/z 28$^+$ (labelled $N_2^+$) is even a bit higher than the corresponding one in (a), as well as the m/z 32+, 40+, etc. The labels of $N_2$ and $O_2$ are confusing, since the peaks also contain the organic and sulfate fragments, respectively. Please modify them with a clearer way.

Panel (a) was removed, since no further discussion is presented in the text. The tags were changed to m/z values.

**References**

[revised manuscript text omitted]

---

## Author Comment (AC2)

AMT-2021-271

**Design, characterization, and first field deployment of a novel aircraft-based aerosol mass spectrometer combining the laser ablation and flash vaporization techniques**

Hünig et al.

**Replies to the comments by Anonymous Referee #2**

General Reply:

First of all, we would like to thank Referee #2 for reviewing our manuscript and for his/her diligent and helpful comments, which significantly contribute to an improvement. In the following we will comment on the individual points.

The reviewer comments are written in this font style and color.

Our answers are written in this font style and color.

Changes to the revised version of the manuscript are printed in red.

Review of *Design, characterization, and first field deployment of a novel aircraft-based aerosol mass spectrometer combining the laser ablation and flash vaporization techniques* by *Hünig et al.*

Anonymous Reviewer

October 2021

**1 Summary**

In this work, Hünig et al. describe, for the first time, the design and characterization of ERICA. At the time of this review, ERICA is a unique instrument, but it does combine two well-known methods: (1) single-particle mass spectrometry using laser ablation to (partially) vaporize single particles and ionize their constituents, and (2) an AMS-style instrument that flash vaporizes the non-refractory component of aerosol using a hot tungsten filament and creates ions using electron impact. Method 1 will be referred to as ERICA-LAMS, and Method 2 will be referred to as ERICA-AMS, per the authors' designation. ERICA LAMS uses two time-of-flight mass spectrometers to analyze the positive and negative ions from a single particle; ERICA-AMS uses a compact-time-of-flight mass spectrometer to analyze positive ions. Both ERICA-LAMS and

ERICA-AMS share a common aerosol focusing inlet (AFI), which is pressure-controlled and has been written about in a separate publications (Molleker et al., 2020). After exiting the AFI, the particles are sized by measuring the particle time-of-flight between two particle detection units (PDU1 and PDU2). Optical sizing was experiementally achieved for PSL between 80 nm and 5.145 µm Particles detection by PDU2 triggers a 266-nm quadrupled Nd:YAG ablation laser to fire (max repetition rate 8 hz−1, ∼4 mJ/pulse). Particles that are not detected by PDU2 or are missed by the ablation laser are collected ∼55 cm from the exit the AFI, and ∼30.1 cm downstream from the ablation laser spot.

The authors give much attention in the paper to the particle beam diameter and the effective laser / vaporizer diameters. All are fitted parameters, which are fitted to a convolution of two functions–a top hat function for the effective laser / vaporizer width and a 2D Gaussian function for the particle beam width. In ERICA-LAMS, the particle beam width ranges from ∼30-40 µm for 335 nm AN particles to ∼100-200 µm for all particles >400 nm to >500 µm for 103 nm PSL. For particles ≥208 nm, the particle beam diameters are smaller than the effective laser diameters in PDU1 and PDU2. For ERICA-AMS, particles with diameters > 91 nm have particle beams smaller than the effective diameter of the vaporizer, which, unlike the effective laser diameters, is similar to the physical dimensions of the vaporizer (3.8 mm).

The most userful meaasured parameters in the paper are the detection efficiency (DE) and the abation efficiency (AE). The former measures the number of particles detected by the PDUs compared to a separate measurement of particles counts by a CPC or OPC; the latter is the number of particles that has mass spectra divided by the number of particles that trigger PDU2. The DE analysis shows that, under ideal conditions (e.g., idea beam position, which changes as a function of size), the DE for PSL is above 0.6 for particles ≥208 nm; however, for real-world particles the DE is generally lower across all sizes measured. Finally, the AE for real-world urban particles was presented. The AE has a maximum value of 0.52 @ 218 nm; however, the authors also found that the AE is a steep function of size, and hovers around 10-20% for particles below ∼200 nm and above ∼300 nm.

The paper finished with some example laboratory particles, as well as some example particles and science from the first aircraft deployment.

Overall, this paper is very well written and very well thought out. The scope of the paper also fits very well within the scope Aerosol Measurement Techniques. At the time of this review, ERICA is a completely unique instrument; thus, a detailed description and characterization paper is well-timed and necessary for future publications. This reviewer only has a few comments, which are outlined below.

We thank the reviewer for this generally positive rating of our manuscript.

**2 General Comments**

1. Section 3.1–It is unclear to this reviewer if the "razor blade" is integrated into the system like the "knife edge" in the PALMS instrument. If so, it is also unclear if ERICA uses the knife-edge to re-position the papers during flight, where they might have moved due to vibrations from the aircraft.

"For characterization of the laser beams of the PDUs and the ablation laser outside the vacuum chamber, a razor blade was moved stepwise perpendicularly into the respective laser beam (with steps of 0.01 mm)."
changed to:
"For characterization of the laser beams of the PDUs and the ablation laser, a razor blade was moved stepwise perpendicularly into the respective laser beam (with steps of 0.01 mm). These characterization experiments were performed in a separate measurement setup."

2. Section 3.3: It is unclear to the reviewer if the "effective laser radius" being much larger than the physical dimensions of the laser is supported by Mie theory (as was done for 108 nm particles). Is this true? Is this akin to a "scattering cross section?" If so, the authors should support that with some calculations in the supplemental. Otherwise, the authors risk comparing the physical beam diameters to a laser diameter that is fitted (as opposed to measured) and perhaps physically unrealistic.

(Numbers of sections and figures refer to the revised manuscript.)

The different definitions of $w_0$ (1/e$^2$-radius, determined by the knife-edge experiment) and the effective laser radius $r_{eff,L}$ (determined by the ADL scan measurements) have to be considered:

A knife-edge moved into the laser beam allows only the intensity of the open half plane to pass. The power measured on the detector is the integral of the intensity over the unshaded area. The integral is the Gaussian error function (see Sect. 3.2.1 and Sect. S2.1 in the supplement). The beam radius is defined as the difference of the position where the transmission is 16 % and 84 %. In the case of a Gaussian beam, the beam diameter thus determined coincides with the 1/e$^2$ width of the intensity distribution (Eichler et al., 2004).

The effective laser beam radius $r_{eff,L}$ is the laser beam radius wherein a particle is registered (see Sect. 3.1.1). The effective laser beam radius was determined by the ADL position scans (convolution of the particle beam and the effective laser beam; see also Molleker et al., 2020) and depends on the particle size. Larger particles scatter the laser light more than smaller particles, resulting in a larger $r_{eff,L}$ value for larger particles. Thus, a $r_{eff,L}$ value larger the $w_0$ value is possible. It means, the intensity at the distance $r_{eff,L}$ is below 1/e$^2$ of the maximum intensity, but the intensity of the scattered light is still sufficient for a particle to be detected. However, $r_{eff,L} = 4.687 \times w_0$ yields unrealisticly low values for the intensity distribution of a Gaussian beam. Possibly the beam shape does not follow a Gaussian distribution at the edges. We added the lines: „This calculation is valid for a Gaussian beam profile, which is most likely not true on the edges of the distribution, and can thus

only be seen as a rough approximation." in Sect. S5.1 in the supplement and „[...] according to a rough estimation (see Sect S5.1 in the supplement)." in the main paper (Sect. 3.1.2).

The calculated response functions of the Mie curve (see Sect. S5.1 in the supplement) increase with particle size (See Fig. S11). However, it cannot fully explain the shape of the $r_{eff,L}$ curves. In addition, $I_{rel}$ seems to be too small for small particles ($d_{va} < 200$ nm) to be detected. However, measurements show (see Figs. 8, S9, and S21) that particles $d_{va} < 200$ nm can be optically detected and ablated.

3. Fig.10: I am slightly confused how it is possible that PDU2 can have higher values than PDU1. Can the authors comment on this?

(Number of figures refer to the manuscript submitted for review.)

Due to the fact that the ADL position where the optical detection efficiency has its maximum deviate (also for different particles sizes differently) to the adjusted particle beam axis (see Fig. S13), the $DE_{KTM}$ values at PDU2 can be higher than at PDU1. This means for these particle sizes that the particle beam might be better adjusted to PDU2 than to PDU1.

In Fig. S13, the measurements with AN particles with particle sizes between 200 nm and 400 nm show that at PDU2 the $x_0$ offset is smaller than at PDU1. This might result in a higher $DE_{KTM}$ for measurements at PDU2.

4. Section 3.4: Because ERICA has both an optical DE for PDU2 and an AE, it would be helpful for the authors to explicitly show a DE for ablation. This wold help the readers understand biases in ERICA number fractions etc.

Due to the limited repetition rate of the UV laser, a DE for ablation that depends on the particle concentration outside the instrument turns out to be not useful in this context. This is especially the case for high particle concentrations outside the instrument (>12 particles cm$^{-3}$), which exceed the temporal resolution of the UV laser.

5. Section 4: Towards the end of the paper, the authors compare ERICA-LAMS data to ERICA-AMS data on the same plot. This caused this reviewer to of biases between the measurements that should be addressed before having a combined interpretation of the LAMS and AMS results. The major bias, as understood by this reviewer from the figures in this paper, is that the "number fraction" will be highly dependent on the size and composition of the particles present. These should somehow be weighted accordingly–by internal DE curves or by normalizing to external quantitative measurements. No discussion of this correction is present in

the current manuscript–this reviewer strongly suggests that the authors address that in this manuscript, as it will affect all future work from this instrument.

(Numbers of sections and figures refer to the manuscript submitted for review.)

In Section 4, it is not intended to be a comparison to highlight the differences of the ERICA-LAMS and the ERICA-AMS, but to demonstrate the possibility of obtaining complementary information and that this information can be merged. Therefore, not the differences are discussed here. In order to prevent the reader's expectation of a discussion on the differences, Fig. 17 was separated into 3 panels. (See also our reply to RC1)

In the revised manuscript, we revised the presentation of a new and future mode of an Optically Triggered AMS (OT-AMS) of the ERICA. With this mode, it might be possible to investigate residuals from the LDI process with the TD-EI method to investigate the biases. The work on the OT-AMS mode is in progress and the results are substance of an upcoming publication. In our publication here, we solely present the ability of particle detection of the ERICA-LAMS and the ERICA-AMS:

„For the same point in time, a data acquisition card is triggered and, similar to the procedure with a light scattering probe on the AMS (Cross et al., 2007; Freutel, 2012), the single particle mass spectrum is recorded. In this way it is possible to quantify the non-refractory components of a single particle. In addition, the size information of the measured single particle is obtained by means of the particle flight time between the two PDUs. Here, a future characterization of interest is the ablation laser's effect to the particles that are only partly ablated and the residuals reach the vaporizer of the ERICA-AMS. For this purpose, a method has to be developed to ensure the linkage of the results to the very same particle. Such a procedure needs more implementations and further laboratory studies."

was changed to

„For the same point in time, the data acquisition card is triggered and the single particle mass spectrum is recorded. For the ERICA this mode is called optically triggered AMS (OT-AMS) mode. With the method of the OT-AMS mode, it is possible to quantify the non-refractory components of single particles when the ablation laser is in idle mode. This method is similar to the procedure with a light scattering probe on the AMS (Cross et al., 2007; Freutel et al., 2013). In addition, the size information of the measured single particle is obtained by means of the particle flight time between the two PDUs. One possible future investigation by means of the OT-AMS mode is the ablation laser's effect on the particles that are only partly ablated and where the residuals reach the vaporizer of the ERICA-AMS. This investigation is only possible with the unique feature, the serial configuration of SMPS and AMS, as in the OT-AMS mode. A method has to be developed to ensure the linkage of the results to the very same particle. Such a procedure needs more implementations and further laboratory studies. "

**3 Minor Comments**

1. P2L40: Since each paper should stand on its own–a brief description of
   the Dragoneas paper should be described here. That way the reader does
   not have to download a separate paper to fully understand your methods.

Dragoneas et al. (2022), meanwhile completed and ready for submission, includes the detailed technical description of the electronics and the hardware of the ERICA. All for understanding necessary details are included in the manuscript. However, the sentence in P2L40 was revised.

"The adopted techniques for automatizing the operation are detailed in the companion paper by Dragoneas et al. (2022)."

was changed to

"The adopted techniques for automating the operation of the ERICA (including pressure and temperature control), details on the electronic hardware, the mechanical adaption, the inlet system, the electrical distribution, and the remote control, are detailed in the separate paper by Dragoneas et al. (2022)."

2. P3L24: "A large fraction" here is largely meaningless without some general
   numbers or statistics.

To give an idea of the amount of the "large fraction", the idle time of the ablation laser was emphasized. It has to be noted that the losses depend also on the ambient aerosol concentration (in the detectable size range).

„A large fraction of the particles is not ablated by laser pulses, either because the laser pulses miss the particles, or because the particles are too small for the optical detection. However, even most particles amenable for laser ablation, which pass through the ablation region, remain undestroyed, because the laser is firing at a limited maximum repetition rate of 8 pulses per second."

was changed to (see also reply to RC1):

"It is assumed that a large fraction of the sampled particles will not generate a single particle spectrum. The major reasons for this effect are: First, the particles are not ablated, because the laser is firing at a limited maximum repetition rate of 8 pulses per second. During the idle time of the Nd:YAG laser, particles remain unablated, even if they are successfully detected by the units PDU1 and PDU2. This actually is by far the largest fraction of the sampled particles emerging from the ADL. If, for example, the ambient number density of particles with diameters above the optical detection limit is $100 \ cm^{-3}{}_{Std}$, then, at most only 5.4 % (8 shots per second and sampling volumetric flow rate of $1.48 \ cm^3 \ s^{-1}$) of the detectable particles are hit by the laser. Second, the particles are too small for optical detection. Third, particles for which the calculation of the trigger failed continue their travel towards the ERICA-AMS vaporizer. Fourth, particles that primarily consist of materials that are transparent at a UV wavelength of 266 nm, such as pure sulfuric acid, are hard to ablate (Murphy et al., 2007). We selected a UV laser with 266 nm wavelength due to smaller dimensions of the laser and the fact, that chemical substances show less fragmentation compared to ablation with

shorter wavelengths (Thomson et al., 1997). In general, however, it is also possible to implement excimer lasers operating at shorter wavelength to ablate pure sulfuric acid droplets. Also, pure sulfuric acid is detected by the ERICA-AMS. Thus, even most particles amenable for laser ablation, which pass through the ablation region, remain undestroyed. Another reason why a spectrum is not triggered over a signal threshold for recording is a low number of generated ions during the LDI process."

3. P3L34: Is the lens and geometry in ERICA the same as the lens in XU et al.?

Yes, see also Sect 2.2.

"The detectable particle size range ($d_{va}$) of the ERICA-LAMS is between ~180 nm and 3170 nm (see Sect. 3.3.3). However, the signal-to-noise ratio of optical particle detection is sufficient for particle time-of-flight calibration between 80 nm and 5 μm (see Sect. 3.2). The detectable particle size range of the ERICA-AMS is assumed to be the same as published by Xu et al. (2017) for the deployed lens type.: ~120 nm to 3.5 μm."

was changed to (Numbers of sections and figures refer to the revised manuscript; see also reply to RC1 and RC3):

"The particle size range within the 50 % cut-off in detection efficiency ($d_{50}$) of the ERICA-LAMS is between 180 nm and 3170 nm (see Sect. 3.2.2). The signal-to-noise ratio of optical particle detection is sufficient for particle time-of-flight calibration between 80 nm and 5000 nm (see Sect. S4 in the supplement). For the ERICA-AMS, the detectable particle size range is determined by the transmission and focusing properties of the aerodynamic lens. For the ADL used in our instrument, Xu et al. (2017), who used this lens in combination with an ACSM (Aerosol Chemical Speciation Monitor), determined a transmission range from ~120 nm to 3500 nm. We assume that the detectable particle size range of the ERICA-AMS matches this transmission range."

4. P8L29: At what aerosol concentration (number and volume / mass), does ERICA-LAMS affect ~30% of the particles? This should be spelled out for the reader? I assume it could affect some areas of the Upper Troposphere.

Here we show two cases: The first, where we have maximum losses at the ERICA-AMS (theoretically 100% for particle detection rates < 8 particles s$^{-1}$, approx. 5 particles cm$^{-3}$ within the detectable size range with a flow into the instrument of 1.48 cm$^3$ s$^{-1}$), and the second, the typical case in the BL (5.4% for particle detection rates >100 particles s$^{-1}$, approx. 68 particles cm$^{-3}$ within the detectable size range).

By calculation, 30 % losses in the particle numbers equal (1/0.3)×8=27 particles s$^{-1}$, approx. 18 particles cm$^{-3}$. In the UTLS (>15 km), we measured a particle detection rate of between 5 and 800 particles s$^{-1}$. Thus, for these measurements, losses for the mass concentration of up to 100 % have to be considered and the uncertainty of 30% has to be adapted.

"However, the losses can be neglected considering the commonly assumed uncertainty of 30 % in AMS instruments."

was changed to:

"However, the losses (in mass) are small considering the commonly assumed uncertainty of 30 % in AMS instruments (Bahreini et al., 2009). By calculation, 30 % losses for the particle numbers equal 27 particles s$^{-1}$, (~18 particles cm$^{-3}$). In the upper troposphere and lower stratosphere (UTLS; >15 km), we measured a particle detection rate of between 5 and 800 particles s$^{-1}$. Thus, for such measurements, losses for the mass concentration of up to 100 % have to be considered and the uncertainty of 30 % has to be adapted."

5. P10L18: That the aerodynamic diameters of AN are similar to PSL suggest that they are spherical and of similar density. This not entirely surprising because AN is notoriously difficult to effloresce; however, the authors state that effective laser radius for AN do not match PSL because the AN are non-spherical. Can you reconcile these two statements?

(Numbers of sections and figures refer to the revised manuscript)

The reviewer is right. The statement that $r_{eff,L}$ (Sect. 3.1.2) does not match PSL because of the non-spherical shape was removed

"The AN measurement results do not agree with the results of the measurements with PSL particles, possibly due to a non-spherical shape and a different refractive index of AN as compared to that of PSL."

was changed to:

"The AN measurement results do not agree with the results of the measurements with PSL particles, possibly due to a different refractive index of AN as compared to that of PSL."

6. P13L32: I'm not sure that "$w_{0,dia}$" is not the most meaningful measurement for overlap. Unlike the signal in PDU1 and PDU2, the intensity of the ablation laser will be essential to the interpretation of the mass spectra–especially for large or coated particles. Thus, a measure of the overlap between the particle beams and where the ablation laser is sufficiently powerful is indeed important to report.

As described for $S_{detect,L}$ (P12L36; refer to the manuscript submitted for review), the laser intensity of a Gaussian beam profile provides intensities larger than zero also for radial distances above $w_0$ and the scattered light might be sufficient for particles to be detected (see also comment on 2.2). However, $w_{0,dia}$ is supposed to be an approximation. Of course, this approximation is good enough to be shown here, as the reviewer also agrees.

7. P13L35: I don't think I saw any evidence that the 80 nm and 5145 nm
   particles were ablated and detected by the MCP. Is this true? If so,
   perhaps a $AE_{max}$ could be shown for PSL particles much like $DE_{max}$
   was?

(Numbers of sections and figures refer to the revised manuscript)

The reviewer is right. Fig. S20 in the supplement shows the size distribution from a research flight during the second aircraft field campaign of StratoClim on 08.08.2017. Here, mass spectra from particles in a size range of between 100 nm and 3700 nm were obtained.

The range from 80 nm to 5245 nm is the size range of the particle size calibration. This is the maximum possible size range where particles are detected by the PDUs (see Sect. S3 and S4 in the supplement) and is theoretically the maximum possible size range for ablated particles.

8. P18L21: This reviewer is not an AMS expert–but, as written, it sounds
   like all RIEs are relative to the nitrate IE. So, why does nitrate have an
   RIE of 1.1?

This is correct. It is explained in Canagaratna et al. (2007): "The RIE values usually used in AMS ambient concentration calculations are 1.4 for organic molecules and 1.1, 1.15, and 3.5–6 for NO3, SO4, and NH4 moieties, respectively. These values are based on many calibrations of laboratory-generated aerosols. The RIEs for NO3 is greater than 1 to account for the fact that although only m/z 30 and 46 are used to track NO3 ion signal during calibrations, NO3 signals at other ion fragments are included in the fragmentation table that is used for calculating NO3 concentrations (Allan et al., 2004; Hogrefe et al., 2004b)."

9. P19L9: As written, it is unclear if it is most desirous to have a "small air
   beam sample" over no air beam sample.

From a measurement statistical point of view an airbeam of zero would be the optimum. However, in practical "no air beam sample" would not be feasible, due to the instrumental design.

"A small airbeam signal is thus desirable, e.g., to reduce the detection limit of aerosol species."

was changed to

"An airbeam signal as small as possible is thus desirable, e.g., to reduce the detection limit of aerosol species."

10. P20L34: Can an estimate of the UT and LT altitude / altitude ranges be
    added to Fig. 17?

The cold point tropopause (17 km) was added as blue dashed line in Fig. 15 (revised version of the manuscript).

11. P22L25: It seems to this reviewer that different removal rates of EC and
    Ctotal suggests that the particles are not well mixed–because they would
    then be removed at the same rates.

We actually do not know the removal rates or the whether the different particle classes are vertically well mixed. In this instrument-focused paper we only describe the observation, in order to highlight that ERICA-LAMS is capable of doing such differentiated measurements in the real atmosphere. The paragraph was revised (see also reply to RC1)
"This indicates within the limitations of the applied methods that the composition of the sampled aerosol is well mixed within the particle boundary layer and in the free troposphere, although $C_{total}$ changes. Thus, the EC particle number fraction cannot be used to define the particle boundary layer. In the ATAL, EC particles seem to play a minor role in the composition of the aerosol, while for the convective outflow levels the data suggest an increase in EC as result of detrainment."
was changed to:
"This indicates, within the limitations of the applied methods, that the EC particle type is well mixed within the boundary layer and in the free troposphere, although $C_{total}$ changes. In the ATAL ($> 16$ km), EC particles seem to play a minor role in the composition of the aerosol, while for the convective outflow levels ($< 16$ km), the data suggest an increase of the EC particle number fraction as result of detrainment."

12. P22L36: Are these EC particles from coagulation? They seem quite high
    to be primary particles.

The EC particles are termed ‚primary', since they are not secondary formed (i.e., not formed from gaseous substances by chemical reaction or by accumulation of reaction products on condensation nuclei). We cannot state whether the EC particles were emitted at these altitudes and grew by coagulation or whether they were transported vertically. This would require more detailed meteorological analyses, e.g., considering air mass trajectories, to see where/how far potential sources might be. This is beyond the scope of the paper. The presented results regarding EC and $C_{tot}$ merely serve to demonstrate ERICAs range of capabilities.

13. P23L5: The authors often differentiate the EREICA-AMS data by say
    "the non-refractory components." This is misleading because ERICA-
    LAMS also measure the non-refractory components.

(Number of pages, lines, and sections refer to the manuscript submitted for review)

The reviewer is right that the ERICA-LAMS is capable to measure non-refractory and refractory components whereas with the ERICA-AMS only non-refractory components can be measured. However, we cannot distinguish, whether sulfate as measured by the ERICA-LAMS is non-refractory or refractory.

Following sentence was added in Sect. 4 (P20L40):

„It has to be noted that the ERICA-LAMS is capable of measuring sulfate species of non-refractory and refractory types, but cannot distinguish between both types."

The sentence (P23L5) „For the non-refractory components, the cations are detected with a C-ToF-MS." was removed

In P23L9 „The cations generated by the TD-EI technique are detected with a C-ToF-MS" was added

14. Figures: It is really hard, especially with the errors bars to differentiate the filled circles from the filled squares. Perhaps switch to filled and open squares?

The markers were changed to non-filled markers to estimate the uncertainty bars. In addition, the markers were enlarged for better differentiation.

15. Figure 10: Using 50% of the max is a bit strange in this plot–it results in PDU1 having larger D50s than PDU2, which is counter-intuitive given that PDU2 has better detection efficiencies.

The reviewer is right. Due to the relatively low maximum $DE_{KTM}$ value for PSL measurements at PDU2 (0.53) compared to PDU1, the found $d_{50}$ values at PDU2 (160 nm and 750 nm) are very small and misleading. An alternative would be another definition of $d_{50}$: 50% absolute.

We still hold the view that 50% of the maximum should be used as a parameter. Because of the small efficiencies and because of the large variation of the measured values, the $d_{50}$ values (interpolated from them) have a large uncertainty and the values determined of 190 nm and 160 nm (the same for 745 nm and 750 nm) are within their uncertainties. Therefore, only the $d_{50}$ values determined from the measurements at PDU1 are shown in Fig. 7a. The corresponding paragraph in Sect. 3.2.2 has been amended:

"In Fig. 10a, the detection efficiency $DE_{KTM}$ of PSL particles is plotted as a function of the particle size $d_{va}$. The graph shows an increase with particle size until a maximum for $DE_{KTM}$ of 0.74 for a particle size of 410 nm. By interpolation, the lower $d_{50}$ values are 190 nm at PDU1 and 160 nm at PDU2. As upper $d_{50}$ values we found 745 nm at PDU1 and 750 nm at PDU2. Furthermore, $d_{50}$ is pronounced differently for particles with optical properties other than PSL such as AN." (Number of figures refer to the manuscript submitted for review)

was changed to

"In Fig. 7a, the detection efficiency $DE_{KTM}$ of PSL particles is plotted as a function of the particle size $d_{va}$. The graph shows an increase with particle size up to a maximum for $DE_{KTM}$ of 0.74 for a particle size of 410 nm. By interpolation, the lower $d_{50}$ value at PDU1 is 190 nm and the upper $d_{50}$ value is 745 nm. Due to the relatively low maximum $DE_{KTM}$ value for PSL measurements at PDU2 (0.53) compared to PDU1, the found $d_{50}$ values at PDU2 (160 nm and 750 nm) are misleading. In Fig. 7b it can be seen that $d_{50}$ is pronounced differently for particles with optical properties other than PSL such as AN." (Number of figures refer to the revised manuscript)

16. Figure 11: Can you make the right side of this plot a log-scale (and also possibly the left?). It is hard to see if you're getting spectra for any particles below ~120 nm or above ~1 μm.

Right axis was changed to log-scale.

17. Figure 12: Why do you have a large $Na^+$ peak in your PAH spectra? Is your mass scale possibly off?

The spectrum showed a sodium contaminated BaA particle. It was replaced by a not contaminated one. It should be noted that Na produces a distinct peak even at very small Na fractions because of its low 1$^{st}$ ionization energy.

**4 Technical Comments**

- P1l11: What does "ERC" stand for?

  ERC stands for 'European Research Council'. The parenthesis was changed from
  "(i.e., ERC Instrument for Chemical composition of Aerosols)"
  to
  „(ERC Instrument for Chemical composition of Aerosols; ERC: European Research Council)"

- P1L15: Perhaps "*The same* aerosol sample can be sampled with both methods *simultaneously*?

  "The aerosol sample can be analyzed with both methods, each using time-of-flight mass spectrometry."
  was changed to
  „ The same aerosol sample can be sampled with both methods simultaneously, each using time-of-flight mass spectrometry."

- P1L20,25,26: The acronyms ADL, B-ToF-MS an C-ToF-MS are defined here, but are not used again in the abstract. The abstract should generally stand alone, and therefore these acronyms can be omitted, but need to be defined at their first use in the main section of the paper.

  Done

- P1L36: You probably can delete the comma after "anthropogenic-"

  Done

- P2L19: Perhaps use "e.g.," instead of "beside others by."

  Done

- P13L33: This reviewer is not sure "However" is the right word here–this statement does not seem to be related to the previous sentence.

  „ However, $S_{ablation}$ smaller than 1 indicates that $1\sigma$ of the particle beam is within the $w_{0,dia}$ of the ablation laser spot."
  was changed to
  „ At least, $S_{ablation}$ smaller than 1 indicates that $1\sigma$ of the particle beam is within the $w_{0,dia}$ of the ablation laser spot. "

- P16L10: It is hard to understand ion peak threshold as currently described. It might be easier to understand by splitting this statement up into two or more sentences.

  „The ion peak area threshold is defined as the ion peak area at $m/z$, which are usually unoccupied ($m/z$ 2 to $m/z$ 6 for cations, $m/z$ 2 to $m/z$ 11 for anions), below which 99% of the baseline noise is present (Köllner et al., 2017)."

  was changed to (see also reply to RC1)
  „The ion peak area threshold is defined as the ion peak area at $m/z$, on which during ambient measurements typically no signals occur ($m/z$ 2 to $m/z$ 6 for cations, $m/z$ 2 to $m/z$ 11 for anions). To determine the ion peak area threshold, the normalized cumulative signal intensity distributions for each usually unoccupied $m/z$ were made and the overall 99 % threshold was determined (Köllner et al., 2017). Below this ion peak area threshold, 99% of the baseline noise is present (Köllner et al., 2017). The result for cations and anions is an ion peak area threshold value of 7 mV·sample."

- P19L19: You can probably delete "especially" in this line.

  Done

- P22L25: The statement "within the limitations of the applied method" is parenthetical and needs commas around it.

  Done

**References**

Allan, J. D., Delia, A. E., Coe, H., Bower, K. N., Alfarra, M. R., Jimenez, J. L., Middlebrook, A. M., Drewnick, F., Onasch, T. B., Canagaratna, M. R., Jayne, J. T., and Worsnop, D. R.: A generalised method for the extraction of chemically resolved mass spectra from Aerodyne aerosol mass spectrometer data, J. Aerosol Sci, 35, 909-922, https://doi.org/10.1016/j.jaerosci.2004.02.007, 2004.

Bahreini, R., Ervens, B., Middlebrook, A. M., Warneke, C., de Gouw, J. A., DeCarlo, P. F., Jimenez, J. L., Brock, C. A., Neuman, J. A., Ryerson, T. B., Stark, H., Atlas, E., Brioude, J., Fried, A., Holloway, J. S., Peischl, J., Richter, D., Walega, J., Weibring, P., Wollny, A. G., and Fehsenfeld, F. C.: Organic aerosol formation in urban and industrial plumes near Houston and Dallas, Texas, J. Geophys. Res.-Atmos., 114, https://doi.org/10.1029/2008JD011493, 2009.

Canagaratna, M. R., Jayne, J. T., Jimenez, J. L., Allan, J. D., Alfarra, M. R., Zhang, Q., Onasch, T. B., Drewnick, F., Coe, H., Middlebrook, A., Delia, A., Williams, L. R., Trimborn, A. M., Northway, M. J., DeCarlo, P. F., Kolb, C. E., Davidovits, P., and Worsnop, D. R.: Chemical and microphysical characterization of ambient aerosols with the aerodyne aerosol mass spectrometer, Mass Spectrom. Rev., 26, 185-222, https://doi.org/10.1002/mas.20115, 2007.

Cross, E. S., Slowik, J. G., Davidovits, P., Allan, J. D., Worsnop, D. R., Jayne, J. T., Lewis †, D. K., Canagaratna, M., and Onasch, T. B.: Laboratory and Ambient Particle Density Determinations using Light Scattering in Conjunction with Aerosol Mass Spectrometry, Aerosol Sci. Technol., 41, 343-359, https://doi.org/10.1080/02786820701199736, 2007.

Dragoneas, A., Molleker, S., Appel, O., Hünig, A., Böttger, T., Hermann, M., Drewnick, F., Schneider, J., Weigel, R., and Borrmann, S.: The realization of autonomous, aircraft-based, real-time aerosol mass spectrometry in the stratosphere, Atmos. Meas. Tech., in preparation, n/a, 2022.

Eichler, J., Dünkel, L., and Eppich, B.: Die Strahlqualität von Lasern – Wie bestimmt man Beugungsmaßzahl und Strahldurchmesser in der Praxis?, Laser Technik Journal, 1, 63-66, https://10.1002/latj.200790019, 2004.

Freutel, F.: Einzelpartikel- und Ensemblemessungen mit dem Aerosolmassenspektrometer (AMS): Untersuchungen zu Quellen und chemischer Prozessierung von Aerosolpartikeln im Submikrometerbereich, PhD thesis, Johannes Gutenberg-Universität Mainz, Mainz, Germany, https://doi.org/10.25358/openscience-4367, 2012.

Freutel, F., Drewnick, F., Schneider, J., Klimach, T., and Borrmann, S.: Quantitative single-particle analysis with the Aerodyne aerosol mass spectrometer: development of a new classification algorithm and its application to field data, Atmos. Meas. Tech., 6, 3131-3145, https://10.5194/amt-6-3131-2013, 2013.

Hogrefe, O., Schwab, J. J., Drewnick, F., Lala, G. G., Peters, S., Demerjian, K. L., Rhoads, K., Felton, H. D., Rattigan, O. V., Husain, L., and Dutkiewicz, V. A.: Semicontinuous PM2.5 Sulfate and Nitrate Measurements at an Urban and a Rural Location in New York: PMTACS-NY Summer 2001 and 2002

Campaigns, J. Air Waste Manag. Assoc., 54, 1040-1060, https://10.1080/10473289.2004.10470972, 2004.

Köllner, F., Schneider, J., Willis, M. D., Klimach, T., Helleis, F., Bozem, H., Kunkel, D., Hoor, P., Burkart, J., Leaitch, W. R., Aliabadi, A. A., Abbatt, J. P. D., Herber, A. B., and Borrmann, S.: Particulate trimethylamine in the summertime Canadian high Arctic lower troposphere, Atmos. Chem. Phys., 17, 13747-13766, https://doi.org/10.5194/acp-17-13747-2017, 2017.

Molleker, S., Helleis, F., Klimach, T., Appel, O., Clemen, H.-C., Dragoneas, A., Gurk, C., Hünig, A., Köllner, F., Rubach, F., Schulz, C., Schneider, J., and Borrmann, S.: Application of an O-ring pinch device as a constant pressure inlet (CPI) for airborne sampling, Atmos. Meas. Tech., 2020, 1-13, https://doi.org/10.5194/amt-2020-66, 2020.

Murphy, D. M., Cziczo, D. J., Hudson, P. K., and Thomson, D. S.: Carbonaceous material in aerosol particles in the lower stratosphere and tropopause region, J. Geophys. Res.-Atmos., 112, https://doi.org/10.1029/2006jd007297, 2007.

Thomson, D. S., Middlebrook, A. M., and Murphy, D. M.: Thresholds for Laser-Induced Ion Formation from Aerosols in a Vacuum Using Ultraviolet and Vacuum-Ultraviolet Laser Wavelengths, Aerosol Sci. Technol., 26, 544-559, https://doi.org/10.1080/02786829708965452, 1997.

Xu, W., Croteau, P., Williams, L., Canagaratna, M., Onasch, T., Cross, E., Zhang, X., Robinson, W., Worsnop, D., and Jayne, J.: Laboratory characterization of an aerosol chemical speciation monitor with PM2.5 measurement capability, Aerosol Sci. Technol., 51, 69-83, https://doi.org/10.1080/02786826.2016.1241859, 2017.

---

## Author Comment (AC3)

AMT-2021-271

**Design, characterization, and first field deployment of a novel aircraft-based aerosol mass spectrometer combining the laser ablation and flash vaporization techniques**

Hünig et al.

**Replies to the comments by Dr. Nicholas Marsden, Referee #3**

General Reply:

First of all, we would like to thank Dr. Nicholas Marsden from the University of Manchester for reviewing our manuscript and for his helpful comments to improve it. In the following we will comment on the individual points.

The reviewer comments are written in this font style and color.

Our answers are written in this font style and color.

Changes to the revised version of the manuscript are printed in red.

The authors present the design and development of a mass spectrometry system for comprehensive measurement of aerosol composition, in which two commonly used techniques, single particle mass spectrometry (SPMS) and aerosol mass spectrometry (AMS) are combined in a single tandem instrument. The manuscript represents a substantial body of work that required considerable expertise in instrument design including differential pumped vacuum systems, optical particle detection and time-of-flight mass spectrometry (TOFMS). A substantial amount of data is presented to evaluate the instrument design. The subject matter is very suitable for this journal but some important issues need to be addressed in the content if this manuscript is to be used as an instrument characterisation reference for future publications.

**Major Comments**

Both instrument use TOFMS as an analyser. This should be introduced and the benefits explained.

We included a short introduction of the TOFMS technique in Sect. 1 and refer to the rich literature on this topic:

"For single particle analysis by the LDI method, a Time-Of-Flight Mass Spectrometer (TOFMS) is a suitable choice, because in this way a full bipolar mass spectrum of a single particle can be recorded (Hinz et al., 1996). The trigger signal for firing the laser pulse that causes the ionization of the particle can be used as the trigger of the TOFMS. Thereby, the ions are separated from neutral molecules in less than a microsecond, preventing further reactions between ions and molecules as for example in an ion trap mass spectrometer (Fachinger et al., 2017). For the TD-EI technique (Aerodyne AMS), a quadrupole mass spectrometer was used in the beginning (Jayne et al., 2000) until it was replaced by TOFMS (Drewnick et

al., 2005; DeCarlo et al., 2006). The advantages of the TOFMS are higher m/z resolution, higher sensitivity and thereby lower detection limits compared to the quadrupole technique (DeCarlo et al., 2006). Additionally, the TOFMS makes it also possible to perform single particle analysis using thermal desorption technique, provided an optical triggering of the detected particles (Cross et al., 2009; Freutel et al., 2013). Furthermore, TOF mass spectrometers are compact and rugged (Noble et al., 1994)."

They both also use aerodynamic lens inlet. The main difference is with the ionisation techniques employed to achieve the desired measurement. The pros and cons to each technique and the consequences on the data should be developed in the introduction. Both techniques are hard ionisation that causes intense fragmentation that has to be dealt with in the data analysis. In the case of laser desorption ionisation (SPMS), this renders the measurements inherently non-quantitative for molecular ion species. The thermal desorption ionisation method used in the AMS method is only quantitative with careful calibration. The authors present some details of the mass calibration in terms or the relative ionisation efficiencies (RIE) of nitrate, sulphate, and ammonium using the same method used for the Aerodyne AMS family of instruments. This is where my first major concern with the work arises.

In various places throughout the document the authors state the ERICA-AMS is 'similar' in design to the Aerodyne AMS, but the similarity is not described nor are the differences. In fact, no detailed description of the vaporiser, ioniser and ion extraction optics is given. The Thermal Desorption ionisation technique (TDI) is not well understood and Quantitative nature of the Aerodyne AMS instrument is underpinned by a large body of publications and method development (See Jimenez 2016 and references therein). If the authors wish to convey these characteristics onto their instrument, they need demonstrate equivalence in the design, particularly regarding the geometry of the ionisation source and the incident particle beam.

Vaporizer, ioniser and ion extraction, as well as the C-ToF mass spectrometer are exactly the same as in the commercial C-ToF-AMS, ToF-ACSM and miniAMS. The details are described in Drewnick et al. (2005), Canagaratna et al. (2007), and Fröhlich et al. (2013).

There are two marked differences: The use of a shutter unit instead of a chopper and a longer particle flight path between aerodynamic lens exit and vaporizer. In the ERICA AMS, quantification is given in the same way as in the commercial AMS, since the shutter performs the same function as the chopper in the AMS.

The corresponding paragraph was revised (including revisions due to other reviewer comments).

"During the idle time of the Nd:YAG laser particles remain unablated, even if they are successfully detected by the units PDU1 and PDU2. This actually is by far the largest fraction of the sampled particles emerging from the ADL. If, for example, the ambient number density of particles with diameters above the detection limit is 100 $cm^{-3}_{Std}$, then, at most only 5.4 % (8 shots per second and sampling volumetric flow rate of 1.48 $cm^3 s^{-1}$) of the detectable particles are hit by the laser. Second, particles for which the calculation of the trigger failed continue their travel towards the ERICA-AMS vaporizer. Third, particles that primarily consist of materials that are transparent at a UV wavelength of 266 nm, such as pure sulfuric acid, are hard to ablate (Murphy, 2007). We selected a UV laser with 266 nm wavelength due to smaller

dimensions and the fact, that chemical substances show less fragmentation compared to ablation with shorter wavelengths (Thomson et al., 1997). In general, however, it is also possible to implement excimer lasers operating at shorter wavelength to ablate pure sulfuric acid droplets. Also, pure sulfuric acid is detected by the ERICA-AMS."

was changed to (Numbers of sections refer to the revised manuscript)

"All particles which are not ablated in ERICA-LAMS (see Sect. 2.3) continue their flight towards the ERICA-AMS instrument part. The design of the ERICA-AMS is the same as the design of the commercial Aerodyne AMS, which is described in the literature (Drewnick et al., 2005; Canagaratna et al., 2007). However, a major difference to the commercial AMS is the use of the SU in the ERICA-AMS instead of a chopper and a longer particle flight path between the ADL and the vaporizer (see below). In the ERICA AMS, quantification is given in the same way as in the commercial AMS, since the shutter performs the same function as the chopper. The vaporizer, ionizer and ion optics, as well as the C-ToF-MS are identical to those in the commercial Aerodyne C-ToF-MS, ToF-ACSM, and miniAMS. The details are described in Drewnick et al. (2005), Canagaratna et al. (2007), and Fröhlich et al. (2013)."

This leads to the second point of major concern with this manuscript regarding the measurement/calculation the particle beam width. The method description is extremely difficult to follow in the current version of the document and it is impossible to get any sense of the error in the calculation. This needs to be addressed. The authors use a method in which the particle beam is tracked across optical detection system which is kept static, in a very similar method to that presented in Marsden 2016 (not cited here) with the LAAPTOF single particle mass spectrometer, an instrument with many common features to the ERICA LAMS. The results are quite different regarding the ratio of particle beam and detection laser beam width compared to the LAAPTOF. This may be due to a superior quality aerodynamic lens, but the result should be discussed with respect to LAAPTOF and other instrument design as this is an important factor in instrument design.

(Numbers of sections and figures refer to the revised manuscript)

The approach of the ADL scan, which is similar to Marsden et al. (2016), was included in the description of the method in Sect. 3.1.1.: „ This approach, which is similar to the method reported by e.g., Marsden et al. (2016) and Clemen et al. (2020), is described by Molleker et al. (2020)."

Based on a comment from Referee #1, Sect. 3, which contains the basic method description, has been restructured. Therefore, the method should be better presented in the revised manuscript. Details on the method to determine the detection efficiencies for AN particles (carrying single or double electrical charges) are provided in the supplement (Sects. S5.2, S5.3, and S5.5). The calculations of the effective laser radii $r_{eff,L}$ for PSL particles (108 nm) and for AN particles (138 nm and 91 nm) are also provided in the supplement (Sect. 5.1). As described in Sect. 3.1.1, the alternative determination of $r_{eff,L}$ of the latter three measurements was necessary, because the losses between PDU1 and PDU2 seemed reasonable due to the particle beam divergence (Huffman et al., 2005).

The visibility of error bars in the graphs (Figs. 3, 4, 5, 6, 7, 8, 12, 14, 15, 16, S6, S7, S10, S13, S14, S15, S16, S17, S18, and S21) has been improved by using non-filled markers.

As mentioned in the captions of the figures (Figs. 3, 4, 5, S16, S17, and S18), the uncertainties of $w_{part}$, $r_{eff,L}$, $r_{eff,V}$, $S_{detect,L}$, $S_{detect,V}$, and $S_{ablation}$ (and $x_{0,shift}$, particle beam divergence $\alpha$, and $A_{scan}$; latter three see Sect. S5.7 in the supplement) result from the curve-fittings (one standard deviation). The uncertainty of $r_{eff,L}$ for the PSL measurement with particle size of 108 nm was estimated to be 0.002 mm (PDU1) and 0.004 mm (PDU2) and the uncertainties of $r_{eff,L}$ for the AN measurements with particle sizes of 138 nm and 91 nm are conservatively estimated to be 0.009 mm at PDU1 and 0.014 mm at PDU2. These values are the approximated maximum uncertainties of $r_{eff,L}$ in the considered size range of 213 nm to 814 nm at PDU1 and PDU2. For the measurement with AN particles of 91 nm in diameter, the uncertainty of $r_{eff,V}$ was estimated to be 0.08 mm, since this was the maximum found for the measurements with AN particles at the vaporizer.

A comparison of ERICA with the LAAPTOF is a logical consequence, since ERICA consists of the basic framework of the LAAPTOF. However, the components that would justify a direct comparison have been replaced with components of a different design. For example, the ERICA contains a different critical orifice, a different ADL, a different optical detection unit (including ellipsoidal reflectors and a different ablation laser (including optics) than the LAAPTOF. The components remaining from the LAAPTOF (the vacuum chamber (including the four-stage TMP), the ADL adjustment mechanics, and the B-ToF-MS) were included in the text (Sect. 2.3):

„The ERICA-LAMS is based on the commercial LAAPTOF (Gemayel et al., 2016; Marsden et al., 2016). However, it had been thoroughly modified, so only the vacuum chamber (including the four-stage TMP), the ADL adjustment mechanics, and the B-ToF-MS remained."

Finally, I have concerns about the dynamic range of the ion detection system in ERICA LAMS. The A/D has only 8bits if vertical dynamic range which equates to 3 orders of magnitude within spectrum signal. This is insufficient in the reviewers experience and will either produce excessive saturation of intense ion signals or the complete loss of minor signals depending on the gain setting. Can the authors comment on this in section 3.5.2?

For each polarity (anions and cations) two channels record the amplified mass spectrometer signal. One channel with a small full range to cover mass spectra of low signal intensities and a second channel with a large full range to cover mass spectra, in case the small channel is saturated. Overall, all four channels are in use. For the cations Channel A is set to 200 mV and Channel B is set to 4 V. For the anions Channel C is set to 100 mV and Channel D to 4 V. During the evaluation, all mass spectra from each channel for small signals (Channel A for cations and Channel C for anions) are checked for saturation. In case a saturation is detected, the channel for large signals (Channel B for cations and Channel D for anions) is used for further evaluation. When no saturation is detected, the spectra from the channel for small signals are used. Both polarities are treated independently for each mass spectrum.

The text in P6 L30 (Sect. 2.4; Number of section refers to the submitted manuscript for review) was revised:

"The two MCP detector outputs for the anions and cations are conditioned and sampled concurrently by two separate channels with different input voltage ranges, an approach for extending the dynamic range of the A-to-D conversion."

was changed to:

"Each of the two MCP outputs, for the anions and cations, is conditioned and sampled simultaneously by two separate channels (two channels for cations and two channels for anions) of different input voltage ranges (full range: cations 200 mV and 4 V, respectively, anions 100 mV and 4 V, respectively), an approach for extending the dynamic range of the A-to-D conversion (Brands et al., 2011)."

**Minor Comments**

Take care to make accurate definitions upfront in the introduction, and then stick to those definition throughout the document.

We checked the entire manuscript for undefined terms and introduced the terms 'Laser Desorption and Ionization (LDI)' and 'Thermal Desorption and Electron impact Ionization (TD-EI)'.

Please check the correct use of commas throughout the document and avoid excessive paragraph length.

The manuscript was revised regarding the use of commas and the length of paragraphs.

The writing style changes part way through the document which is rather odd.

The manuscript was revised regarding the writing style.

**Introduction**

Page 1  ln 35    Chemical composition measurements can provide…

Done

Ln39            Comma after 'in situ' not required

Done

Page 2, Ln 1    Define the 'pulsed laser technique' as 'single particle mass spectrometry (SPMS)'

Reply:

LDI and SPMS were defined and the sentence was changed: "The first method uses a pulsed laser to vaporize and ionize individual submicron to micrometer sized particles by Laser Desorption and Ionization (LDI; Suess and Prather, 1999) for single particle mass spectrometry (SPMS)."

Page2, Line 5    the correct term is 'Thermal Desorption (TD)' and should be used throughout the document.

TD-EI was defined and the sentence changed to:
"The second method is based on the Thermal Desorption and electron impact Ionization (TD-EI) method, to quantitatively measure non-refractory species (sulfate, nitrate, ammonium, chloride, and organic compounds) in ensembles of particles (Drewnick et al., 2005)."

Page2, Ln8        This sentence is a little muddled. Maybe replace 'previous' with 'former'?

"previous method" was replaced by "LDI method"

Page2, Ln10    Froyd et al. (2019) demonstrates a method for quantifying particle classes, not absolute mass concentrations of specific ions. There is an important distinction.

"Within certain limitations this may become possible, if the data of other instruments are included in the analysis (e.g., in Froyd et al. (2019)).

was changed to:

„ Within certain limitations this may become possible, if the data of other instruments are included in the analysis (e.g., Ault et al., 2009; Healy et al., 2012; Gunsch et al., 2018; Köllner et al., 2021)."

Page2, Ln 11    Consider starting a new paragraph

Done

Page2, Ln 30    Perhaps introduce the term 'tandem measurement'

We do not consider the term "tandem measurement" to be appropriate here.

For us the term "tandem measurement" means that two measurements are carried out which, coupled with different approaches, investigate the same thing and thus provide a more comprehensive understanding. A typical tandem measurement is possible using GC-MS (Gas Chromatography–Mass Spectrometry), for example. This type of tandem measurement has not yet been realized with the ERICA. Tandem measurements are only realized when the same particle would be analyzed with both (ERICA-LAMS *and* ERICA-AMS) methods. If only a part of the aerosol is measured with one method and another part with another method, this is not yet a tandem measurement, even if both instruments are connected in a rack and vacuum system, because they are not coupled.

Page2, Ln31    Replace 'repetition rate' with the term 'temporal resolution'

The term 'repetition rate' was replaced with the term 'temporal resolution'.

Page2, Ln37 '    Tandem Instrument'?

We do not consider the term "tandem" to be appropriate here (see our reply to the comment on Page2, Ln 30).

**Instrument Description**

I brief principal of operation required before getting into the detail. Both techniques are sampling to same particle beam with the ERICA AMS at the end of the particle path. The LDI is requires optical detection to size particles and trigger the pulsed laser part way along the path.

Page3, Ln12    More effort should be made to describe Fig1.

The entire paragraph (until line 28, revised manuscript) is intended to be the description of Fig. 1. Thus, we changed as follows:

"The principal configuration of the ERICA with its inlet system, the laser ablation section (denominated as ERICA-LAMS), and the thermal vaporization section (ERICA-AMS) is shown in Fig. 1."

was changed to

"The principal configuration of the ERICA with its inlet system, the LDI section (denominated as ERICA-LAMS), and the TD-EI section (ERICA-AMS) is shown in Fig. 1 and is described in the following."

Page3, Ln12.    Define LAMS and AMS in the introduction or consider changing to Laser desorption ionisation (LDI) and Thermal desorption Ionisation (TDI) therefor highlight the actual distinction between the two techniques.

The laser desorption and ionization method and the thermal desorption and electron impact ionisation method (with the terms LDI and TD-EI) were introduced and explained in Sect. 1. In Sect. 2, the terms ERICA-LAMS and ERICA-AMS were introduced and the methods (LDI and TD-EI) linked to the instrument parts:

"The principal configuration of the ERICA with its inlet system, the laser ablation section (denominated as ERICA-LAMS), and the thermal vaporization section (ERICA-AMS) is shown in Fig. 1."

was changed to:

"The principal configuration of the ERICA with its inlet system, the LDI section (denominated as ERICA-LAMS), and the TD-EI section (ERICA-AMS) is shown in Fig. 1 and is described in the following."

Page3, Ln14    Why is a constant pressure inlet required? Should this have already been introduced as part of the challenges of aircraft measurement?

(Numbers of sections refer to the revised manuscript)

Yes, the reviewer is right, challenges of aircraft operation under conditions of rapidly changing ambient pressure. This is briefly mentioned in Section 2.2 but the detailed explanations are provided in Molleker et al. (2020). For clarification, the abbreviation "CPI" for Constant Pressure Inlet was introduced:

"During aircraft operation the sample air flow is provided by a constant pressure inlet (Molleker et al., 2020) serving as a critical orifice at the instrument's front end."

was changed to:

"During aircraft operation, the sample air flow is provided by a Constant Pressure Inlet (CPI; Molleker et al., 2020) serving as a critical orifice at the instrument's front end (see Sect. 2.2)."

And

"However, in order to achieve a constant pressure in the ADL ($p_{ADL}$ = 4.5 hPa), the mass flow rate needs to be kept constant during flight operations with largely varying ambient pressures (for the M-55 *Geophysica* ranging from ground pressure to 50 hPa). If $p_{ADL}$ is not maintained constant, the transmission of the particles through the inlet into the vacuum system becomes altitude dependent (Zhang et al., 2002). For this purpose, a newly developed, automatically-controlled compressible rubber O-ring setup is deployed (Molleker et al., 2020)."

was changed to

"However, in order to achieve a constant pressure in the ADL ($p_{ADL}$ = 4.5 hPa), the mass flow rate needs to be kept constant during flight operations with largely varying ambient pressures (for the M-55

*Geophysica* ranging from ground pressure to 50 hPa). If $p_{ADL}$ is not maintained constant, the transmission of the particles through the inlet into the vacuum system becomes altitude dependent (Zhang et al., 2002). For this purpose, a newly developed, automatically-controlled compressible rubber O-ring setup, the so-called CPI, is deployed (Molleker et al., 2020)."

Page 3, Ln23      The term 'ion extraction' instead of acceleration would be more appropriate

"The resulting cations and anions are accelerated into a bipolar time-of-flight mass spectrometer (B-ToF-MS) and detected by micro-channel plates (MCPs)."

was changed to

"The resulting cations and anions are extracted into a bipolar time-of-flight mass spectrometer (B-ToF-MS) and detected by micro-channel plates (MCPs)."

Page3, Ln25      Some particles are partially vaporised. What happens to particle fragment and partly ablated material?

This is a very interesting and important question, which up to now could not be studied further, because the optical triggering for the AMS part of ERICA had not been implemented during the time of this study, but is currently work in progress. The so-called OT-AMS (optically triggered AMS) will allow to record quantitative information of single particles. If both MS (LAMS and AMS) are triggered by the detection unit, we will be able to see if a non-ablated remainder of a particle will hit the vaporizer. This was briefly touched upon in the "summary and outlook" section (submitted manuscript, page 24, lines 32 - 41). However, the paragraph was revised:

„For the same point in time, a data acquisition card is triggered and, similar to the procedure with a light scattering probe on the AMS (Cross et al., 2007; Freutel, 2012), the single particle mass spectrum is recorded. In this way it is possible to quantify the non-refractory components of a single particle. In addition, the size information of the measured single particle is obtained by means of the particle flight time between the two PDUs. Here, a future characterization of interest is the ablation laser's effect to the particles that are only partly ablated and the residuals reach the vaporizer of the ERICA-AMS. For this purpose, a method has to be developed to ensure the linkage of the results to the very same particle. Such a procedure needs more implementations and further laboratory studies."

was changed to

„For the same point in time, the data acquisition card is triggered and the single particle mass spectrum is recorded. For the ERICA this mode is called optically triggered AMS (OT-AMS) mode. With the method of the OT-AMS mode, it is possible to quantify the non-refractory components of single particles when the ablation laser is in idle mode. This method is similar to the procedure with a light scattering probe on the AMS (Cross et al., 2007; Freutel et al., 2013). In addition, the size information of the measured single particle is obtained by means of the particle flight time between the two PDUs. One possible future investigation by means of the OT-AMS mode is the ablation laser's effect on the particles that are only partly ablated and where the residuals reach the vaporizer of the ERICA-AMS. This investigation is only

possible with the unique feature, the serial configuration of SMPS and AMS, as in the OT-AMS mode. A method has to be developed to ensure the linkage of the results to the very same particle. Such a procedure needs more implementations and further laboratory studies. "

Page3, Ln28     Un-ablated particles do not pass through the B-TOF-MS section because they are not extracted.

"B-ToF-MS section" was changed to „ablation region".

Page3, Ln31     use 'extracted' instead of 'injected.

Done

Page3, Ln31     C-TOF-MS has not been properly introduced.

The term 'C-ToF-MS' is introduced in Sect. 1 as 'Compact Time-of-Flight Mass Spectrometer'. For clarification, we added the manufacturer:

"The thermal vaporization and electron impact ionization technique were deployed on research aircraft using a C-ToF-MS (Compact Time-of-Flight Mass Spectrometer) beside others by Bahreini et al. (2009), Morgan et al. (2010), Schmale et al. (2010), Brito et al. (2018), Schulz et al. (2018), and Haslett et al. (2019), while a mAMS (mini Aerosol Mass Spectrometer) was used for example by Vu et al. (2016) and Goetz et al. (2018)."

Was changed to

"The TD-EI technique were deployed on research aircraft using a C-ToF-MS (Compact Time-of-Flight Mass Spectrometer from Tofwerk AG, Switzerland) e.g., by Bahreini et al. (2009), Morgan et al. (2010), Schmale et al. (2010), Brito et al. (2018), Schulz et al. (2018), and Haslett et al. (2019), while a mAMS (mini Aerosol Mass Spectrometer) was used for example by Vu et al. (2016) and Goetz et al. (2018)."

Page3, Ln31,     You have to be more specific than 'Detectable particle size' as that would appear to conflict the next sentence. Do you mean you get composition measurement from that size range?

(Number of sections refer to the submitted manuscript for review)

"The detectable particle size range ($d_{va}$) of the ERICA-LAMS is between ~180 nm and 3170 nm (see Sect. 3.3.3). However, the signal-to-noise ratio of optical particle detection is sufficient for particle time-of-flight calibration between 80 nm and 5 μm (see Sect. 3.2)."

was changed to (Number of sections refer to the revised manuscript):

"The particle size range within the 50 % cut-off in detection efficiency ($d_{50}$) of the ERICA-LAMS is between 180 nm and 3170 nm (see Sect. 3.2.2). The signal-to-noise ratio of optical particle detection is sufficient for particle time-of-flight calibration between 80 nm and 5000 nm (see Sect. S4 in the supplement)."

Page3, Ln33    Xu 2017 describes the ACSM – please state that. Is it valid to assume the detectable particle size range is the same as the ACSM? This requires some discussion.

(Number of figures refer to the revised manuscript)

The detectable particle size of the thermal desorption instrument is determined by the transmission and focussing properties of the aerodynamic lens. Therefore, we refer to the paper by Xu et al. (2017), who used the same aerodynamic lens. The fact that they used an ACSM does not make a fundamental difference here. The longer particle flight path in the ERICA compared to the ACSM may cause that small particles that show a wider divergence do not hit the vaporizer to 100%, thereby reducing detection efficiency for small particles. However, as our measurements show (Fig. 12) this is not the case for particles down to 90 nm.

"The detectable particle size range ($d_{va}$) of the ERICA-LAMS is between ~180 nm and 3170 nm (see Sect. 3.3.3). However, the signal-to-noise ratio of optical particle detection is sufficient for particle time-of-flight calibration between 80 nm and 5 μm (see Sect. 3.2). The detectable particle size range of the ERICA-AMS is assumed to be the same as published by Xu et al. (2017) for the deployed lens type.: ~120 nm to 3.5 μm."

was changed to (Numbers of sections and figures refer to the revised manuscript; see also reply to RC1 and RC3):

"The particle size range within the 50 % cut-off in detection efficiency ($d_{50}$) of the ERICA-LAMS is between 180 nm and 3170 nm (see Sect. 3.2.2). The signal-to-noise ratio of optical particle detection is sufficient for particle time-of-flight calibration between 80 nm and 5000 nm (see Sect. S4 in the supplement). For the ERICA-AMS, the detectable particle size range is determined by the transmission and focusing properties of the aerodynamic lens. For the ADL used in our instrument, Xu et al. (2017), who used this lens in combination with an ACSM (Aerosol Chemical Speciation Monitor), determined a transmission range from ~120 nm to 3500 nm. We assume that the detectable particle size range of the ERICA-AMS matches this transmission range."

Page 3, Ln39    Consider putting the final paragraph of this section as part of the introduction.

Done

Page4, Ln 30    Are the vacuum pressures measured or calculated? A schematic of the vacuum system would be helpful.

The presented pressures values were measured. A schematic of the vacuum system and a table of the pressures and pumping rates (read out from the manuals) are now included in the supplement (Sect. S1.2 in the supplement; revised manuscript).

Page5, Ln15    How is the vacuum seal achieved on a movable assembly?

We added following sentence:

"An O-ring around the holding tube for the four aperture rings seals the vacuum at the pivot point."

Page 5, Ln20    How do you know that the system collects 75% of the scattered light. Has this been modelled or measured?

We had to correct the value to 70 %. The value of the total scattered light has been modelled considering Mie-Theory and the geometry of the elliptical reflectors.

"This design collects approximately 75 % of the total scattered light, not considering the losses at the pinholes."

was changed to

"This design collects in maximum 70 % of the total scattered light from a spherical particle (100 nm), according to model calculations adopting Mie theory and using the geometry of the detection unit except for the pinholes (which cause losses)."

Page6, Ln10    What shape beam profile is produced by the pulsed laser system. Is there variation in the power density with respect to position on the particle beam axis?

The beam shape of the ablation laser is considered to be Gaussian. Thus, the power density is depending on the position of the particle beam axis.

"Gaussian beam shape" was added in parenthesis for the detection lasers and the ablation laser in Section 2.1 (revised manuscript)

Following sentence was added (Number of the section refers to the revised manuscript):

"Considering a nearly Gaussian beam shape, as measured and confirmed by the fitting method in Sect. 3.2.1, the power density available to ablate the particle is depending on the position of the particle beam axis."

Page6, Ln29    8bits the effective dynamic range including the noise? This equates to around 3 orders of magnitude.

Yes, the noise is included and is < 1bit. Please note: The text was revised (see answer to ,Major comment' No. 4)

Page6, Ln30    The positive and negative ion signals are measured by separate detection systems. Whilst having different gain on each channel is beneficial, it does not actually increase the dynamic range of the A/D, nor the dynamic range within the spectra. This is misleading.

The text was revised (see answer to ,Major comment' No. 4). The explanation of the extension of the dynamic range should be much clearer now.

Section 2.5    The writing style changes to prose, which is rather odd.

Section 2.5 (submitted manuscript for review) was revised regarding the writing style.

Page 8, Ln1    Replace 'serial configuration' with 'tandem configuration'

We do not consider the term "tandem" to be appropriate here (see our reply to the comment on Page2, Ln 30).

Section 2.6    Is the data for 5% reduction in particle mass on the AMS with LAMS switched on actually presented in this paper? Where?

We removed the statement, since the presentation of this measurement will be part of an upcoming publication about the OT-AMS mode.

Section 3.1    The detection laser beam waist (250um) is much smaller than particle beam, but much larger that the particle diameters. Particles can encounter very different laser fluence depending on their trajectory through the Gaussian profile, therefore the effective irradiance encountered cannot be calculated by diciding the laser power by the beam area. See Marsden et al 2018.

Here, the average irradiance $E_e$ over the beam cross section ($1/e^2$ of intensity) of the laser is presented to provide a value for an instrument-specific parameter. It is calculated by (with beam waist radius $w_0$ and intensity $P$):

$$E_e = P/(\pi * w_0{}^2)$$

The statement from Marsden et al. (2018) that particles can encounter very different laser irradiance depending on their trajectory through the Gaussian profile, since the detection laser beam waist diameter (250 µm) is much larger than the particle diameters was added in the text.

[revised manuscript text omitted]